

# Seasonal variations of triple oxygen isotopic compositions of atmospheric sulfate, nitrate and ozone at Dumont d'Urville, coastal Antarctica

Sakiko Ishino[1], Shohei Hattori[1], Joel Savarino[2,3], Bruno Jourdain[2,3], Susanne Preunkert[2,3], Michel Legrand[2,3], Nicolas Caillon[2,3], Albane Barbero[2,3], Kota Kuribayashi[4], Naohiro Yoshida[1,5]

[1]Department of Chemical Science and Engineering, School of Materials and Chemical Technology, Tokyo Institute of Technology, 4259 Nagatsuta-cho, Midori-ku, Yokohama 226-8502, Japan
[2]Université Grenoble Alpes, Laboratoire de Glaciologie et Géophysique de l'Environnement, 38000 Grenoble, France
[3]CNRS, Laboratoire de Glaciologie et Géophysique de l'Environnement, 38000 Grenoble, France
[4]Department of Environmental Chemistry and Engineering, Tokyo Institute of Technology, 4259 Nagatsuta-cho, Midori-ku, Yokohama 226-8502, Japan
[5]Earth-Life Science Institute, Tokyo Institute of Technology, Meguro-ku, Tokyo 152-8551, Japan

*Correspondence to*: Sakiko Ishino (ishino.s.ab@m.titech.ac.jp)

## Abstract

Reconstruction of the oxidative capacity of the atmosphere is of great importance to understanding climate change, because of its key role in determining the life times of trace gases. Triple oxygen isotopic compositions ($\Delta^{17}O = \delta^{17}O - 0.52 \times \delta^{18}O$) of atmospheric sulfate ($SO_4^{2-}$) and nitrate ($NO_3^-$) in the Antarctic ice cores have shown potential as stable proxies, because they reflect the oxidation chemistry involved in their formation processes. However, observations of $\Delta^{17}O$ values of $SO_4^{2-}$, $NO_3^-$, and ozone in the present-day Antarctic atmosphere are very limited, and their complex chemistry is not fully understood in this region. We present the first simultaneous measurement of $\Delta^{17}O$ values of atmospheric sulfate, nitrate, and ozone collected at Dumont d'Urville station (66°40'S, 140°01'E) throughout 2011. $\Delta^{17}O$ values of sulfate and nitrate exhibited seasonal variation characterized by summer minima and winter maxima, within the ranges of 0.9–3.4 ‰ and 23.0–41.9 ‰, respectively. In contrast, $\Delta^{17}O$ values of ozone showed no significant seasonal variation, with values of 26 ± 1 ‰ through the year. These contrasting seasonal trends suggest that $\Delta^{17}O(O_3)$ is not the major factor determining seasonal changes in $\Delta^{17}O(SO_4^{2-})$ and $\Delta^{17}O(NO_3^-)$ values. The summer/winter trends for $\Delta^{17}O(SO_4^{2-})$ and $\Delta^{17}O(NO_3^-)$ values are caused by sunlight-driven changes in $O_3/RO_X$ ratios, which decrease in summer through ozone destruction and photo-oxidants production, resulting in co-variation between ozone mixing ratios and $\Delta^{17}O(SO_4^{2-})$ and $\Delta^{17}O(NO_3^-)$ values. However, despite similar ranges of ozone mixing ratios in spring (September to November) and fall (March to May), $\Delta^{17}O(SO_4^{2-})$ values observed in spring were lower than in fall. The relatively low sensitivity of $\Delta^{17}O(SO_4^{2-})$ values to the ozone mixing ratio in spring is possibly explained by (i) lower $O_3/RO_X$ ratios caused by $NO_X$ emission from snowpack and/or (ii) $SO_2$ oxidation by hypohalous acids (HOX = HOCl + HOBr) in the aqueous phase.



## 1 Introduction

The reconstruction of changes in the past oxidative capacity of the atmosphere over preindustrial–industrial and glacial–interglacial scales is of great importance for understanding climate change because of its key role in determining the lifetimes of trace gases, such as $CH_4$, CO and hydrochlorofluorocarbons (Alexander and Mickley, 2015; Wang and Jacob,

1998). In early studies, attempts to reconstruct the past oxidative capacity were based on $H_2O_2$ and HCHO in polar ice cores (Neftel et al., 1984; Staffelbach et al., 1991). However, it was found that these two chemical species are not irreversibly trapped in snow and some evaporation and/or decomposition occur after their deposition in surface snow (Hutterli et al., 2001). Thus, the interpretation of past atmospheric concentrations from ice records was not as straightforward as originally thought.

The isotopic composition of atmospheric sulfate ($SO_4^{2-}$) and nitrate ($NO_3^-$) can provide valuable information not available only from their concentrations. In particular, triple oxygen isotopic compositions ($\Delta^{17}O = \delta^{17}O - 0.52 \times \delta^{18}O$) of atmospheric sulfate and nitrate have shown potential as stable proxies of the oxidative capacity of the atmosphere, because they indicate the past relative importance of various oxidation pathways involved in their formations (Erbland et al., 2013; Frey et al., 2009; Kunasek et al., 2010; Sofen et al., 2014). Atmospheric sulfate and nitrate are produced from $SO_2$ and $NO_X$ (= NO +

$NO_2$), respectively, and are the ultimate products of the sulfur and nitrogen cycles as results of oxidation reactions by various oxidants such as OH radicals, $O_3$, $H_2O_2$, $RO_2$ and/or halogen oxides (Seinfeld and Pandis, 2012). In general, $O_3$ oxidation produces sulfate and nitrate possessing high $\Delta^{17}O$ values since $O_3$ has high $\Delta^{17}O$ values approximately 26 ‰ (Vicars et al., 2012; 2014), whereas oxidation by $RO_X$ species such as OH, $RO_2$ and $H_2O_2$ produces sulfate and nitrate with low $\Delta^{17}O$ values since $RO_X$ have $\Delta^{17}O$ values of approximately 0 ‰ (Morin et al., 2011). Thus, the $\Delta^{17}O$ values of both species trapped

in ice cores reflect the past regional oxidation chemistry involved in sulfur and nitrogen cycles, and can provide insight into past oxidant variations (e.g., Alexander et al., 2004; Kunasek et al., 2010; Sofen et al., 2014).

In Antarctica, such oxidative capacity is influenced by complex chemistry, involving halogen chemistry and high oxidant production induced by photolysis of chemical species in the snowpack (Wang et al., 2007; Bloss et al., 2010). Indeed, recent observation of various oxidants in East Antarctica, such as $O_3$ (Legrand et al., 2009), OH and $RO_2$ (Kukui et al., 2012),

revealed that air masses, having strong oxidative capacity formed by snowpack $NO_X$ emission at inland sites, are frequently exported to coastal sites. The coupling of this distinctive oxidative capacity and air mass transport affects the $\Delta^{17}O$ values of both sulfate and nitrate. Seasonal variation of $\Delta^{17}O(NO_3^-)$ values, with summer minima and winter maxima, have been well documented at both coastal and inland Antarctic sites. This variation is explained by increased $NO_X$ (= NO + $NO_2$) oxidation to nitrate by photochemical oxidants such as OH and $RO_2$ in summer, as well as deposition of nitrate contained in polar

stratospheric clouds in winter (Erbland et al., 2013; Frey et al., 2009; Savarino et al., 2007). Only one report of seasonal variation in $\Delta^{17}O(SO_4^{2-})$ values at Dome C, an inland Antarctic site, showed lower values in summer and higher values in spring and fall (Hill-Falkenthal et al., 2013). However, Hill-Falkenthal et al. (2013) observed significant declines in winter $\Delta^{17}O(SO_4^{2-})$ values at Dome C, although mechanisms for these unexpected winter declines are yet to be constrained. Since





this study, $\Delta^{17}O(SO_4^{2-})$ values in the Antarctic atmosphere have not been re-examined. More recently, using a new analytical method developed by Vicars et al. (2012; 2014), year-round observation of $\Delta^{17}O(O_3)$ values at Dome C have been carried out, showing quite stable $\Delta^{17}O(O_3)$ values, close to 26 ‰, with small increases in summer and winter (Savarino et al., 2016). This variation is clearly different from seasonal variations in sulfate and nitrate. Hence, observations of $\Delta^{17}O$ values of sulfate,

nitrate and ozone in the present-day Antarctic atmosphere are very limited. Thus, it is important to investigate the spatial and temporal variations of these isotopic signatures to constrain the atmospheric chemistry, involving sulfate, nitrate and ozone, using simultaneous observations.

In this study, we present the first simultaneous observations of $\Delta^{17}O$ of sulfate, nitrate and ozone at the coastal Antarctic site, Dumont d'Urville Station (DDU) throughout 2011. These measurements were conducted within the framework of the

Oxidant Production over Antarctic Land and its Export project (OPALE; Preunkert et al., 2012), which enabled us to combine the isotopic measurements of sulfate and nitrate with various meteorological and other oxidant observations. The primary purpose of this study was to examine the relationship between $\Delta^{17}O$ values of sulfate and nitrate to determine their complex chemistry in the Antarctic atmosphere.

**2 Samples and analytical methods**

**2.1 Sampling site and aerosol sample collection**

**2.1.1 Sampling site**

Samples were collected at DDU (66°40'S, 140°01'E; 40 m above the sea level), located on a small island, 1 km off the coast of Antarctica. The climate of DDU is described in Konig-Langlo et al. (1998). Compared with other parts of Antarctica,

DDU is temperate, with temperatures ranging from −30 to 5°C throughout the year. Most parts of the island are free of snow, and the sea ice disappears completely during summer. Recent observations of several oxidants, like surface ozone and the OH radical (e.g., Legrand et al., 2016; Kukui et al., 2012), suggest that DDU is exposed to a strong oxidizing atmosphere, coming from the East Antarctic Plateau, where the atmosphere is highly oxidative because of snowpack emission of reactive nitrogen species.

**2.1.2 Aerosol sample collection**

Aerosol samples were collected using a high-volume air sampler (HVAS; General Metal Works GL 2000H Hi Vol TSP; Tisch Environmental, Cleves, OH, USA). Coarse (> 1 μm) and fine (< 1 μm) particles were collected separately, using a four-stage cascade impactor and a backup glass fiber filter, respectively. The slotted 12.7 cm × 17.8 cm glass fiber filters




were mounted on the cascade impactor, while 20.3 cm × 25.4 cm glass fiber filters were used for backup. The HVAS was placed on a platform, 1 m above ground, 50 m from the coast, and 20 m away from the closest building. Aerosol collection was carried out at weekly intervals, with flow rates of ~1.5 m³/min, yielding an average pumped air volume of 15000 m³ per sample. Samples collected between January 2011 and January 2012 were used for this study. Once per month, a field blank

was checked by mounting filters onto the filter holder, and running the cascade impactor for 1 min.

After each collection period, the filters were removed from the cascade impactor inside a clean chemical hood; they were wrapped in aluminum foil and stored in plastic bags at −20°C. The four filters on the impactor stage were grouped together as "coarse" particle samples, while the backup filters were kept as "fine" particle samples. Samples were transported back to Laboratoire de Glaciologie et Géophysique de l'Environnement (LGGE; Grenoble, France) for chemical and isotopic

analyses, while frozen.

### 2.1.3 Quantification of ionic species

The soluble compounds in the aerosols were extracted with ultra-pure water (Millipore filter, 18 MΩcm; EMD Millipore, MA, USA), according to the process described in Savarino et al. (2007); more than 98% of the initial water volume was recovered. Field blank filters were processed in the same way.

Small aliquots of these sample solutions were taken for quantification of ionic species. Anions (Cl⁻, $NO_3^-$, $SO_4^{2-}$) and sodium ($Na^+$) concentrations were analyzed using ion chromatography systems described in Savarino et al. (2007) and Jourdain and Legrand (2002), respectively. The accuracy of these analyses is determined by the accuracy of the ion chromatography, which is typically 5%.

Atmospheric concentrations of these ionic species were calculated using the aerosol filter loading, and the air volume

pumped through the filter. The air volume was corrected to standard temperature and pressure (T = 273.15 K, p = 101325 Pa) based on meteorological data from DDU provided by Meteo France. The uncertainties for atmospheric concentrations were estimated using the minimum and maximum values of all filter blank measurements.

### 2.2 Oxygen isotopic analyses of sulfate and nitrate in aerosols

### 2.2.1 Definition of triple oxygen isotopic compositions

Given the two isotope ratios, notated as $^{17}R$ (= $^{17}O/^{16}O$) and $^{18}R$ (= $^{18}O/^{16}O$), stable oxygen isotope ratios are conventionally scaled using a delta ($\delta$) notation:

$$\delta\ ^xO = \frac{^xR_{sample}}{^xR_{VSMOW}} - 1 \qquad (1)$$





where $R_{VSMOW}$ denotes the isotope ratio of the standard material, Vienna Standard Mean Ocean Water (VSMOW); and $x$ is 17 or 18. Despite the robust relationship of the mass-dependent law ($\delta^{17}O = 0.52 \times \delta^{18}O$) in most of the oxygen-containing species, (e.g., $O_2$ and $H_2O$), atmospheric ozone does not follow mass-dependent fractionation and possesses a significant positive $\Delta^{17}O$ (= $\delta^{17}O - 0.52 \times \delta^{18}O$), inherited from mass-independent fractionation associated with its formation process

(Gao and Marcus, 2001). Since non-zero $\Delta^{17}O$ values can be observed in various atmospheric species bearing oxygen atoms inherited from $O_3$ (e.g., sulfate and nitrate), the $\Delta^{17}O$ signature is a powerful tracer, used to investigate the relative contribution of $O_3$ to oxidation processes.

### 2.2.2 Oxygen isotopic analysis of sulfate and data correction

All $\Delta^{17}O$ values of sulfate were measured with an isotope ratio mass spectrometer (IRMS) (MAT253; Thermo Fisher

Scientific, Bremen, Germany), coupled with an in-house measurement system at Tokyo Institute of Technology. The measurement system for $\Delta^{17}O(SO_4^{2-})$ follows Savarino et al. (2001), with modifications described in several studies (Schauer et al., 2012; Geng et al., 2013). Briefly, 1 μmol of sulfate is separated from other ions using ion chromatography and chemically converted to silver sulfate ($Ag_2SO_4$). This $Ag_2SO_4$ powder is transported in a custom-made quartz cup, which is dropped into a furnace at 1000°C within a high temperature conversion elemental analyzer (TC/EA; Thermo Fisher

Scientific, Bremen, Germany) and thermally decomposed into $O_2$ and $SO_2$. Gas products from this sample pyrolysis are carried by ultrahigh-purity He (>99.99995 % purity; Japan Air Gases Co., Tokyo, Japan), which is first purified using a molecular sieve (5Å) held at −196°C (Hattori et al., 2015). The gas products $O_2$ and $SO_2$ are carried through a cleanup trap (trap 1) held at −196°C to trap $SO_2$ and trace $SO_3$, while $O_2$ continues to another molecular sieve (5Å) in a 1/16 inch o. d. tubing trap (trap 2) held at −196°C to trap $O_2$ separately from the other gas products. The $O_2$ is purified using a gas

chromatograph, with a CP-Molsieve (5Å) column (0.32 mm i.d., 30 m length, 10 μm film; Agilent Technologies Inc., Santa Clara, CA, USA) held at 40°C, before being introduced to the IRMS system to measure $m/z$ = 32, 33, and 34. The inter-laboratory calibrated standards (Sulf-α, β and ε; Schauer et al., 2012) were used to assess the accuracy of our measurements; our values were in good agreement with published ones (Fig. 1). The precision of $\Delta^{17}O$ is typically better than ±0.2 ‰ based on replicate analyses of the standards.

Since sea salt sulfate aerosols (ss-$SO_4^{2-}$) are of little importance to atmospheric sulfur oxidation processes (i.e., $\Delta^{17}O$(ss-$SO_4^{2-}$) = 0‰), both total sulfate concentrations and $\Delta^{17}O$ values were corrected for their ss-$SO_4^{2-}$ component to obtain their non-sea salt sulfate (nss-$SO_4^{2-}$) content, using Eq. (2) and (3) below.

$$[nss - SO_4^{2-}] = [total - SO_4^{2-}] - k \times [Na^+] \tag{2}$$

$$\Delta^{17}O(nss - SO_4^{2-}) = \frac{[total - SO_4^{2-}]}{[nss - SO_4^{2-}]} \times \Delta^{17}O(total - SO_4^{2-}) \tag{3}$$

where "total" is the quantity measured by ion chromatography, corresponding to the sum of ss- and nss-$SO_4^{2-}$ components; and $k$ is the mass ratio of $[SO_4^{2-}]/[Na^+]$ in sea water (0.25; Holland et al., 1986). To take into account sea salt fractionation





processes that affect the Antarctic region in winter, when temperatures drop below $-8°C$ in the presence of sea-ice (Wagenbach et al., 1998), we used a $k$ value of $0.13\pm0.04$, estimated from the average at winter DDU previously by Jourdain and Legrand (2002); this was applied to samples collected from May to October.

The corrections for $\Delta^{17}O(\text{nss-SO}_4^{2-})$ values were only carried out for the fine mode samples, because sulfate in the coarse mode samples consists of more than 80% $\text{ss-SO}_4^{2-}$. Eq. (3) is the isotope mass balance equation between ss- and $\text{nss-SO}_4^{2-}$, with $\Delta^{17}O(\text{ss-SO}_4^{2-}) = 0$ ‰. The influence of penguin excrement on $Na^+$ concentration was not taken into account, since it mainly affects supermicron (coarse) aerosols (Jourdain and Legrand, 2002). The total uncertainties for $\Delta^{17}O(\text{nss-SO}_4^{2-})$ values were calculated using the precision of $\Delta^{17}O$ measurement and the uncertainty of $k$ value.

### 2.2.3 Oxygen isotopic analysis of nitrate

The $\Delta^{17}O$ value of nitrate was measured simultaneously with $\delta^{18}O$ and $\delta^{15}N$ values using a bacterial denitrifier method (Casciotti et al., 2002), coupled with IRMS measurement using our in-house peripheral system at LGGE (Morin et al., 2009). All nitrates in our samples were converted to $N_2O$ via bacterial denitrification. This $N_2O$ was introduced to the measurement system, separated from $CO_2$, $H_2O$ and other volatile organic compounds, and pre-concentrated in a cold trap. The trapped $N_2O$ was converted into $O_2$ and $N_2$ by pyrolysis at $900°C$, using a gold tube furnace, followed by separation of $O_2$ and $N_2$ via a 10 m Molsieve (5Å) gas chromatography column, before being introduced to the IRMS system. Measurements were performed simultaneously for samples equivalent to 100 nmol nitrate, as well as a subset of international nitrate reference materials (US Geological Survey 32, 34, and 35, as well as their mixtures) for correction and calibration of $\Delta^{17}O$ and $\delta^{18}O$ values relative to VSMOW and $\delta^{15}N$ values relative to air $N_2$. Analytical uncertainty was estimated based on the standard deviation of the residuals from a linear regression between the measured reference materials and their expected values. The uncertainties ($1\sigma$) for $\Delta^{17}O(\text{NO}_3^-)$ and $\delta^{15}N(\text{NO}_3^-)$ were 0.4 ‰ and 0.3 ‰, respectively.

### 2.3 Sampling and analytical methods of oxygen isotopic composition of ozone

The sampling and isotopic analysis of surface ozone were performed by coupling the nitrite-coated filter method with nitrate isotopic measurements described in Vicars et al. (2012; 2014). The principle of ozone collection underlying this technique is the filter-based chemical trapping of ozone via its reaction with nitrite:

$$NO_2^- + O_3 \rightarrow NO_3^- + O_2 \quad . \tag{R1}$$

During R1, one of the three oxygen atoms of nitrate is transferred from one of the two terminal oxygen atoms of ozone, while the other two oxygen atoms are derived from the reagent nitrite. Since the $\Delta^{17}O$ signature of ozone is located only on the terminal atoms of ozone (Bhattacharya et al., 2008; Janssen and Tuzson, 2006), simple mass balance implies that $\Delta^{17}O(O_3)_{\text{term}}$ is 2/3 of $\Delta^{17}O(O_3)_{\text{bulk}}$. Thus, $\Delta^{17}O(O_3)_{\text{term}}$ values can be inferred using the simple mass-balance of Eq. (4):

$$\Delta^{17}O(O_3)_{\text{term.}} = 3 \times \Delta^{17}O(NO_3^-) - 2 \times \Delta^{17}O(NaNO_2), \tag{4}$$





where $\Delta^{17}O(NaNO_2)$ of the reagent is confirmed to be zero (Vicars et al., 2012). Therefore, the $\Delta^{17}O$ value of ozone can be determined from the oxygen isotopic composition of nitrate produced on the coated filter via R1, determined by the same measurement system described above.

Ozone sampling was carried out by pumping ambient air, using a low-volume vacuum pump (Model 2522C-02; Welch, IL,

USA), through a glass fiber filter (Ø 47 mm, GF/A type; Whatman, UK), pre-coated with a mixture of $NaNO_2$, $K_2CO_3$ and glycerol. Sampling was conducted once per week from May 2011 to April 2012, with 24–48 h sampling intervals. After sampling, filter samples and procedural blanks were extracted in 18MΩ water. Any unreacted nitrite reagent was removed using the reaction with sulfamic acid, neutralized later with NaOH solutions (Granger and Sigman, 2009; Vicars et al., 2012). The sample solutions were stored in the dark at −20°C, and transported back to Grenoble. After nitrate concentration analysis

using a colorimetric technique (Frey et al., 2009), the isotopic analysis of nitrate (i.e., ozone) was performed using the same protocol as the nitrate isotope analysis. In addition to the isotope measurements of ozone, we aligned the mixing ratio of surface ozone to the weekly average using data reported in Legrand et al. (2016a) to fit the time resolution of our aerosol sampling.

### 2.4 Complementary analyses

To investigate relationships between the origins of the air masses and the $\Delta^{17}O$ signatures of sulfate and nitrate, transport pathways of sampled air masses were analyzed using the HYSPLIT (Hybrid Single-Particle Lagrangian Integrated Trajectory) model (Draxler and Rolph, NOAA Air Resources Laboratory, Silver Spring, Maryland, 2003; available at http://ready.arl.noaa.gov/HYSPLIT.php). Five-day backward trajectories for air masses arriving at the DDU at an altitude of 40 m above sea level were computed twice per day for each day during sampling periods.

The sea ice area fraction around the Antarctic continent was derived from the Advanced Microwave Scanning Radiometer on-board NASA's Earth Observing System Aqua satellite using the ARTIST sea ice algorithm (Kaleschke et al., 2001). The contact times of these air masses with the Antarctic continent and sea ice were calculated using five-day backward trajectories and sea ice area fractions.

## 3 Results

### 3.1 Sulfate

Seasonal variations in atmospheric concentrations and $\Delta^{17}O$ values of $SO_4^{2-}$ are shown in Fig. 2a. Atmospheric concentrations of nss-$SO_4^{2-}$ showed a clear seasonal trend. The [nss-$SO_4^{2-}$] had a summer maximum of up to ~280 ng m$^{-3}$ from January to February, but decreased to a background level (~10 ng m$^{-3}$) during May to August, before increasing as





summer returned. This trend in [nss-SO$_4^{2-}$] at coastal Antarctic sites results from enhanced marine biogenic activity, emitting dimethyl sulfide (DMS) in circum-Antarctic regions in summer,.as often been reported previously (e.g., Wagenbach et al., 1998; Minikin et al., 1998; Jourdain and Legrand, 2002; Preunkert et al., 2008). As Antarctica is surrounded by ocean, DMS is the major source of atmospheric non-sea salt sulfur (Minikin et al., 1998; Jourdain and Legrand, 2002). Interestingly, a sample from 18–25 July had an anomalously high value of 46 ng m$^{-3}$, four times the monthly mean level for July (~12 ng m$^{-3}$).

The $\Delta^{17}$O(nss-SO$_4^{2-}$) values showed the reverse trend, with a summer minimum and a winter maximum. The $\Delta^{17}$O(nss-SO$_4^{2-}$) value increased from 1.0 ‰ observed in January to a maximum of 3.4 ‰ at the end of June, decreasing to 0.9 ‰ in December. The annual weighted mean value of $\Delta^{17}$O(nss-SO$_4^{2-}$) was 1.4 ± 0.1 ‰. Higher values (greater than 2 ‰) were generally observed during April to July, but the anomalous peak from 18–25 July was characterized by a low $\Delta^{17}$O(nss-SO$_4^{2-}$) value of 0.9 ‰. Consequently, the monthly mean value had a maximum in July (2.6 ± 0.6 ‰), when the 18–25 July data were excluded.

### 3.2 Nitrate

Seasonal variations in atmospheric concentrations and $\Delta^{17}$O values for nitrate are shown in Fig. 2b. Nitrate concentrations increased to 55 ng m$^{-3}$ in January but gradually decreased to less than 10 ng m$^{-3}$ in March to May. In July, a significant peak of 28 ng m$^{-3}$ was observed, followed by a seasonal increase as summer returned. The $\Delta^{17}$O(NO$_3^-$) values showed a simple seasonal variation, with a summer minimum and a winter maximum. $\Delta^{17}$O(NO$_3^-$) increased from 27 ‰ in January to over 40 ‰ in July, decreasing moderately to a minimum value of 23 ‰ in December. These trends in nitrate concentrations and $\Delta^{17}$O(NO$_3^-$) values are consistent with those observed at this site 10 years ago (Savarino et al., 2007).

### 3.3 Ozone

Daily averaged ozone mixing ratios are presented in Fig. 2c; these exhibit a distinct seasonal variation, with a summer minimum and a winter maximum. The minimum ozone mixing ratio was observed in January, having a value lower than 10 ppbv, while the maximum was observed during July to August, having a value higher than 35 ppbv. From November to December, sudden increases in ozone levels to values over 30 ppbv were observed a few times, consistent with seasonal trends for ozone at DDU (Legrand et al., 2009). The $\Delta^{17}$O(O$_3$)$_{bulk}$ values showed an insignificant variation, with a summer maximum of 28 ‰ and a winter minimum of 23 ‰, and an annual mean $\Delta^{17}$O(O$_3$)$_{bulk}$ value of 26 ± 1 ‰.



## 4 Discussion

### 4.1 $\Delta^{17}O$ values and atmospheric formation pathways of sulfate and nitrate

#### 4.1.1 $\Delta^{17}O$ values of sulfate

Since $SO_2$ quickly exchanges its oxygen atoms with abundant water vapor in the atmosphere, the $\Delta^{17}O(SO_2)$ value is assumed to be 0 ‰ (Holt et al., 1983). Thus, the $\Delta^{17}O(SO_4^{2-})$ value is dependent only on the oxidation pathway of $SO_2$ to $SO_4^{2-}$. $SO_2$ oxidation by OH ($\Delta^{17}O(OH) = \sim0$ ‰) in the gas phase produces sulfuric acid ($H_2SO_4$) which possesses the $\Delta^{17}O(SO_4^{2-})$ value of approximately 0‰.

$$SO_2 + OH \xrightarrow{O_2,H_2O,M} H_2SO_4 \qquad \Delta^{17}O(SO_4^{2-}) = 0\ ‰ \qquad \text{(R2)}$$

$SO_2$ can also dissolve into the aqueous phase on aerosol surfaces, where it can react with various oxidants ($O_3$, $H_2O_2$ or $O_2$) catalyzed by transition metal ions (such as Fe(III) and Mn(II)) to form sulfate (Seinfeld and Pandis, 2012). Given that $\Delta^{17}O(O_3)_{bulk}$ values of approximately 26 ‰ have been observed (Vicars et al., 2014), the $\Delta^{17}O(SO_4^{2-})$ value of sulfate produced by ozone should be around 6.5 ‰, based on a $\Delta^{17}O$ signature transfer factor of 0.25 (Savarino et al., 2000).

$$SO_3^{2-} + O_3 \longrightarrow SO_4^{2-} + O_2 \qquad \Delta^{17}O(SO_4^{2-}) = 6.5\ ‰ \qquad \text{(R3)}$$

Give the $\Delta^{17}O(H_2O_2)$ values of 1.6 ‰ on average (Savarino et al., 1999), the $\Delta^{17}O(SO_4^{2-})$ of sulfate produced by $H_2O_2$ is estimated to be 0.8 ‰, using a transfer factor of 0.5 (Savarino et al., 2000).

$$HSO_3^- + H_2O_2 \longrightarrow HSO_4^- + H_2O \qquad \Delta^{17}O(SO_4^{2-}) = 0.8‰ \qquad \text{(R4)}$$

The $\Delta^{17}O(O_2)$ value was measured to be $-0.3$ ‰ (Barkan and Luz, 2003), producing sulfate with a $\Delta^{17}O(SO_4^{2-})$ value of almost 0 ‰ (Savarino et al., 2000).

$$SO_3^{2-} + O_2 \xrightarrow{Fe,Mn} SO_4^{2-} \qquad \Delta^{17}O(SO_4^{2-}) = -0.1\ ‰ \qquad \text{(R5)}$$

Thus, $\Delta^{17}O(SO_4^{2-})$ of nss-$SO_4^{2-}$ results from a subtle balance between various oxidation reactions (R2–R5), each one transferring a specific amount of $\Delta^{17}O$ signature to sulfate.

#### 4.1.2 $\Delta^{17}O$ values of nitrate

The $\Delta^{17}O(NO_3^-)$ value is dependent on both the $\Delta^{17}O(NO_2)$ value and the oxidation pathways of $NO_2$ to $NO_3^-$. The $\Delta^{17}O(NO_2)$ is determined by the relative abundance of $O_3$, $RO_2$, and BrO. Since all non-zero $\Delta^{17}O$ of ozone is positioned in the terminal oxygen atoms (Battacharya et al., 2008), which preferentially react with NO (Savarino et al., 2008), $NO_2$ formed by ozone exhibits a higher isotopic value than the bulk $\Delta^{17}O(O_3)$. The NO + BrO pathway may also produce $NO_2$ with high $\Delta^{17}O$ values (Morin et al., 2007), because BrO is thought to possess the terminal oxygen atom of ozone. Following the




principle that two of the three oxygen atoms in $NO_3^-$ come from $NO_2$ and one arises through conversion of $NO_2$ to $NO_3^-$, the $\Delta^{17}O(NO_3^-)$ value of nitrate produced by each pathway can be expressed as Eq. (5).

$$\Delta^{17}O(NO_3^-) = \frac{2}{3} \times \Delta^{17}O(NO_2) + \frac{1}{3} \times \Delta^{17}O(Oxidant) \qquad (5)$$

For a given value of $\Delta^{17}O(NO_2)$, the $NO_2$ + OH pathway produces the lowest $\Delta^{17}O(NO_3^-)$ value, while the $NO_3$ + RH

pathway or $BrONO_2$ hydrolysis produce the highest $\Delta^{17}O(NO_3^-)$ values.

### 4.2 General trend of seasonal variations in $\Delta^{17}O$ values of sulfate and nitrate

The $\Delta^{17}O$ signatures of atmospheric sulfate and nitrate originate from the oxygen transfers from ozone, via oxidation of their precursors. Thus, changes in $\Delta^{17}O(O_3)$ values likely affect both $\Delta^{17}O(nss\text{-}SO_4^{2-})$ and $\Delta^{17}O(NO_3^-)$ values. However, given the small seasonal variability of the $\Delta^{17}O(O_3)_{bulk}$ (ca. 5 ‰), and assuming that all oxygen atoms transferred to $SO_2$ and $NO_X$ are

from the terminal oxygen of ozone, the expected variability of $\Delta^{17}O(nss\text{-}SO_4^{2-})$ and $\Delta^{17}O(NO_3^-)$ should not exceed 1.3 ‰ and 7.5 ‰, respectively. Clearly, these upper limits do not explain the 2.5 ‰ and 19 ‰ seasonal variability observed in $\Delta^{17}O(nss\text{-}SO_4^{2-})$ and $\Delta^{17}O(NO_3^-)$ at DDU, respectively. Furthermore, the seasonal variation in $\Delta^{17}O(O_3)$ values, with a summer maximum and a winter minimum, is the reverse pattern to $\Delta^{17}O(nss\text{-}SO_4^{2-})$ and $\Delta^{17}O(NO_3^-)$ values, with summer minima and winter maxima. These inconsistencies suggest that variability in $\Delta^{17}O(O_3)$ values is not the major factor

influencing the seasonal variations in $\Delta^{17}O(nss\text{-}SO_4^{2-})$ and $\Delta^{17}O(NO_3^-)$ values.

Meanwhile, $\Delta^{17}O(nss\text{-}SO_4^{2-})$ and $\Delta^{17}O(NO_3^-)$ values are dependent on the relative importance of various oxidation pathways involved in their formation as described in the previous sections. Since the relative importance of these oxidation pathways is sensitive to the relative concentrations of oxidants in the atmosphere, and there is seasonal variation for ozone mixing ratios at a continental scale (Crawford et al., 2001; Legrand et al., 2009), the mixing ratio of ozone is expected to correlate

with $\Delta^{17}O(nss\text{-}SO_4^{2-})$ and $\Delta^{17}O(NO_3^-)$ values. Indeed, $\Delta^{17}O(nss\text{-}SO_4^{2-})$ and $\Delta^{17}O(NO_3^-)$ values, as well as ozone mixing ratios, all display similar seasonal variations, as shown in Fig. 2. The seasonal variation in the ozone mixing ratios at DDU is generally explained by accumulation of ozone in winter, and its photochemical destruction in summer (Legrand et al., 2009; 2016a), which produces $RO_X$ species (OH, $HO_2$, $RO_2$, $H_2O_2$) in the summer period. Therefore, we propose that seasonal variations in $\Delta^{17}O(nss\text{-}SO_4^{2-})$ and $\Delta^{17}O(NO_3^-)$ result from a shift in oxidation pathways from $O_3$ to $RO_X$. Decreases in

$\Delta^{17}O(nss\text{-}SO_4^{2-})$ and $\Delta^{17}O(NO_3^-)$ values are caused by the combining effect of decrease in the ozone concentration and decrease in the transfer efficiency of $\Delta^{17}O(O_3)$ to the final products. Thus, the changes in relative concentrations of $O_3$ and $RO_X$, along with the changes in sunlight level, are the main factors controlling the seasonal variations of $\Delta^{17}O(nss\text{-}SO_4^{2-})$ and $\Delta^{17}O(NO_3^-)$ values.

A similar seasonal variation in $\Delta^{17}O(nss\text{-}SO_4^{2-})$ values has been observed at Dome C, an inland Antarctic site (Hill-

Falkenthal et al., 2013). However, the $\Delta^{17}O(nss\text{-}SO_4^{2-})$ values observed at Dome C significantly declined in July and August,





in contrast to our observations that showed only a single significant decline in $\Delta^{17}O(\text{nss-SO}_4^{2-})$ values during the period of 18–25 July. This low $\Delta^{17}O(\text{nss-SO}_4^{2-})$ sample is also characterized by high nss-SO$_4^{2-}$ concentration (Fig. 2a). We don't have any evidence of contamination from station activities or laboratory works. The preliminary result of sulfur isotope analysis of sulfate in the same sample, showing $\delta^{34}S$ value of 17.6 ‰ (to be published), suggests that this sulfate results from marine

biogenic sulfur (i.e., DMS) which possesses $\delta^{34}S$ values ranging 16–20 ‰ (Oduro et al., 2012; Amrani et al., 2013). However, the results of the back trajectory analyses exhibit that DDU was under continental outflow condition over this period, as well as throughout July (Fig. 3). It is hence difficult to identify the origin of this low $\Delta^{17}O(\text{nss-SO}_4^{2-})$ value during 18–25 July. Nevertheless, this point doesn't change the interpretation that $\Delta^{17}O(\text{nss-SO}_4^{2-})$ values are generally lower in summer and higher in winter. Thus, this sample was excluded from consideration for the following discussion and does not

impair our further interpretation.

### 4.3 Sensitivity of sulfate and nitrate $\Delta^{17}O$ values to the ozone mixing ratio

As shown in Fig. 4, $\Delta^{17}O(\text{nss-SO}_4^{2-})$ and $\Delta^{17}O(\text{NO}_3^-)$ values are co-varied with ozone mixing ratios, suggesting that a change in the ozone mixing ratio is one of the main factors controlling $\Delta^{17}O(\text{nss-SO}_4^{2-})$ and $\Delta^{17}O(\text{NO}_3^-)$ values. It is important to mention that DMS levels are quite low in the dark polar winter (Preunkert et al. 2007; Legrand et al., 2001);

hence, the sulfate collected at DDU is likely produced at a lower latitude region under more sunlight. Thus, the observed $\Delta^{17}O(\text{nss-SO}_4^{2-})$ values in winter may not solely reflect oxidation chemistry in the local atmosphere at DDU. It is also known that $\Delta^{17}O(\text{NO}_3^-)$ values are not representative of the oxidative capacity of the atmosphere at this site, because nitrate deposition from polar stratospheric clouds occurs during the Antarctic winter (Wagenbach et al., 1998; Savarino et al., 2007). Thus, in winter, $\Delta^{17}O(\text{nss-SO}_4^{2-})$ and $\Delta^{17}O(\text{NO}_3^-)$ values are not directly linked to ozone mixing ratios at DDU.

Excluding winter data (June to August), the regression lines for sulfate and nitrate have slopes, intercepts and correlation coefficients ($R^2$) of 0.07, −0.01 and 0.50, and 0.55, 18.2 and 0.80, respectively (Fig. 4). These slopes indicate how sensitive $\Delta^{17}O(\text{nss-SO}_4^{2-})$ and $\Delta^{17}O(\text{NO}_3^-)$ values are to ozone mixing ratios. Meanwhile, the intercepts indicate $\Delta^{17}O(\text{nss-SO}_4^{2-})$ and $\Delta^{17}O(\text{NO}_3^-)$ values when assuming ozone was not involved in the oxidation of SO$_2$ and NO$_2$. If this were true, it would imply that the seasonal cycles of $\Delta^{17}O(\text{nss-SO}_4^{2-})$ and $\Delta^{17}O(\text{NO}_3^-)$ are controlled largely by the last stage of oxidation and

not by the $\Delta^{17}O$ values of their precursors. In the case of sulfate, this is likely to be true, given that $\Delta^{17}O(\text{SO}_2) = 0$ ‰ year-round, because of oxygen isotopic exchange between SO$_2$ and water vapor. Therefore, the intercept for sulfate of approximately 0 ‰ is consistent with $\Delta^{17}O(\text{nss-SO}_4^{2-})$ values for sulfate produced via the SO$_2$ + OH pathway. Meanwhile, the intercept of approximately 18 ‰ for nitrate (Fig. 4b) corresponds to a $\Delta^{17}O(\text{NO}_2)$ value of 27 ‰ from Eq. (5), assuming all nitrate is produced via the NO$_2$ + OH pathway. This $\Delta^{17}O(\text{NO}_2)$ value is in a good agreement with the value of 26 ‰

calculated for conditions at 80°N, under the assumption that $\Delta^{17}O(\text{OH}) = 0$ ‰ (Morin et al, 2011). Using the calculation processes of Morin et al. (2011), we calculated $\Delta^{17}O(\text{NO}_2)$ to be 26.5 ‰, consistent with the intercept value. In this




calculation, we employed DDU conditions for summer, with a mean temperature of about 0°C, $\Delta^{17}O(O_3)$ = 26 ‰, mixing ratios of $O_3$ and $RO_2$ of $5.6 \times 10^{11}$ molecules cm$^{-3}$ and $3.3 \times 10^8$ molecules cm$^{-3}$, respectively (Legrand et al., 2009; Kukui et al., 2012). Thus, the plot of $\Delta^{17}O(NO_3^-)$ values as a function of ozone mixing ratios has potential to be used for estimating $\Delta^{17}O(NO_2)$ values, which they have never been measured.

The sensitivity of $\Delta^{17}O(\text{nss-}SO_4^{2-})$ values to ozone mixing ratios in spring (September to November) seems to be lower than in fall (March to May) (Fig. 4). Fixing the intercept to −0.01, the slopes between $\Delta^{17}O(\text{nss-}SO_4^{2-})$ values and ozone mixing ratios for spring and fall were $0.063 \pm 0.004$ and $0.084 \pm 0.005$, respectively, showing a clear difference over their range of deviation. In contrast to $\Delta^{17}O(\text{nss-}SO_4^{2-})$ values, the slopes between $\Delta^{17}O(NO_3^-)$ values and ozone mixing ratios for an intercept fixed to 18.2 show less difference between spring and fall, with the values of $0.54 \pm 0.02$ and $0.57 \pm 0.01$,

respectively. However, $\Delta^{17}O(\text{nss-}SO_4^{2-})$ is also sensitive to $RO_X$. If the relative abundance of $O_3$ and $RO_X$ are regulated only by changes in solar irradiation as discussed in section 4.2, then the slopes between $\Delta^{17}O$ values for sulfate and nitrate against ozone mixing ratios should be the same for spring and fall. The different slopes observed for sulfate in spring and fall indicate the effects of various oxidation processes, decreasing $\Delta^{17}O(\text{nss-}SO_4^{2-})$ in spring and/or increasing $\Delta^{17}O(\text{nss-}SO_4^{2-})$ in fall. There are several processes that could explain such a spring/fall difference.

One possible explanation involves the influence of $NO_X$ emissions from snowpack covering the East Antarctic Plateau (Davis et al., 2001; Crawford et al., 2001; Chen et al., 2001). The Antarctic atmosphere is strongly affected by $NO_X$ emissions from snowpack, starting at the beginning of spring with the return of the sunlight. These snow $NO_X$ emissions subsequently enhance both $O_3$ and OH productions, with OH in greater proportion than $O_3$. This is particularly true at DDU in summer, where katabatic air from the East Antarctic Plateau causes the ozone mixing ratio to be in the range of 10–40

ppbv (Legrand et al., 2009; 2016), with a mean OH concentration of $2.1 \times 10^6$ molecules cm$^{-3}$ (Kukui et al., 2012). In contrast, Palmer Station is exposed to oceanic air, producing ozone mixing ratios within the range of 9–20 ppbv, with a mean OH concentration of about $1.0 \times 10^5$ molecules cm$^{-3}$ (Jefferson et al., 1998). These observations suggest that $NO_X$ emissions increase oxidants at DDU, compared with the Palmer Station, about 2-fold for $O_3$ and more than 10-fold for OH. Chemical transport models over the Antarctic continent show that $NO_X$ emitted from snow during summer increase $O_3$ and OH by a

factor of 2 and 7, respectively, compared with estimations not including snow $NO_X$ emissions (Zatko et al., 2016). Additionally, [15]N depletion in nitrate starts from the beginning of September (See Supplement), which is consistent with previous measurement at DDU (Savarino et al., 2007), supporting that the snow $NO_X$ emission happens in early spring at DDU. Thus, by snow $NO_X$ emission, OH production is enhanced more efficiently than $O_3$ production in spring, possibly resulting in lower $\Delta^{17}O(\text{nss-}SO_4^{2-})$ spring values.

Another possible explanation is that hypohalous acids (HOX = HOCl, HOBr) act as important oxidants of $SO_2$ via the aqueous phase reaction (R6 and R7) in the marine boundary layer (Fogelman et al., 1989; von Glasow, 2002):

$$HOX + SO_3^{2-} \rightarrow OH^- + XSO_3^{2-} \qquad , \qquad\qquad\qquad\qquad (R6)$$

$$XSO_3^{2-} + H_2O \rightarrow SO_4^{2-} + X^- + 2H^+ \qquad . \qquad\qquad\qquad\qquad (R7)$$





This reaction is expected to produce sulfate with $\Delta^{17}O = 0$ ‰, as all oxygen atoms of sulfate originate from water (Fogelman et al., 1989; Troy and Margerum, 1991: Yiin and Margerum, 1988), leading to lower $\Delta^{17}O(\text{nss-SO}_4^{2-})$ values in the atmosphere. Indeed, unexpectedly low $\Delta^{17}O(\text{nss-SO}_4^{2-})$ values have been observed in marine aerosols, which is possibly explained by a contribution of HOX oxidation of 33%–50% to total sulfate production in the marine boundary layer (Chen et

al., 2016). Chen et al. (2016) estimated that a minimum concentration of gaseous HOX of 0.1 pptv could account for half of the sulfate production in the marine boundary layer. At DDU, year-round observations of gaseous inorganic bromine species ($[\text{Br}_y{}^*] = [\text{HBr}] + [\text{HOBr}] + 0.9[\text{Br}_2] + 0.4[\text{BrO}] + [\text{BrNO}_2] + [\text{BrONO}_2] + [\text{Br}]$) revealed that maximum concentrations are observed in September, with values of $13.0 \pm 6.5$ ng m$^{-3}$ (~ 3.6 pptv) (Legrand et al., 2016b). Even if only one third of the $\text{Br}_y{}^*$ corresponds to HOBr, as estimated using our model calculations under summer conditions, then it is expected that HOX

at DDU in spring is > 1 pptv; thus, HOX could play a significant role in sulfate production in spring. Likewise, $\text{Br}_y{}^*$ concentration at DDU is at minimum in May, with values of $3.4 \pm 1.0$ ng m$^{-3}$ (Legrand et al., 2016b), corresponding to less than one third of spring values. This would lead to a lower contribution of HOX to sulfate production in fall, compared with spring. Hence, sulfate production via aqueous oxidation by HOX may explain the lower $\Delta^{17}O(\text{SO}_4^{2-})$ in spring, relative to fall.

A change in pH of the aqueous phase on the aerosol surfaces may also explain the spring/fall difference in $\Delta^{17}O(\text{nss-SO}_4^{2-})$. Given that $\text{SO}_2$ oxidation by ozone in the aqueous phase is favored at high pH (>5.5) (Seinfeld and Pandis, 2012), if the pH of aerosol droplets is higher in fall than in spring, then the relative importance of the $\text{SO}_2 + \text{O}_3$ reaction resulting in higher $\Delta^{17}O$ values would increase in fall. However, ion concentration analyses of aerosols collected at DDU exhibited higher alkalinity in spring than in fall (Jourdain and Legrand, 2002; Legrand et al., 2016b), which is inconsistent with this

explanation. Hence, this process can be excluded from further consideration.

It should be noted that smaller difference was observed between spring and fall $\Delta^{17}O(\text{NO}_3^-)$ values, which would be also affected by the above two processes. Snow $\text{NO}_X$ emission would decrease $\Delta^{17}O(\text{NO}_3^-)$ through depression of $\text{O}_3/\text{RO}_X$ ratios, while halogen chemistry would lead to high $\Delta^{17}O(\text{NO}_3^-)$ values through an oxygen atom transfer from $\text{O}_3$ to BrO and consequently to nitrate (Morin et al., 2007; Savarino et al., 2013). Although it is difficult to identify the precise processes

involved, observations of $\Delta^{17}O(\text{nss-SO}_4^{2-})$ and $\Delta^{17}O(\text{NO}_3^-)$ values at an inland site (e.g., Concordia Station) would enable us to determine which process causes the spring/fall difference in the oxidation chemistry in the DDU atmosphere. If snow $\text{NO}_X$ emission is the source of low $\Delta^{17}O(\text{nss-SO}_4^{2-})$ at DDU in spring, then $\Delta^{17}O(\text{nss-SO}_4^{2-})$ and $\Delta^{17}O(\text{NO}_3^-)$ at an inland site would also exhibit lower values in spring than in fall.

### 4.4 Air mass origin analysis

Using observations of several oxidants at DDU and Concordia Station (e.g., Legrand et al., 2009; 2016a; Kukui et al, 2012; Grilli et al., 2013; Kukui et al., 2014), it has been suggested that the oxidative capacity of the atmosphere at DDU is




influenced by air masses transported from the East Antarctic Plateau during katabatic wind outflows. The strong $NO_X$ emission from snowpack in inland Antarctic stimulates the ozone and $RO_X$ production through complex photochemistry, consequently constructing a strong oxidizing canopy. Thus, the atmosphere at DDU is highly oxidative, when air masses are from inland regions; while it is less, when air masses are from the ocean (Legrand et al., 2009). Therefore, we expected that

$\Delta^{17}O$ values for sulfate and nitrate also depend on air mass origin.

In Fig. 5, the $\Delta^{17}O$ values for sulfate and nitrate are given as a function of the time that air masses were over the continent during the five-days travel prior to arriving at DDU. Summer data show that the contact times of air masses with the continent varied between 20 and 120 h (i.e., continental air and oceanic air are well mixed). Similarly, their $\Delta^{17}O$ values show insignificant variation, having low values of around 1‰ and 25‰ for sulfate and nitrate, respectively. This trend

reflects two different phenomena, decreased $\Delta^{17}O$ values in summer because of the high contribution of photo-oxidants to atmospheric chemistry, and increased import of oceanic air. In contrast, plots for other seasons show that the $\Delta^{17}O$ values exhibit high variation, although most of the air masses originate from the continent. It is important to note that this non-correlation does not mean that there is no link between $\Delta^{17}O$ values and air mass origin. The influence of air mass transport on oxidative capacity has been demonstrated in daily observations of oxidants (e.g., Legrand et al., 2009). Given that air

masses at this site are from a variety of directions, and are mixed together, this makes interpretation of weekly averaged analyses more complicated. Hence, no significant correlation between $\Delta^{17}O$ values and air mass origin in weekly data could reflect a real lack of correlation or too broad a time resolution to these data.

## 5 Summary

To develop an understanding of the oxidizing capacity of the Antarctic atmosphere, seasonal variations of $\Delta^{17}O$ values of atmospheric sulfate, nitrate and ozone were analyzed using the aerosol samples collected at DDU throughout 2011. Both $\Delta^{17}O(nss\text{-}SO_4^{2-})$ and $\Delta^{17}O(NO_3^-)$ values exhibited clear seasonal variations, with summer minima and winter maxima. In contrast, $\Delta^{17}O$ values of ozone showed limited variability throughout the year, indicating that $\Delta^{17}O(O_3)$ values do not significantly influence summer/winter trends in $\Delta^{17}O(nss\text{-}SO_4^{2-})$ and $\Delta^{17}O(NO_3^-)$ values. We concluded that the main factor

controlling the $\Delta^{17}O(nss\text{-}SO_4^{2-})$ and $\Delta^{17}O(NO_3^-)$ values is local oxidation chemistry. Thus, the summer/winter trends of $\Delta^{17}O(nss\text{-}SO_4^{2-})$ and $\Delta^{17}O(NO_3^-)$ values reflect sunlight-driven changes in the relative importance of photochemical oxidants (i.e., $O_3/RO_X$ ratios) in oxidation, which is supported by co-variation of the $\Delta^{17}O(nss\text{-}SO_4^{2-})$ and $\Delta^{17}O(NO_3^-)$ values with ozone mixing ratios.

However, despite having a similar range of ozone mixing ratios in spring and fall, we found that the $\Delta^{17}O(nss\text{-}SO_4^{2-})$ values

in spring were lower than in fall, while there was no clear difference between $\Delta^{17}O(NO_3^-)$ values. Possible explanations for the spring/fall differences for sulfate include: (i) low $O_3/RO_X$ ratios in spring induced by reactive nitrogen emissions from





snowpack at inland sites being transported to coastal sites; and (ii) effects of $SO_2$ oxidation by hypohalous acids (HOCl, HOBr), enhanced in spring by interaction of sea salt particles with photo-oxidants. Further observations of $\Delta^{17}O(nss\text{-}SO_4^{2-})$ and $\Delta^{17}O(NO_3^-)$ in aerosols collected at Antarctic inland sites will help us to identify the processes causing such different sulfate and nitrate formation in spring and fall. Clearly, chemical transport models, connecting $\Delta^{17}O$ values with the

oxidation processes involved in sulfate and nitrate formation are necessary for quantitative interpretation of the spatio-temporal patterns of $\Delta^{17}O$ signatures.

**Author contributions**

S. Ishino, S. Hattori and J. Savarino designed the research. M. Legrand, B. Jourdain and S. Preunkert provided qualified
complementary data, organized the Antarctic field campaign, and collected samples. S. Ishino, S. Hattori., J. Savarino, A. Barbero and N. Caillon performed experiments. S. Ishino, S. Hattori, J. Savarino, K. Kuribayashi, and N. Yoshida analyzed data. S. Ishino, S. Hattori and J. Savarino prepared the manuscript, with contributions from all other co-authors.

**Data availability**

The data used for the figures and the interpretations are shown in Supplement materials.

**Acknowledgments**

Financial support and field supplies for winter and summer campaigns at DDU were provided by the Institut Paul Emile Victor from Program 414 and the French Environmental Observation Service CESOA (Etude du cycle atmosphérique du
Soufre en relation avec le climat aux moyennes et hautes latitudes Sud; http://www-lgge.ujf-grenoble.fr/CESOA/rubri-que.php3? id_rubrique=2) dedicated to the study of the sulfur cycle at middle and high southern latitudes, and supported by the National Centre for Scientific Research (CNRS) Institute for Earth Sciences and Astronomy (INSU). This work is also supported by a Grant in-Aid for Scientific Research (S) (23224013) from the Ministry of Education, Culture, Sports, Science and Technology (MEXT), Japan, a Grant in-Aid for Young Scientist (A) (16H05884) of MEXT, Japan, and by a research
fund from Asahi Group Foundation. S.H., S. I., and N. Y. are supported by a Japan–France Research Cooperative Program (SAKURA and CNRS) of MEXT, Japan. J. S. and N. C. thank the CNRS/INSU (PRC program 207394) and the PH-SAKURA program of the French Embassy in Japan (project 31897PM) for financial support for this collaboration. We are grateful to Basile de Fleurian who carried out winter sampling at DDU in 2011. The authors gratefully acknowledge Erwann





Vince of the Laboratoire d'étude des Transferts en Hydrologie & Environnement forpreparation of bacterial cultures used for nitrate and ozone isotope analyses. Meteo France are acknowledged for providing meteorological data. Lars Kaleschke and his team members provided sea ice data. We also acknowledge the Air Resources Laboratory (ARL) for the use of their HYSPLIT transport and dispersion model available on the READY Web site (http://www.arl.noaa.gov/ready.html).

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





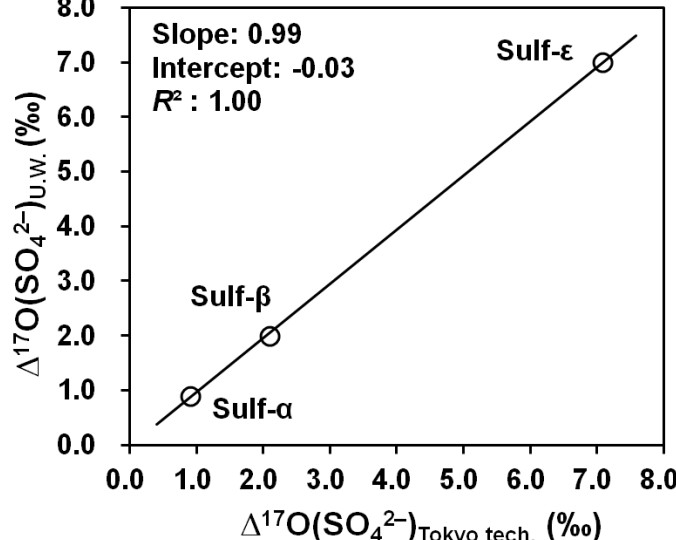

**Figure 1: Comparison of $\Delta^{17}O$ values of sulfate for standard materials measured at the University of Washington (U.W.) and Tokyo Institute of Technology (Tokyo tech.).**



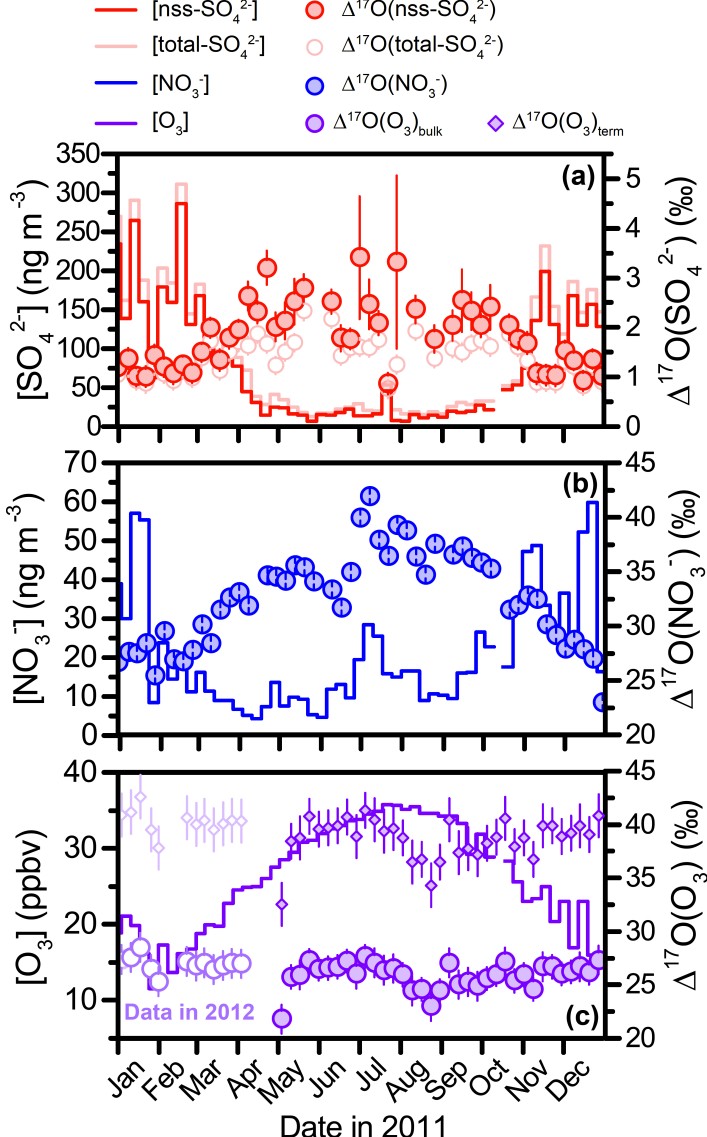

**Figure 2: Seasonal variations of concentrations (solid line) and Δ$^{17}$O values (circles) of sulfate (red), nitrate (blue), and ozone (purple) at Dumont d'Urville Station during 2011. Δ$^{17}$O values of ozone are shown as bulk (circle) and terminal (square) values. Δ$^{17}$O values of ozone include samples collected from January to April 2012.**



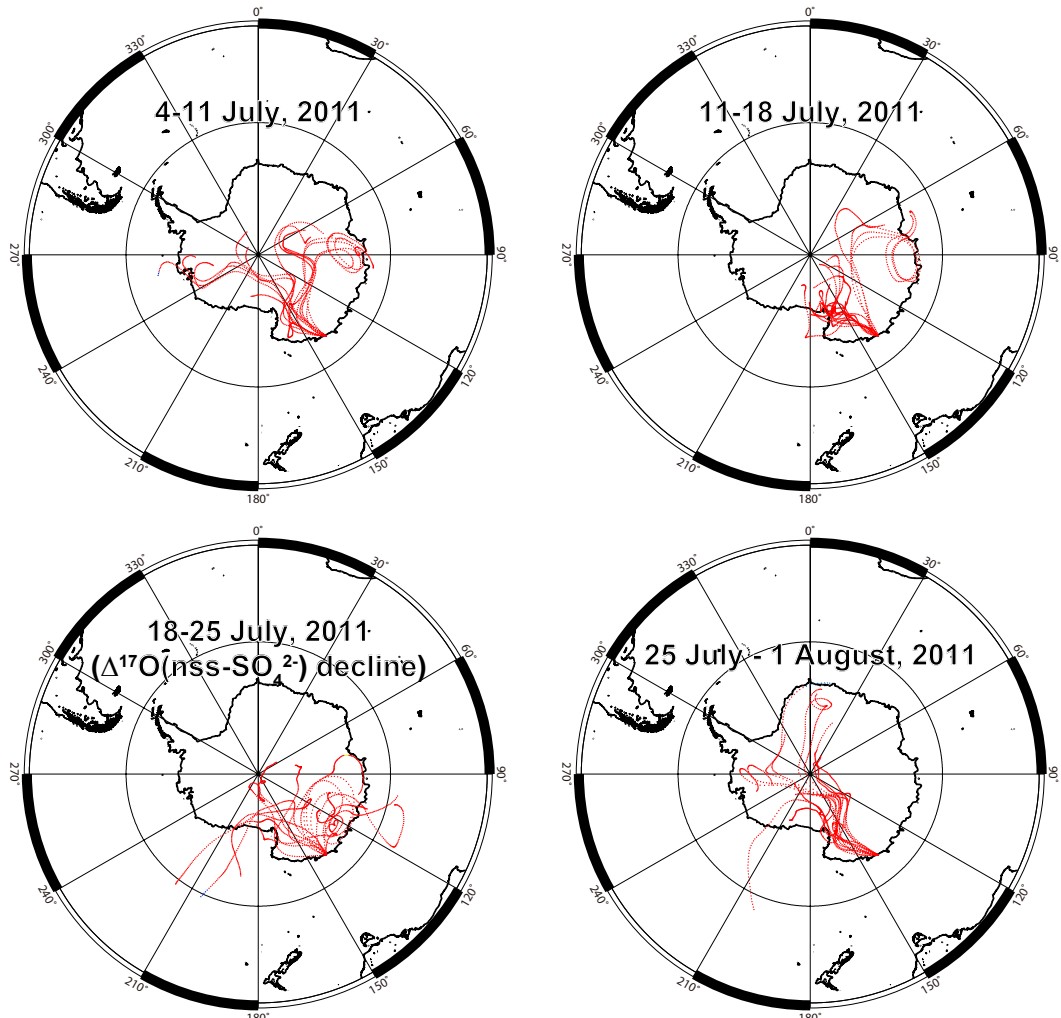

**Figure 3: Air mass pathways arriving at Dumont d'Urville Station during sampling periods in July 2011.**



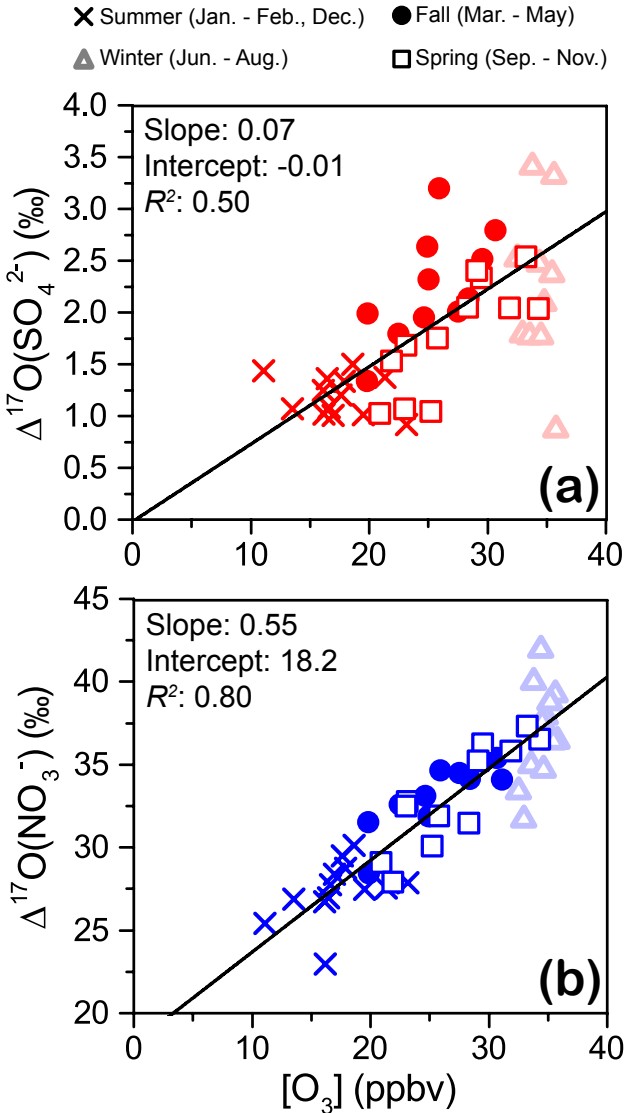

**Figure 4: Δ¹⁷O values of (a) sulfate, and (b) nitrate as a function of ozone mixing ratios [O₃]. Winter values were not taken into account in the calculation of slopes and intercepts.**



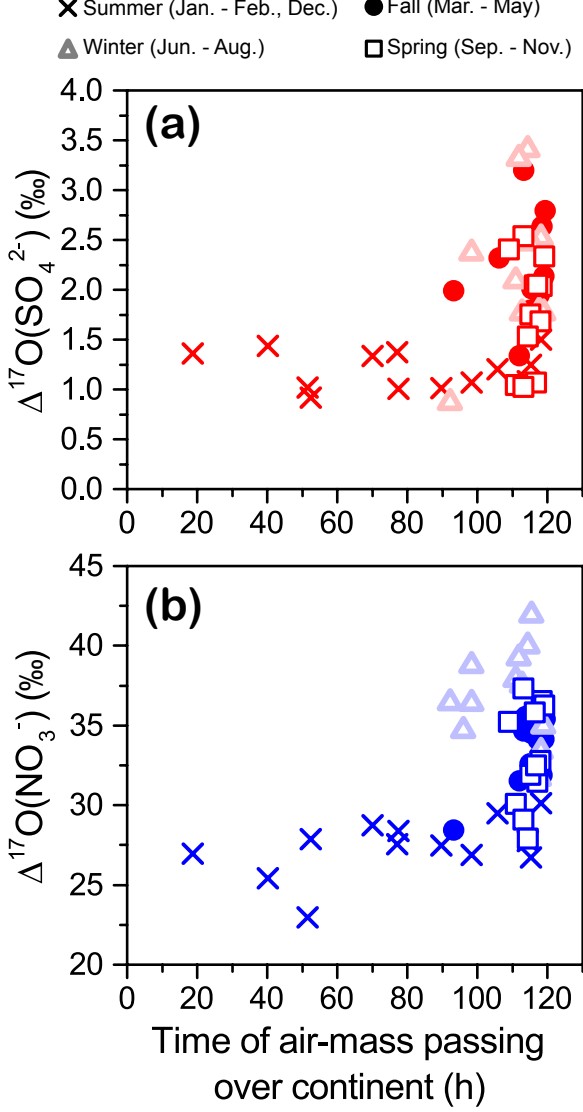

**Figure 5:** Relationship between $\Delta^{17}O$ values of (a) sulfate, and (b) nitrate, with the time taken for air masses to pass over the Antarctic continent (i.e., air mass contact time with surface snow).