# Peer review of "Seasonal variations of triple oxygen isotopic compositions of atmospheric sulfate, nitrate and ozone at Dumont d'Urville, coastal Antarctica"

_Atmospheric Chemistry and Physics, 2016_

## Referee Comment (RC1) · Anonymous Referee #1 · 14 Nov 2016

This is the first simultaneous measurements of concentration as well as triple oxygen isotope composition of atmospheric sulfate, nitrate, and ozone in an air-shed. I am impressed by the quality of the dataset, especially its capability in revealing the role of ozone/ROx ratio, HOX, and the sulfate or nitrate precursor chemistry as demonstrated by the authors.

A couple of general comments:

1. It seems to me that the variation seen in the $\Delta^{17}O$ of nitrate and sulfate are due more to changes in oxidation pathways and less to the oxidative capacity of the atmo-

sphere. The use of "oxidative capacity" to me is less accurate or at least poorly defined. I think the current atmosphere has plenty of "oxidative capacity" and is unlikely running below some kind of oxidative threshold. It's the oxidation pathway, being different for different species, that changes spatially and temporarily. And that "pathway" is what this study is going after.

2. A positive correlation between $\Delta 17O$ and $\delta 18O$ for nss sulfate is expected. Thus, the $\delta 18O$-nss SO4 would be a line of independent evidence for the conclusions. However, the $\delta 18O$ data is never mentioned, which needs some explanation.

3. Some apparent observational discrepancies are presented in Introduction but a clearer working hypothesis would improve the presentation. In other words, a recommended approach is to predict a potential seasonal pattern based on previous observational data and atmospheric chemistry models, and then go on to say that there are a couple of key parameters that we have not yet monitored in coastal Antarctica. In the end, parameters can only make sense when they are incorporated into a comprehensive atmospheric chemistry-transport model.

Specifics:

Abstract and the rest: I suggest when it's the first time mention "summer", add that it is the warm months or the austral summer.

Page 3 line 1: "This" is ambiguous.

Page 3 line 12-13: The final sentence can be deleted.

2.1.1.: Can you offer quantitative data instead of saying that ". . . the atmosphere is highly oxidative"?

Page 6 line 28: Cited reference "Bhattacharya et al 2008" is not found in the reference list.

---

## Referee Comment (RC2) · Anonymous Referee #2 · 16 Nov 2016

General Comments: My main concern after reading this manuscript is that it does not do more to quantify to what extent the triple oxygen isotopic compositions of sulfate and nitrate can be used as a measure of atmospheric oxidation capacity. Dumont d'Urville (DDU) should represent a well known case where the many contributing factors could be examined by applying statistics and modeling. I am convinced that the analytical method is sound, the samples come from a unique and potentially very important location, and a nice time series is delivered. It is not clear whether the goal is to establish the technique of using a combined oxygen triple isotope analysis in O3, NO3- and SO4– as an important proxy, or to use these measurements to tell us something new

and interesting about the earth system (in which case what?).

Specific Comments: The abstract starts with a big promise: 'Reconstruction of the oxidative capacity of the atmosphere is of great importance...Triple oxygen isotopic compositions..in the Antarctic ice cores have shown potential as stable proxies because they reflect the oxidation chemistry involved in their formation processes.'

A useful proxy must have a good correlation with the thing we can't measure directly. In this case the authors propose that the triple oxygen isotope anomalies in nitrate and sulfate are a useful proxy for the oxidative capacity of the atmosphere. This is a great goal because potentially, sulfate and nitrate in ice cores (or from other places, sediments, fern..) could be used to deduce past oxidative capacity. My concern is that the authors have not defined what it is exactly they are trying to determine based on their oxygen isotope measurements, and, they have not demonstrated that there is a correlation between the measurements and whatever that is, and therefore, they cannot claim that the triple oxygen istope anomalies in NO3- and SO4– are useful proxies. First, the authors should define what they mean by oxidative capacity. The oxidative capacity is not an exactly defined property as it could mean oxidation by O2, O3, OH, HO2, RO2, H2O2, O(1D), O(3P), NO3, Cl, BrO and so on. Oxidative capacity is sometimes taken to mean OH, but oxidation is a general process, not a specific one. The use of D17O would seem to be a better measure of relative exposure to ozone than [OH], since OH in the troposphere does not carry the D17O signal. The authors note (R2), oxidation by OH, will not transfer any of the anomaly from ozone to sulfate, and reactions R3, R4 and R5 transfer variable amounts of the ozone anomaly to sulfate. Because of the many pathways of SO2 oxidation it is difficult or impossible to find the relative contributions of the four proposed formation mechanisms based on one observable. In any case, since R2-R4 are all oxidation reactions converting S(IV) to S(VI), they all qualify as components of the atmosphere's oxidation capacity. The authors should be more exact about what it is that they propose to do with the measurements. Second, the discussion contains a lot of speculation about what may

ACPD

or may not cause the patterns shown in Figure 2. No firm conclusions are ever made from this discussion, and clearly, if you cannot show what causes the signal that is measured, there is no hope to use that same signal, measured at a different location, to make conclusions about its origin. There is a bit of a 'chicken vs egg' element to the discussion in which the data are assumed to be important and then used to justify assertions in the abstract about the O3/ROx and hypohalous acid mechanisms, and I would like the authors to be more clear in the logical progression: first show that this is a useful proxy (i.e. correlated to some observable e.g. [O3] or [OH]), and then as a second step, if possible, use the proxy to make a prediction or conclusion about the atmosphere.

Equation (2) is used to determine non sea salt sulfate. The amount of sea salt sulfate is approximated by multiplying the sodium concentration by a factor 'k' which is the mass ratio of sulfate to sodium in sea water (0.25), and this is subtracted from total sulfate, leaving non sea salt sulfate. The ratio of the concentration of sulfate to sodium in sea water is well known, 0.25, but a value of '0.13 plus or minus 0.04' is used for samples collected from May to October to account for 'sea salt fractionation processes that affect the Antarctic region in winter when temperatures drop below -8oC'. First, please rewrite to clarify that this is a chemical and not an isotopic fractionation. Second, how was the error of plus or minus 0.04 propagated in the calculation? I do not see error bars in the corrected valued in Figure 2. Third, the paper by Jourdain and Legrand (2002) states that the summer sulfate to sodium ratio exceeds the seawater value due to biogenic sulfate, ornithogenic sulfate DDU is famous for having many penguins) and heterogeneous uptake of SO2. Why wasn't a similar correction applied to summer sulfate? Fourth, the winter chemical fractionation is believed to be due to the precipitation of mirabilite when seawater freezes, and is thus dependent on the location of sea ice relative to DDU. Have there been any changes in sea ice and sea surface temperatures around DDU over the last 15 years that would have influenced the fractionation? Finally, if the correction is an empirical value taking into account sea ice, biogenic sulfate, penguin activity, heterogeneous chemistry and sea surface temperature, is the

resulting value truly representative of just sea salt aerosol?

As discussed, the D17O(SO4–) anomaly results from a combination of four mechanisms. The D17O(NO3-) anomaly depends on D17O of NO2, and of the oxidation mechanism. The authors discuss that NO2 formed from NO + O3 will contain a terminal oxygen atom from ozone, and these carry the D17O anomaly, resulting in preferential transfer to the NO2. First, what is known about photolysis? It plays a role in the equilibrium between NO, NO2 and O3, but does it produce an isotope anomaly? Second, in Section 4.1.2 I would have appreciated an estimate of the D17O value in nitrate for each of the mechanisms discussed. Third, would the authors estimate how much of NO3- is produced via NO2 + OH + M –> HNO3 + M, and how much by the dark reaction NO2 + O3 –> NO3, NO3 + RH –> HNO3.

Consider adding reaction schemes or figures to describe the S(IV) –> S(VI) and N(IV) –> N(V) conversions and the propagation of ozone in these mechanisms.

Many mechanisms are discussed, but it would be useful to have a statistical link between the data shown in Figure 2 and other data, for example, the output of a chemical model or a transport model (back trajectories, sea surface/ice conditions, etc), or measurements of [O3] and so on. This would clearly show whether these measurements are a good proxy for the oxidation capacity. Many atmospheric measurements have been made at the DDU station, and it seems that it ought to be possible to look for statistical correlations between the data in Figure 2 and station measurements (ozone, sunlight, humidity, NOx, modeled radical concentrations, temperature, wind speed and direction) – this data would be the key to establishing what the D17O proxies means.

The authors conclude that the seasonal changes in D17O in sulfate and nitrate are not due to seasonal trends in D17O in ozone; presumably the trend is due rather to different relative contributions by ozone oxidation to the oxygen in the sulfate and nitrate.

It seems difficult to figure out where sulfate and nitrate come from (sea salt, many atmospheric chemistry mechanisms, transported by katabatic wind from the stratosphere

or entrained upper troposphere or wind from the sea), and without the knowledge of how the material formed, how is it possible to determine the amount of ozone or OH that was present along the trajectory? And, if all that additional knowledge was necessary to determine the oxidation capacity along the path, this would seem to severely limit the power of this proxy. Ideally the DDU measurements would be an easy ideal test case to establish the proxy, a well studied site where all of the contributing factors can be qualtified. But, if the oxygen isotope anomaly cannot even be understood here, what hope is there for these measurements at less well studied sites, and at times in the past when there is uncertainty about basic things like extent of sea ice, air flow patterns, atmospheric chemistry, etc.

Technical Comments: 2,3 it is not clear from the grammar if 'its' refers to 'The reconstruction of changes in the past oxidative capacity' or 'climate change' 2,4 assuming that past oxidatidative capacity is what is meant, then I don't understand the inclusion of HCFCs in the list as these are a modern anthropogenic trace gas whose lifetime is only determined by the modern oxidative capacity, no comparison to preindustrial atmospheric chemistry can be made. 2,18 please double check definition or ROx, which would seem to indicate organic odd oxygen species (organic oxy and peroxy radicals). Is it standard to include OH and H2O2? 8,2 check 'summer,.as *has often been reported previously' 14, 2-3 'complex photochemistry' 'strong oxidizing canopy' 'highly oxidative' in each case I am wondering what these modifiers mean. Complex, strong and highly relative to what? I think many locations could be found with a much higher oxidizing capacity than DDU, and also, with photochemistries more complex than at DDU. 19, 33 Seinfeld and Pandis published the second edition of their book Atmospheric Chemistry and Physics in 2006 and the third edition in 2016. They did not publish any book with this title in 2012? Please specify edition and year.

---

## Author Comment (AC2) · 21 Feb 2017

**Author Response to Referee #2**

We thank Referee #2 for the helpful comment. Please find our responses below.

**General Comments: My main concern after reading this manuscript is that it does not do more to quantify to what extent the triple oxygen isotopic compositions of sulfate and nitrate can be used as a measure of atmospheric oxidation capacity. Dumont d'Urville (DDU) should represent a well known case where the many contributing factors could be examined by applying statistics and modeling. I am convinced that the analytical method is sound, the samples come from a unique and potentially very important location, and a nice time series is delivered. It is not clear whether the goal is to establish the technique of using a combined oxygen triple isotope analysis in O3, NO3- and SO4– as an important proxy, or to use these measurements to tell us something new and interesting about the earth system (in which case what?).**

We thank Referee #2 for this general but fundamental comment and we think that in some way this agrees with the comment of Referee #1 concerning the definition of the oxidation capacity of the atmosphere (OCA in short). We are fully aware of the limitation of the $\Delta^{17}O$ tracer. By essence, $\Delta^{17}O$ is an integrator of the different oxidation pathways and thus give a broad view of the relative importance of the involved oxidants. By no means its value is a measure of the oxidation capacity, first because $\Delta^{17}O$ is not a measure of the total oxidant concentration (it is a measure of the relative importance of different oxidation pathway), and second as mentioned by Referee #1, the oxidation capacity itself is not very well defined. So we should abandon the idea that from $\Delta^{17}O$, we can extract the OCA. $\Delta^{17}O$ should be seen as a way of constraining the oxidation scheme of a model by different mean than concentration measurements alone. And there are few examples now in the literature where $\Delta^{17}O$ has proven to be meaningful with respect to concentration analysis (e.g. Morin et al. (2008) and Savarino et al. (2013) with the bromine nitrate formation, McCabe et al. (2006) with the metal catalyzed sulfate formation, Alexander et al. (2012) and Chen et al. (2016) with the sulfate formation by halogen oxidation, etc.). Therefore, combining the measurement of $\Delta^{17}O$ values of $SO_4^{2-}$, $NO_3^-$ and $O_3$ has three main objectives.

1- Producing a set of data that can be used to constrain chemistry/transport model scheme, and because production schemes of nitrate and sulfate are interconnected through oxidant, the two species provide more constrain that each taken separately.

2- Demonstrating that the seasonality of $\Delta^{17}O(SO_4^{2-})$ and $\Delta^{17}O(NO_3^-)$ is a direct consequence of the oxidation scheme and not the seasonality of $\Delta^{17}O(O_3)$ (This is the first time the seasonality of $\Delta^{17}O(O_3)$ is confronted with $\Delta^{17}O(SO_4^{2-})$ and $\Delta^{17}O(NO_3^-)$. It has always been assumed before).

3- Revealing any features that can help to decipher the oxidation mechanisms of the precursors at this specific location.

As demonstrated by Legrand et al. (2009, 2016) DDU is in fact a very difficult case to treat because the chemical state of its atmosphere depends strongly on the export of oxidants and precursors from the plateau during katabatic winds. Even halogen chemistry is very different than other coastal sites. Considered with a sampling resolution of a week, we don't think that modeling the DDU data is an easy task and would certainly require a specific study, if not a specific model. So in short and to respond directly to the referee's comment, the goal of our paper is to document the $\Delta^{17}O$ variability over space and time. In summary section, we have added that future study using atmospheric chemistry and transport model is required.

Taking into account the throughout comment by Referee #2 as well as the comments by Referee #1 concerning the unclearness of hypothesis in this study, we have re-written our introduction in a way that we think now better emphasize the interest of measuring these isotopic tracers. We hope our new introduction answer the referee comments and questions.

**Specific comments: The abstract starts with a big promise: 'Reconstruction of the oxidative capacity of the atmosphere is of great importance...Triple oxygen isotopic compositions.. in the Antarctic ice cores have shown potential as stable proxies because they reflect the oxidation chemistry involved in their formation processes.'**

**A useful proxy must have a good correlation with the thing we can't measure directly. In this case the authors propose that the triple oxygen isotope anomalies in nitrate and sulfate are a useful proxy for the oxidative capacity of the atmosphere. This is a great goal because potentially, sulfate and nitrate in ice cores (or from other places, sediments, fern..) could be used to deduce past oxidative capacity. My concern is that the authors have not defined what it is exactly they are trying to determine based on their oxygen isotope measurements, and, they have not demonstrated that there is a correlation between the measurements and whatever that is, and therefore, they cannot claim that the triple**

oxygen isotope anomalies in NO3- and SO4– are useful proxies.

First, the authors should define what they mean by oxidative capacity. The oxidative capacity is not an exactly defined property as it could mean oxidation by O2, O3, OH, HO2, RO2, H2O2, O(1D), O(3P), NO3, Cl, BrO and so on. Oxidative capacity is sometimes taken to mean OH, but oxidation is a general process, not a specific one. The use of D17O would seem to be a better measure of relative exposure to ozone than [OH], since OH in the troposphere does not carry the D17O signal. The authors note (R2), oxidation by OH, will not transfer any of the anomaly from ozone to sulfate, and reactions R3, R4 and R5 transfer variable amounts of the ozone anomaly to sulfate. Because of the many pathways of SO2 oxidation it is difficult or impossible to find the relative contributions of the four proposed formation mechanisms based on one observable. In any case, since R2-R4 are all oxidation reactions converting S(IV) to S(VI), they all qualify as components of the atmosphere's oxidation capacity. The authors should be more exact about what it is that they propose to do with the measurements.

Second, the discussion contains a lot of speculation about what may or may not cause the patterns shown in Figure 2. No firm conclusions are ever made from this discussion, and clearly, if you cannot show what causes the signal that is measured, there is no hope to use that same signal, measured at a different location, to make conclusions about its origin. There is a bit of a 'chicken vs egg' element to the discussion in which the data are assumed to be important and then used to justify assertions in the abstract about the O3/ROx and hypohalous acid mechanisms, and I would like the authors to be more clear in the logical progression: first show that this is a useful proxy (i.e. correlated to some observable e.g. [O3] or [OH]), and then as a second step, if possible, use the proxy to make a prediction or conclusion about the atmosphere.

After some thoughts and discussions with the co-authors, we fully agree with the referee's comment and decide to rephrase the motivation of our study. First DDU cannot be used as a place to establish any ice core proxy. The coastal chemistry and transport is very different than inland sites. To establish a proxy, more than a strong correlation, a mechanism invariant in time and space is needed, and clearly the correlation between $O_3$ and $\Delta^{17}O$ at DDU should not be seen as a proxy of ozone and used to interpret ice cores. We know that such correlation is only fortuitous and results mainly from phenomena others than a simple cause-consequence effect induced by a change in ozone concentration (e.g. transport, radiations, aerosol burden, acidity all influence $\Delta^{17}O$). However, DDU with strong seasonal contrasts and proximity with the ocean

source is an interesting place to study $\Delta^{17}O$ with the goal to understand the different mechanisms at play. For instance, we show that $\Delta^{17}O(O_3)$ does not change with season despite contrasted environmental conditions during the year. One direct consequence, actually applicable to ice cores is that any variation of $\Delta^{17}O$ of sulfate and nitrate in ice core is solely a consequence of the oxidation mechanisms and mixing. Rephrased, the objective of our study is more to confront the theory behind $\Delta^{17}O$ (i.e. the transfer of the $\Delta^{17}O$ of ozone to other compounds) than directly producing an oxidation proxy for ice core. It should be bare in mind that the $^{17}O$ transfer theory relies mainly on hypothesis not validated by any observation (see e.g. Morin et al. 2011) and continuous observation are thus necessary. However, $\Delta^{17}O$ of ozone, as well as testing the theory indirectly benefit to the ice core study. In order to dissipate this misunderstanding, our text has been modified to better focus on the general understanding of the $\Delta^{17}O$ as a tool to probe the chemistry of the atmosphere than producing an ice core proxy.

**Equation (2) is used to determine non sea salt sulfate. The amount of sea salt sulfate is approximated by multiplying the sodium concentration by a factor 'k' which is the mass ratio of sulfate to sodium in sea water (0.25), and this is subtracted from total sulfate, leaving non sea salt sulfate. The ratio of the concentration of sulfate to sodium in sea water is well known, 0.25, but a value of '0.13 plus or minus 0.04' is used for samples collected from May to October to account for 'sea salt fractionation processes that affect the Antarctic region in winter when temperatures drop below -8oC'. First, please rewrite to clarify that this is a chemical and not an isotopic fractionation.**

We modified the expression as suggested by Referee #2, to emphasize that sea salt fractionation is a fractionation in chemical component and different from an isotope fractionation (p.6, l.30). It is due to the formation of the mirabilite, a $Na_2SO_4$ crystal phase that precipitates at $-8°C$ and thus depleting the seawater in $SO_4^{2-}$ relative to $Na^+$.

**Second, how was the error of plus or minus 0.04 propagated in the calculation? I do not see error bars in the corrected valued in Figure 2.**

In Supplementary material, we added the error of $[\text{nss-}SO_4^{2-}]$ propagated from the uncertainty in concentration measurement by IC (i.e., 5%) and standard deviation (1σ) of filter blank values, and $k$ value ($0.13 \pm 0.04$ in winter). The error propagated to $\Delta^{17}O(\text{nss-}SO_4^{2-})$ values are shown in

Figure 2, by red vertical line with each circle.

Through this correction, the standard deviation filter blank was newly added into consideration for the uncertainty analysis. Due to this correction, the uncertainty for $\Delta^{17}O(nss\text{-}SO_4^{2-})$ values was also slightly changed, which is now applied to Figure 2. Also the explanation for the uncertainty analysis was modified (p.5, l.7-12 and p.6, l.31 - p.7, l.2).

**Third, the paper by Jourdain and Legrand (2002) states that the summer sulfate to sodium ratio exceeds the seawater value due to biogenic sulfate, ornithogenic sulfate (DDU is famous for having many penguins) and heterogeneous uptake of SO2. Why wasn't a similar correction applied to summer sulfate?**

In our study, $\Delta^{17}O(SO_4^{2-})$ measurement was performed only on the submicron (fine) particles, as sulfate in the supermicron (coarse) mode particles consists of more than 80% of ss-$SO_4^{2-}$, and will results in a large uncertainty in $\Delta^{17}O(nss\text{-}SO_4^{2-})$ if this fraction was used. As discussed by Jourdain and Legrand (2002), the ornithogenic soil input affects mainly this supermicron (coarse) size particles. Thus, the penguin's emissions are not considered to impact our data. This is specifically mentioned in section 2.2.2 in the previous manuscript. However to better emphasize this, we have modified the order of sentences and separated the paragraph dealing with this (p.7, l.3-5).

**Fourth, the winter chemical fractionation is believed to be due to the precipitation of mirabilite when seawater freezes, and is thus dependent on the location of sea ice relative to DDU. Have there been any changes in sea ice and sea surface temperatures around DDU over the last 15 years that would have influenced the fractionation?**

**Finally, if the correction is an empirical value taking into account sea ice, biogenic sulfate, penguin activity, heterogeneous chemistry and sea surface temperature, is the resulting value truly representative of just sea salt aerosol?**

The $k$ value of $0.13 \pm 0.04$ is derived from the examination of the $[SO_4^{2-}]/[Na^+]$ ratio of aerosol present in supermicron modes at DDU (Jourdain and Legrand, 2002) from May to October. Also for our dataset of the $[SO_4^{2-}]/[Na^+]$ ratio of submicron mode particle, we obtained k value of 0.13 by the same calculation with Jourdain and Legrand (2002). (Note that for this calculation, we removed data of 18/07/11 because of anomalously high sulfate loading as discussed in the manuscript, and data of 18/10/11 and 24/10/11 because they are included in the latter half of October.)

Thus indeed, this factor is completely empirical and probably average few processes. However, it was done by Jourdain and Legrand (2002) over two winters when the population of penguins has long ago vanished (there is still a Emperor colony but its number unit has no comparison with the 10 000 Adélie penguins present in summer), the station recovered its thick snow blanket and biogenic emissions completely ended. Furthermore, since a physical separation is necessary to generate depleted sulfate aerosols, the precipitation can only happen on the sea ice. In winter the temperature drop well below $-8°C$ each year, and the sea-ice area is rather similar from one to another year. We have therefore assumed that the value of 0.13 is typical and can be applied for any year.

**As discussed, the D17O(SO4–) anomaly results from a combination of four mechanisms. The D17O(NO3-) anomaly depends on D17O of NO2, and of the oxidation mechanism. The authors discuss that NO2 formed from NO + O3 will contain a terminal oxygen atom from ozone, and these carry the D17O anomaly, resulting in preferential transfer to the NO2. First, what is known about photolysis? It plays a role in the equilibrium between NO, NO2 and O3, but does it produce an isotope anomaly?**

A study of the $NO_2$ photo-dissociation dynamic (Jost et al, 2005) has not been able to demonstrate the MIF nature of this photo-dissociation. Thus, $NO_2$ photolysis is believed to be a reaction without a detectable oxygen isotope anomaly, and the NO products preserve $\Delta^{17}O$ values of $NO_2$ from a statistical point of view.

**Second, in Section 4.1.2 I would have appreciated an estimate of the D17O value in nitrate for each of the mechanisms discussed.**

We added Table 1 to show both $\Delta^{17}O(SO_4^{2-})$ and $\Delta^{17}O(NO_3^-)$ values for each reaction and added the reaction schemes of nitrate formation at section 4.1.2 (p.11, 1.4-25). The $\Delta^{17}O(NO_3^-)$ values shown in Table 1 were estimated by Morin et al. (2011), using photochemical box model with photochemical steady state approximation. Since $\Delta^{17}O(SO_4^{2-})$ values are also summarized in Table 1, we also modified section 4.1.1 (p.10, 1.11-28), $\Delta^{17}O$ values of sulfate, to avoid the repetition and the lack of information. Additionally, to make the explanation clearer, we summarized the principle of $\Delta^{17}O$ estimations in section 4.1 (p.10, 1.2-9).

**Third, would the authors estimate how much of NO3- is produced via NO2 + OH + M –> HNO3 + M, and how much by the dark reaction NO2 + O3 –> NO3, NO3 + RH –> HNO3.**

We added one paragraph in section 4.2 (p.13, l.7-26) to mention about the relative contribution of oxidation pathways, estimated from observed $\Delta^{17}O(SO_4^{2-})$ and $\Delta^{17}O(NO_3^-)$ values. For sulfate, the relative contribution of only $SO_3^{2-} + O_3$ oxidation can be calculated. For nitrate, the relative contribution of oxidation pathways can be estimated only for summer samples, because in winter, the main nitrate source is believed to result mainly from the deposition of polar stratospheric clouds (Santacesaria et al, 2001; Savarino et al, 2007), which is not representative of oxidation chemistry occurring in the boundary layer of DDU. Therefore, only for summer sample, the relative contribution of $NO_2 + OH$ pathway was estimated. For further constraints on the relative contribution of other oxidation pathways, a coupled stratosphere/troposphere CTM will be necessary, which is clearly beyond the scope of this paper.

**Consider adding reaction schemes or figures to describe the S(IV) –> S(VI) and N(IV) –> N(V) conversions and the propagation of ozone in these mechanisms.**

We added Table 1 to describe $\Delta^{17}O(SO_4^{2-})$ and $\Delta^{17}O(NO_3^-)$ values produced via each oxidation pathways.

**Many mechanisms are discussed, but it would be useful to have a statistical link between the data shown in Figure 2 and other data, for example, the output of a chemical model or a transport model (back trajectories, sea surface/ice conditions, etc), or measurements of [O3] and so on. This would clearly show whether these measurements are a good proxy for the oxidation capacity. Many atmospheric measurements have been made at the DDU station, and it seems that it ought to be possible to look for statistical correlations between the data in Figure 2 and station measurements (ozone, sunlight, humidity, NOx, modeled radical concentrations, temperature, wind speed and direction) – this data would be the key to establishing what the D17O proxies means.**

As shown in figure 4, we compared the $\Delta^{17}O(SO_4^{2-})$ and $\Delta^{17}O(NO_3^-)$ values to [O_3]. We also found correlations between $\Delta^{17}O$ values and the time of air-mass under sunlight, which was

calculated using back trajectory and daytime data, exhibiting the correlation coefficients of 0.51 and 0.65 for sulfate and nitrate, respectively. However, seasonal variation in [$O_3$] is known to be mainly controlled by its photochemical destruction, and indeed, sunlight data was correlated to [$O_3$] with $R^2$ value of 0.68. Therefore, we used the comparison with [$O_3$]. One sentence was added at the beginning of section 4.3 to describe the reason why ozone mixing ratio was used.

Although we compared the $\Delta^{17}O$ values with other parameters, such as temperature and humidity, we couldn't find better correlation than $\Delta^{17}O$ values vs. [$O_3$]. There is no annual observation neither model estimation of [OH] or [$NO_X$] (OPALE experiment, which included HOx, NOx and other reactive species quantifications was unfortunately conducted in 2010/2011, a year ahead of our time coverage), limiting us to compare our data to the other oxidant than $O_3$. As correctly pointed out by Referee #2, there is a need to use a chemistry/transport model to estimate $\Delta^{17}O$ values for further quantitative analyses. However, because of the unique chemical state depending on the transportation of oxidants and precursors from the plateau as well as surrounding ocean (Legrand et al., 2009, 2016), modeling of the DDU data is not an easy task and would certainly require a specific study. In this study, we provide the dataset to constrain the model, and demonstrate one of the assumption for estimates of $\Delta^{17}O$ values by modeling. We consider that this is one big step on the way for prognosticating the oxidation chemistry using $\Delta^{17}O$ values.

**The authors conclude that the seasonal changes in D17O in sulfate and nitrate are not due to seasonal trends in D17O in ozone; presumably the trend is due rather to different relative contributions by ozone oxidation to the oxygen in the sulfate and nitrate.**

**It seems difficult to figure out where sulfate and nitrate come from (sea salt, many atmospheric chemistry mechanisms, transported by katabatic wind from the stratosphere or entrained upper troposphere or wind from the sea), and without the knowledge of how the material formed, how is it possible to determine the amount of ozone or OH that was present along the trajectory? And, if all that additional knowledge was necessary to determine the oxidation capacity along the path, this would seem to severely limit the power of this proxy. Ideally the DDU measurements would be an easy ideal test case to establish the proxy, a well studied site where all of the contributing factors can be qualtified. But, if the oxygen isotope anomaly cannot even be understood here, what hope**

**is there for these measurements at less well studied sites, and at times in the past when there is uncertainty about basic things like extent of sea ice, air flow patterns, atmospheric chemistry, etc.**

As mentioned before we totally agree with the reviewer with his critic concerning the building of a new proxy from $\Delta^{17}O$ of sulfate and nitrate. It is still too early to for that. The main difficulty resides in the fact that the different oxidants ($O_3$, OH, XO) have very different chemical lifetimes among them and with respect to sulfate and nitrate, disconnecting the causal effect especially in winter. Moreover, the one-week integration of the aerosol collection does not help, neither. Therefore, measurement at one site is certainly not sufficient to establish any causal effect between $\Delta^{17}O$ and oxidant concentrations. More observations around the global are necessary. In the revised version, the idea of a proxy is now abandoned for the profit of more general concerns, which is the understanding of the $\Delta^{17}O$ build up. We think that a major result of our study is to show that the seasonal trend of $\Delta^{17}O$ sulfate and nitrate has nothing to do with the seasonal trend of $\Delta^{17}O$ ozone, which is a new and important result for the interpretation of the $\Delta^{17}O$ sulfate and nitrate. This result demonstrates that all the variability of $\Delta^{17}O$ of sulfate and nitrate is somehow embedded in the oxidation schemes of $NO_X$ and $SO_2$. As a last thought, it may in fact be impossible to establish an oxidant proxy from $\Delta^{17}O$ of nitrate and sulfate because fundamentally sulfate and nitrate are antagonist with respect of the oxidation capacity of the atmosphere, there are too short lifetime to be integrators of a large part of the atmosphere but at the same time too long lifetime to transcribe the local oxidative state of the atmosphere.

**Technical Comments:**

**2,3 it is not clear from the grammar if 'its' refers to 'The reconstruction of changes in the past oxidative capacity' or 'climate change'**
**2,4 assuming that past oxidative capacity is what is meant, then I don't understand the inclusion of HCFCs in the list as these are a modern anthropogenic trace gas whose lifetime is only determined by the modern oxidative capacity, no comparison to preindustrial atmospheric chemistry can be made.**
Since we modified our introduction, the indicated parts were removed.

**2,18 please double check definition or ROx, which would seem to indicate organic odd oxygen species (organic oxy and peroxy radicals). Is it standard to include OH and H2O2?**

As suggested by Referee #2, we changed the 'RO$_X$' into 'HO$_X$, RO$_X$, and H$_2$O$_2$' throughout the manuscript.

**8,2 check 'summer,.as \*has often been reported previously'**

We corrected to 'summer, as has often been reported previously' (p.8, l.28-29).

**14, 2-3 'complex photochemistry' 'strong oxidizing canopy' 'highly oxidative' in each case I am wondering what these modifiers mean. Complex, strong and highly relative to what? I think many locations could be found with a much higher oxidizing capacity than DDU, and also, with photochemistries more complex than at DDU.**

We modified the sentence as follows to avoid ambiguous expression (p.16, l.16-18).

・'complex photochemistry' → 'the enhanced HOx cycle'

・'strong oxidizing canopy' → Deleted.

・'highly oxidative' → 'compared to when air mass come from inland and ocean side. '

**19, 33 Seinfeld and Pandis published the second edition of their book Atmospheric Chemistry and Physics in 2006 and the third edition in 2016. They did not publish any book with this title in 2012? Please specify edition and year.**

We deeply thank to Referee#2 for careful reading. We checked the edition and published year, and corrected the reference list; Seinfeld, J. H., and Pandis, S. N.: Atmospheric Chemistry and Physics, 2nd edition, Wiley & Sons, New York, USA, 2006.

**References**

Alexander, B., Allman, D. J., Amos, H. M., Fairlie, T. D., Dachs, J., Hegg, D. A., and Sletten, R. S.: Isotopic constraints on the formation pathways of sulfate aerosol in the marine boundary layer of the

subtropical northeast Atlantic Ocean, J. Geophys. Res., 117, D06304, doi:10.1029/2011JD016773, 2012.

Chen, Q., Geng, L., Schmidt, J. A., Xie, Z., Kang, H., Dachs, J., Cole-Dai, J., Schauer, A. J., Camp, M. G., and Alexander, B.: Isotopic constraints on the role of hypohalous acids in sulfate aerosol formation in the remote marine boundary layer, Atmos. Chem. Phys., 16, 11433–11450, 2016.

Jost, R., Michalski, G., and Thiemens, M.: Comparison of rovibronic density of asymmetric versus symmetric $NO_2$ isotopologues at dissociation threshold: Broken symmetry effects, The Journal of Chemical Physics, 123, 054320, 10.1063/1.1978873, 2005.

Legrand, M., Preunkert, S., Jourdain, B., Gallée, H., Goutail, F., Weller, R., and Savarino, J.: Year-round record of surface ozone at coastal (Dumont d'Urville) and inland (Concordia) sites in East Antarctica, J. Geophys. Res., 114, D20306, 10.1029/2008jd011667, 2009.

Legrand, M., Preunkert, S., Savarino, J., Frey, M. M., Kukui, A., Helmig, D., Jourdain, B., Jones, A. E., Weller, R., Brough, N., and Gallée, H.: Inter-annual variability of surface ozone at coastal (Dumont d'Urville, 2004–2014) and inland (Concordia, 2007–2014) sites in East Antarctica, Atmos. Chem. Phys., 16, 8053-8069, 10.5194/acp-16-8053-2016, 2016.

McCabe, J. R., Savarino, J., Alexander, B., Gong, S., and Thiemens, M. H.: Isotopic constraints on non-photochemical sulfate production in the Arctic winter, Geophys. Res. Lett., 33, L05810, doi:10.1029/ 2005GL025164, 2006.

Morin, S., Savarino, J., Frey, M. M., Yan, N., Bekki, S., Bottenheim, J. W., and Martins, J. M. F.: Tracing the origin and fate of NOx in the Arctic atmosphere using stable isotopes in nitrate, Science, 322, 730–732, doi:10.1126/science.1161910, 2008.

Morin, S., Sander, R., and Savarino, J.: Simulation of the diurnal variations of the oxygen isotope anomaly ($\Delta^{17}O$) of reactive atmospheric species, Atmos. Chem. Phys., 11, 3653-3671, 10.5194/acp-11-3653-2011, 2011.

Santacesaria, V., MacKenzie, A. R., and Stefanutti, L.: A climatological study of polar stratospheric clouds (1989-1997) from LIDAR measurements over Dumont d'Urville (Antarctica), Tellus, 53B, 306-321, 2001.

Savarino, J., Kaiser, J., Morin, S., Sigman, D. M., and Thiemens, M. H.: Nitrogen and oxygen isotopic constraints on the origin of atmospheric nitrate in coastal Antarctica, Atmos. Chem. Phys., 7, 1925-1945, 10.5194/acp-7-1925-2007, 2007.

Savarino, J., Morin, S., Erbland, J., Grannec, F., Patey, M. D., Vicars, W., Alexander, B., and Achterberg, E. P.: Isotopic composition of atmospheric nitrate in a tropical marine boundary layer, Proc. Natl. Acad. Sci. USA, 110, 17668–17673, 2013.

---

## Author Response (AR1)

Author's comments in response to the anonymous referees for "Seasonal variations of triple oxygen isotopic compositions of atmospheric sulfate, nitrate and ozone at Dumont d'Urville, coastal Antarctica" by S. Ishino et al.

We thank the referees for their careful reading and helpful comments. Followings are the referee's comments in blue and our response in black, followed by the revised manuscript. We hope our response and the revised manuscript fully answer the referee's questions and suggestions.

Please see also Supplement materials since we modified our data, tables and figures as follows.

- ➤ We added  $\delta^{17}$ O and  $\delta^{18}$ O values of sulfate as supplement data in response to the question by Referee #1. Also, data for the uncertainty of [nss-SO42-] and  $\Delta^{17}$ O(nss-SO42-) were corrected after recalculation of error propagation through the response to Referee #2.
- > Table 1 was added as summary of oxidation reactions and corresponding  $\Delta^{17}$ O values of sulfate and nitrate, in response to suggestion by Referee #2.
- > Figure 2 was replaced into corrected version using the corrected uncertainty for  $\Delta^{17}O(nss-SO_4^{2-})$ .

In addition to the changes suggested by referees, we have made the following relevant change of the revised manuscript.

- Laboratoire de Glaciologie et Géophysique de l'Environnement (LGGE) changed into Institut des géosciences de l'environnement (IGE) from January 1st of this year. Therefore, we changed our affiliation (p.1, 1.8) and two applicable part (p.4, 1.26 and p.7, 1.8).
- The explanation of the back trajectory analysis using NOAA's HYSPLIT was corrected as we used Windows-based version, not the on-line version (p.8, 1.13-15).

**Author Response to Referee #1**

We thank Referee #1 for the helpful comment. Please find our responses below.

This is the first simultaneous measurements of concentration as well as triple oxygen isotope composition of atmospheric sulfate, nitrate, and ozone in an air-shed. I am impressed by the quality of the dataset, especially its capability in revealing the role of ozone/ROx ratio, HOX, and the sulfate or nitrate precursor chemistry as demonstrated by the authors.

We deeply appreciate for careful review by Referee#1 and understanding on the significance of this study.

**A couple of general comments:**

1. It seems to me that the variation seen in the  $\Delta 170$  of nitrate and sulfate are due more to changes in oxidation pathways and less to the oxidative capacity of the atmosphere. The use of "oxidative capacity" to me is less accurate or at least poorly defined. I think the current atmosphere has plenty of "oxidative capacity" and is unlikely running below some kind of oxidative threshold. It's the oxidation pathway, being different for different species, that changes spatially and temporarily. And that "pathway" is what this study is going after.

Thanks to the pointing out. We agree with Referee#1.  $\Delta^{17}$ O values of sulfate and nitrate reflect the relative contribution of various oxidation pathways involved in sulfate and nitrate formation rather than the proxies of the oxidative capacity. Therefore, we changed the word 'the oxidative capacity' to 'the oxidation pathways of SO2 and NOx' throughout the manuscript as suggested by Referee#1.

We also modified Introduction section, where explaining what do  $\Delta^{17}$ O values of sulfate and nitrate mean and how they will be connected to the reconstruction of the oxidative capacity. The relative contribution of the oxidation pathways depends on the relative abundance of each oxidant, and therefore offers the possibility to probe the past relative concentration of O3, OH and other oxidants, which are the main oxidative agents of the atmosphere. This was also mentioned by the comment of Referee#2 in terms of the interest of this study. Taking into consideration this comment and the comment of the same subject by Referee#2, we have re-written our introduction in a way that we think now better emphasize the interest of measuring these isotopic tracers. We hope our new introduction answer the referee comments and questions.

2. A positive correlation between  $\Delta^{17}O$  and  $\delta^{18}O$  for nss sulfate is expected. Thus, the  $\delta^{18}O$ -nss SO4 would be a line of independent evidence for the conclusions. However, the  $\delta^{18}O$  data is never mentioned, which needs some explanation.

As discussed by Schauer et al. (2012), the Ag2SO4 thermo-decomposition method is not reliable for the  $\delta^{18}$ O of sulfate due to the oxygen isotope exchange between the O2 products and the quartz reactor, while  $\Delta^{17}$ O of O2 can be corrected from this effect as the  $\Delta^{17}$ O values of quartz materials is assumed to be 0 ‰. But for  $\delta^{18}$ O values, such correction is not available due to the unknown of  $\delta^{18}$ O value of the quartz reactor and isotopic fractionation associated with this exchange. The explanation is now added in section 2.2.2 of the revised manuscript to explain the unreliable  $\delta^{18}$ O measurement. This is the main reason why we don't discuss the  $\delta^{18}$ O values of sulfate.

However, the  $\delta^{18}$ O data is provided in Supplementary materials on demand in case if someone wants to calculate the  $\Delta^{17}$ O value with another definition. Note that those  $\delta^{18}$ O and  $\delta^{17}$ O values are relative to reference O2, not calibrated relative to VSMOW. Based on this data, the plot of  $\Delta^{17}$ O as a function of  $\delta^{18}$ O does not show the positive correlation (slope: -0.03,  $R^2$ : 0.02), despite the clear seasonal variation in  $\delta^{18}$ O values with the summer minimum and winter maximum. This may result from the different controlling factors in  $\Delta^{17}$ O and  $\delta^{18}$ O. Whereas  $\Delta^{17}$ O signature depends on the oxidant which provides one of four oxygen atoms of sulfate,  $\delta^{18}$ O is expected to mainly depend on the oxygen atoms of water which provides three oxygen atoms of sulfate through the equilibrium between SO2 and water, as examined by the laboratory experiment by Holt et al. (1983). This is out of the main subject of our discussion about oxidation chemistry in the atmosphere.

Therefore, because of the unreliability of the data, as well as the inconsistency of the subject, we don't discuss the  $\delta^{18}$ O values of sulfate.

**3.** Some apparent observational discrepancies are presented in Introduction but a clearer working hypothesis would improve the presentation. In other words, a recommended

approach is to predict a potential seasonal pattern based on previous observational data and atmospheric chemistry models, and then go on to say that there are a couple of key parameters that we have not yet monitored in coastal Antarctica. In the end, parameters can only make sense when they are incorporated into a comprehensive atmospheric chemistry-transport model.

We thank to Referee #1 for the helpful suggestion. Based on previous observations,  $\Delta^{17}O(SO_4^{2-})$ and  $\Delta^{17}O(NO_3^{-})$  values are expected to show the seasonal variations with summer minima and winter maxima, because the oxidation pathways should shift from O3 oxidation in winter to HOx or H2O2 oxidations in summer along with sunlight driven changes in the relative abundance of those photochemical oxidants. However, due to the lack of  $\Delta^{17}O(O_3)$  values at coastal Antarctica, the possibility had been remaining that the seasonal variations of  $\Delta^{17}O(SO_4^{2-})$  and  $\Delta^{17}O(NO_3^{-})$  values at the site are influenced not only by the oxidation chemistry but also the variations in  $\Delta^{17}O(O_3)$  values. If the  $\Delta^{17}O(O_3)$  values are examined to have the flat value throughout the year, we can remove the possibility of the influence by  $\Delta^{17}O(O_3)$  values, and go on to the interpretation of  $\Delta^{17}O(SO_4^{2-})$  and  $\Delta^{17}O(NO_3^{-})$  values in terms of the chemistry. This is the main working hypothesis of this study.

We agree to Referee#1 that estimates by atmospheric chemistry-transport model are necessary when we want to check if the present understanding on sulfur and nitrogen chemistry can explain the observation. However, the observational data is indispensable to constrain the models. Therefore, the second purpose of this study is to provide the reliable observational data for the future model analyses. Nevertheless, we observed the discrepancy between our data and the present understanding on the chemistry for sulfate production.

Along with the response to the first comment, we have re-written the Introduction to present clear working hypotheses and the purpose of this study, as suggested by Referee#1. Few sentences are also added in the Summary to emphasize the necessity of model analyses in the future.

**Specifics:**

Abstract and the rest: I suggest when it's the first time mention "summer", add that it is the warm months or the austral summer.

We modified the expressions as follows.

-- p.1, 1.21: 'characterized by summer minima and winter maxima'  $\rightarrow$  'characterized by minima in the austral summer and maxima in winter'

-- p.3, 1.10: 'showing an austral summer minimum and a winter maximum'

-- p.8, 1.26-27: 'The [nss-SO42–] had a summer maximum of up to ~280 ng m-3 from January to February,'  $\rightarrow$  'The [nss-SO42–] had a maximum of up to ~280 ng m-3 from January to February, the months corresponding to the austral summer,'

Page 3 line 1: "This" is ambiguous.

Page 3 line 12-13: The final sentence can be deleted.

Since we changed a large part of our Introduction, those parts were all deleted.

**2.1.1.:** Can you offer quantitative data instead of saying that ": : : the atmosphere is highly oxidative"?**

We added the quantitative explanation to suggest how the atmosphere at DDU is oxidative at section 2.1.1., as suggested by Referee#1.

**Page 6 line 28: Cited reference "Bhattacharya et al 2008" is not found in the reference list.**

We thank to Referee#1 for careful reading. We added Bhattacharya et al. (2008) in the reference list (p.19, 1.5).

**References**

Schauer, A. J., Kunasek, S. A., Sofen, E. D., Erbland, J., Savarino, J., Johnson, B. W., Amos, H. M., Shaheen, R., Abaunza, M., Jackson, T. L., Thiemens, M. H., and Alexander, B.: Oxygen isotope exchange with quartz during pyrolysis of silver sulfate and silver nitrate, Rapid Commun. Mass. Sp., 26, 2151–2157, 2012.

Holt, B. D., Cunningham, P. T., Engelkemeir, A. G., Graczyk, D. G., and Kumar, R.: Oxygen-18 study of

nonaqueous-phase oxidation of sulfur dioxide, Atmos. Env., 17, 625-632, 1983.

**Author Response to Referee #2**

We thank Referee #2 for the helpful comment. Please find our responses below.

General Comments: My main concern after reading this manuscript is that it does not do more to quantify to what extent the triple oxygen isotopic compositions of sulfate and nitrate can be used as a measure of atmospheric oxidation capacity. Dumont d'Urville (DDU) should represent a well known case where the many contributing factors could be examined by applying statistics and modeling. I am convinced that the analytical method is sound, the samples come from a unique and potentially very important location, and a nice time series is delivered. It is not clear whether the goal is to establish the technique of using a combined oxygen triple isotope analysis in O3, NO3- and SO4– as an important proxy, or to use these measurements to tell us something new and interesting about the earth system (in which case what?).

We thank Referee #2 for this general but fundamental comment and we think that in some way this agrees with the comment of Referee #1 concerning the definition of the oxidation capacity of the atmosphere (OCA in short). We are fully aware of the limitation of the  $\Delta^{17}$ O tracer. By essence,  $\Delta^{17}$ O is an integrator of the different oxidation pathways and thus give a broad view of the relative importance of the involved oxidants. By no means its value is a measure of the oxidation capacity, first because  $\Delta^{17}$ O is not a measure of the total oxidant concentration (it is a measure of the relative importance of different oxidation pathway), and second as mentioned by Referee #1, the oxidation capacity itself is not very well defined. So we should abandon the idea that from  $\Delta^{17}$ O, we can extract the OCA.  $\Delta^{17}$ O should be seen as a way of constraining the oxidation scheme of a model by different mean than concentration measurements alone. And there are few examples now in the literature where  $\Delta^{17}$ O has proven to be meaningful with respect to concentration analysis (e.g. Morin et al. (2008) and Savarino et al. (2013) with the bromine nitrate formation, McCabe et al. (2006) with the metal catalyzed sulfate formation, Alexander et al. (2012) and Chen et al. (2016) with the sulfate formation by halogen oxidation, etc.). Therefore, combining the measurement of  $\Delta^{17}$ O values of SO42-, NO3- and O3 has three main objectives.

1- Producing a set of data that can be used to constrain chemistry/transport model scheme, and because production schemes of nitrate and sulfate are interconnected through oxidant, the two species provide more constrain that each taken separately.

- 2- Demonstrating that the seasonality of  $\Delta^{17}O(SO_4^{2-})$  and  $\Delta^{17}O(NO_3^{-})$  is a direct consequence of the oxidation scheme and not the seasonality of  $\Delta^{17}O(O_3)$  (This is the first time the seasonality of  $\Delta^{17}O(O_3)$  is confronted with  $\Delta^{17}O(SO_4^{2-})$  and  $\Delta^{17}O(NO_3^{-})$ . It has always been assumed before).
- 3- Revealing any features that can help to decipher the oxidation mechanisms of the precursors at this specific location.

As demonstrated by Legrand et al. (2009, 2016) DDU is in fact a very difficult case to treat because the chemical state of its atmosphere depends strongly on the export of oxidants and precursors from the plateau during katabatic winds. Even halogen chemistry is very different than other coastal sites. Considered with a sampling resolution of a week, we don't think that modeling the DDU data is an easy task and would certainly require a specific study, if not a specific model. So in short and to respond directly to the referee's comment, the goal of our paper is to document the  $\Delta^{17}$ O variability over space and time. In summary section, we have added that future study using atmospheric chemistry and transport model is required.

Taking into account the throughout comment by Referee #2 as well as the comments by Referee #1 concerning the unclearness of hypothesis in this study, we have re-written our introduction in a way that we think now better emphasize the interest of measuring these isotopic tracers. We hope our new introduction answer the referee comments and questions.

Specific comments: The abstract starts with a big promise: 'Reconstruction of the oxidative capacity of the atmosphere is of great importance...Triple oxygen isotopic compositions.. in the Antarctic ice cores have shown potential as stable proxies because they reflect the oxidation chemistry involved in their formation processes.'

A useful proxy must have a good correlation with the thing we can't measure directly. In this case the authors propose that the triple oxygen isotope anomalies in nitrate and sulfate are a useful proxy for the oxidative capacity of the atmosphere. This is a great goal because potentially, sulfate and nitrate in ice cores (or from other places, sediments, fern..) could be used to deduce past oxidative capacity. My concern is that the authors have not defined what it is exactly they are trying to determine based on their oxygen isotope measurements, and, they have not demonstrated that there is a correlation between the measurements and whatever that is, and therefore, they cannot claim that the triple oxygen isotope anomalies in NO3- and SO4- are useful proxies.

First, the authors should define what they mean by oxidative capacity. The oxidative capacity is not an exactly defined property as it could mean oxidation by O2, O3, OH, HO2, RO2, H2O2, O(1D), O(3P), NO3, Cl, BrO and so on. Oxidative capacity is sometimes taken to mean OH, but oxidation is a general process, not a specific one. The use of D17O would seem to be a better measure of relative exposure to ozone than [OH], since OH in the troposphere does not carry the D17O signal. The authors note (R2), oxidation by OH, will not transfer any of the anomaly from ozone to sulfate, and reactions R3, R4 and R5 transfer variable amounts of the ozone anomaly to sulfate. Because of the many pathways of SO2 oxidation it is difficult or impossible to find the relative contributions of the four proposed formation mechanisms based on one observable. In any case, since R2-R4 are all oxidation reactions converting S(IV) to S(VI), they all qualify as components of the atmosphere's oxidation capacity. The authors should be more exact about what it is that they propose to do with the measurements.

Second, the discussion contains a lot of speculation about what may or may not cause the patterns shown in Figure 2. No firm conclusions are ever made from this discussion, and clearly, if you cannot show what causes the signal that is measured, there is no hope to use that same signal, measured at a different location, to make conclusions about its origin. There is a bit of a 'chicken vs egg' element to the discussion in which the data are assumed to be important and then used to justify assertions in the abstract about the O3/ROx and hypohalous acid mechanisms, and I would like the authors to be more clear in the logical progression: first show that this is a useful proxy (i.e. correlated to some observable e.g. [O3] or [OH]), and then as a second step, if possible, use the proxy to make a prediction or conclusion about the atmosphere.

After some thoughts and discussions with the co-authors, we fully agree with the referee's comment and decide to rephrase the motivation of our study. First DDU cannot be used as a place to establish any ice core proxy. The coastal chemistry and transport is very different than inland sites. To establish a proxy, more than a strong correlation, a mechanism invariant in time and space is needed, and clearly the correlation between  $O_3$  and  $\Delta^{17}O$  at DDU should not be seen as a proxy of ozone and used to interpret ice cores. We know that such correlation is only fortuitous and results mainly from phenomena others than a simple cause-consequence effect induced by a change in ozone concentration (e.g. transport, radiations, aerosol burden, acidity all influence  $\Delta^{17}O$ ). However, DDU with strong seasonal contrasts and proximity with the ocean

source is an interesting place to study  $\Delta^{17}O$  with the goal to understand the different mechanisms at play. For instance, we show that  $\Delta^{17}O(O_3)$  does not change with season despite contrasted environmental conditions during the year. One direct consequence, actually applicable to ice cores is that any variation of  $\Delta^{17}O$  of sulfate and nitrate in ice core is solely a consequence of the oxidation mechanisms and mixing. Rephrased, the objective of our study is more to confront the theory behind  $\Delta^{17}O$  (i.e. the transfer of the  $\Delta^{17}O$  of ozone to other compounds) than directly producing an oxidation proxy for ice core. It should be bare in mind that the 17O transfer theory relies mainly on hypothesis not validated by any observation (see e.g. Morin et al. 2011) and continuous observation are thus necessary. However,  $\Delta^{17}O$  of ozone, as well as testing the theory indirectly benefit to the ice core study. In order to dissipate this misunderstanding, our text has been modified to better focus on the general understanding of the  $\Delta^{17}O$  as a tool to probe the chemistry of the atmosphere than producing an ice core proxy.

Equation (2) is used to determine non sea salt sulfate. The amount of sea salt sulfate is approximated by multiplying the sodium concentration by a factor 'k' which is the mass ratio of sulfate to sodium in sea water (0.25), and this is subtracted from total sulfate, leaving non sea salt sulfate. The ratio of the concentration of sulfate to sodium in sea water is well known, 0.25, but a value of '0.13 plus or minus 0.04' is used for samples collected from May to October to account for 'sea salt fractionation processes that affect the Antarctic region in winter when temperatures drop below -8oC'. First, please rewrite to clarify that this is a chemical and not an isotopic fractionation.

We modified the expression as suggested by Referee #2, to emphasize that sea salt fractionation is a fractionation in chemical component and different from an isotope fractionation (p.6, 1.30). It is due to the formation of the mirabilite, a Na2SO4 crystal phase that precipitates at  $-8^{\circ}$ C and thus depleting the seawater in SO42- relative to Na+.

**Second, how was the error of plus or minus 0.04 propagated in the calculation? I do not see error bars in the corrected valued in Figure 2.**

In Supplementary material, we added the error of  $[nss-SO_4^{2^-}]$  propagated from the uncertainty in concentration measurement by IC (i.e., 5%) and standard deviation (1 $\sigma$ ) of filter blank values, and *k* value (0.13±0.04 in winter). The error propagated to  $\Delta^{17}O(nss-SO_4^{2^-})$  values are shown in

Figure 2, by red vertical line with each circle.

Through this correction, the standard deviation filter blank was newly added into consideration for the uncertainty analysis. Due to this correction, the uncertainty for  $\Delta^{17}O(nss-SO_4^{2-})$  values was also slightly changed, which is now applied to Figure 2. Also the explanation for the uncertainty analysis was modified (p.5, 1.7-12 and p.6, 1.31 - p.7, 1.2).

Third, the paper by Jourdain and Legrand (2002) states that the summer sulfate to sodium ratio exceeds the seawater value due to biogenic sulfate, ornithogenic sulfate (DDU is famous for having many penguins) and heterogeneous uptake of SO2. Why wasn't a similar correction applied to summer sulfate?

In our study,  $\Delta^{17}O(SO_4^{2-})$  measurement was performed only on the submicron (fine) particles, as sulfate in the supermicron (coarse) mode particles consists of more than 80% of ss-SO42-, and will results in a large uncertainty in  $\Delta^{17}O(nss-SO_4^{2-})$  if this fraction was used. As discussed by Jourdain and Legrand (2002), the ornithogenic soil input affects mainly this supermicron (coarse) size particles. Thus, the penguin's emissions are not considered to impact our data. This is specifically mentioned in section 2.2.2 in the previous manuscript. However to better emphasize this, we have modified the order of sentences and separated the paragraph dealing with this (p.7, 1.3-5).

Fourth, the winter chemical fractionation is believed to be due to the precipitation of mirabilite when seawater freezes, and is thus dependent on the location of sea ice relative to DDU. Have there been any changes in sea ice and sea surface temperatures around DDU over the last 15 years that would have influenced the fractionation?

Finally, if the correction is an empirical value taking into account sea ice, biogenic sulfate, penguin activity, heterogeneous chemistry and sea surface temperature, is the resulting value truly representative of just sea salt aerosol?

The *k* value of  $0.13\pm0.04$  is derived from the examination of the  $[SO_4^{2^-}]/[Na^+]$  ratio of aerosol present in supermicron modes at DDU (Jourdain and Legrand, 2002) from May to October. Also for our dataset of the  $[SO_4^{2^-}]/[Na^+]$  ratio of submicron mode particle, we obtained k value of 0.13 by the same calculation with Jourdain and Legrand (2002). (Note that for this calculation, we removed data of 18/07/11 because of anomalously high sulfate loading as discussed in the manuscript, and data of 18/10/11 and 24/10/11 because they are included in the latter half of October.)

Thus indeed, this factor is completely empirical and probably average few processes. However, it was done by Jourdain and Legrand (2002) over two winters when the population of penguins has long ago vanished (there is still a Emperor colony but its number unit has no comparison with the 10 000 Adélie penguins present in summer), the station recovered its thick snow blanket and biogenic emissions completely ended. Furthermore, since a physical separation is necessary to generate depleted sulfate aerosols, the precipitation can only happen on the sea ice. In winter the temperature drop well below  $-8^{\circ}$ C each year, and the sea-ice area is rather similar from one to another year. We have therefore assumed that the value of 0.13 is typical and can be applied for any year.

As discussed, the D17O(SO4–) anomaly results from a combination of four mechanisms. The D17O(NO3-) anomaly depends on D17O of NO2, and of the oxidation mechanism. The authors discuss that NO2 formed from NO + O3 will contain a terminal oxygen atom from ozone, and these carry the D17O anomaly, resulting in preferential transfer to the NO2. First, what is known about photolysis? It plays a role in the equilibrium between NO, NO2 and O3, but does it produce an isotope anomaly?

A study of the NO2 photo-dissociation dynamic (Jost et al, 2005) has not been able to demonstrate the MIF nature of this photo-dissociation. Thus, NO2 photolysis is believed to be a reaction without a detectable oxygen isotope anomaly, and the NO products preserve  $\Delta^{17}$ O values of NO2 from a statistical point of view.

Second, in Section 4.1.2 I would have appreciated an estimate of the D17O value in nitrate for each of the mechanisms discussed.

We added Table 1 to show both  $\Delta^{17}O(SO_4^{2-})$  and  $\Delta^{17}O(NO_3^{-})$  values for each reaction and added the reaction schemes of nitrate formation at section 4.1.2 (p.11, 1.4-25). The  $\Delta^{17}O(NO_3^{-})$  values shown in Table 1 were estimated by Morin et al. (2011), using photochemical box model with photochemical steady state approximation. Since  $\Delta^{17}O(SO_4^{2-})$  values are also summarized in Table 1, we also modified section 4.1.1 (p.10, 1.11-28),  $\Delta^{17}O$  values of sulfate, to avoid the repetition and the lack of information. Additionally, to make the explanation clearer, we summarized the principle of  $\Delta^{17}O$  estimations in section 4.1 (p.10, 1.2-9). Third, would the authors estimate how much of NO3- is produced via NO2 + OH + M  $\rightarrow$ HNO3 + M, and how much by the dark reaction NO2 + O3  $\rightarrow$  NO3, NO3 + RH  $\rightarrow$  HNO3. We added one paragraph in section 4.2 (p.13, 1.7-26) to mention about the relative contribution of oxidation pathways, estimated from observed  $\Delta^{17}O(SO_4^{2-})$  and  $\Delta^{17}O(NO_3^{-})$  values. For sulfate, the relative contribution of only SO32- + O3 oxidation can be calculated. For nitrate, the relative contribution of oxidation pathways can be estimated only for summer samples, because in winter, the main nitrate source is believed to result mainly from the deposition of polar stratospheric clouds (Santacesaria et al, 2001; Savarino et al, 2007), which is not representative of oxidation chemistry occurring in the boundary layer of DDU. Therefore, only for summer sample, the relative contribution of NO2 + OH pathway was estimated. For further constraints on the relative contribution of other oxidation pathways, a coupled stratosphere/troposphere CTM will be necessary, which is clearly beyond the scope of this paper.

Consider adding reaction schemes or figures to describe the S(IV) -> S(VI) and N(IV) -> N(V) conversions and the propagation of ozone in these mechanisms.

We added Table 1 to describe  $\Delta^{17}O(SO_4^{2-})$  and  $\Delta^{17}O(NO_3^{-})$  values produced via each oxidation pathways.

Many mechanisms are discussed, but it would be useful to have a statistical link between the data shown in Figure 2 and other data, for example, the output of a chemical model or a transport model (back trajectories, sea surface/ice conditions, etc), or measurements of [O3] and so on. This would clearly show whether these measurements are a good proxy for the oxidation capacity. Many atmospheric measurements have been made at the DDU station, and it seems that it ought to be possible to look for statistical correlations between the data in Figure 2 and station measurements (ozone, sunlight, humidity, NOx, modeled radical concentrations, temperature, wind speed and direction) – this data would be the key to establishing what the D17O proxies means.

As shown in figure 4, we compared the  $\Delta^{17}O(SO_4^{2-})$  and  $\Delta^{17}O(NO_3^{-})$  values to  $[O_3]$ . We also found correlations between  $\Delta^{17}O$  values and the time of air-mass under sunlight, which was

calculated using back trajectory and daytime data, exhibiting the correlation coefficients of 0.51 and 0.65 for sulfate and nitrate, respectively. However, seasonal variation in  $[O_3]$  is known to be mainly controlled by its photochemical destruction, and indeed, sunlight data was correlated to  $[O_3]$  with  $R^2$  value of 0.68. Therefore, we used the comparison with  $[O_3]$ . One sentence was added at the beginning of section 4.3 to describe the reason why ozone mixing ratio was used. Although we compared the  $\Delta^{17}$ O values with other parameters, such as temperature and humidity, we couldn't find better correlation than  $\Delta^{17}$ O values vs. [O3]. There is no annual observation neither model estimation of [OH] or [NOx] (OPALE experiment, which included HOx, NOx and other reactive species quantifications was unfortunately conducted in 2010/2011, a year ahead of our time coverage), limiting us to compare our data to the other oxidant than  $O_3$ . As correctly pointed out by Referee #2, there is a need to use a chemistry/transport model to estimate  $\Delta^{17}$ O values for further quantitative analyses. However, because of the unique chemical state depending on the transportation of oxidants and precursors from the plateau as well as surrounding ocean (Legrand et al., 2009, 2016), modeling of the DDU data is not an easy task and would certainly require a specific study. In this study, we provide the dataset to constrain the model, and demonstrate one of the assumption for estimates of  $\Delta^{17}$ O values by modeling. We consider that this is one big step on the way for prognosticating the oxidation chemistry using  $\Delta^{17}$ O values.

The authors conclude that the seasonal changes in D17O in sulfate and nitrate are not due to seasonal trends in D17O in ozone; presumably the trend is due rather to different relative contributions by ozone oxidation to the oxygen in the sulfate and nitrate.

It seems difficult to figure out where sulfate and nitrate come from (sea salt, many atmospheric chemistry mechanisms, transported by katabatic wind from the stratosphere or entrained upper troposphere or wind from the sea), and without the knowledge of how the material formed, how is it possible to determine the amount of ozone or OH that was present along the trajectory? And, if all that additional knowledge was necessary to determine the oxidation capacity along the path, this would seem to severely limit the power of this proxy. Ideally the DDU measurements would be an easy ideal test case to establish the proxy, a well studied site where all of the contributing factors can be qualtified. But, if the oxygen isotope anomaly cannot even be understood here, what hope is there for these measurements at less well studied sites, and at times in the past when there is uncertainty about basic things like extent of sea ice, air flow patterns, atmospheric chemistry, etc.

As mentioned before we totally agree with the reviewer with his critic concerning the building of a new proxy from  $\Delta^{17}O$  of sulfate and nitrate. It is still too early to for that. The main difficulty resides in the fact that the different oxidants  $(O_3, OH, XO)$  have very different chemical lifetimes among them and with respect to sulfate and nitrate, disconnecting the causal effect especially in winter. Moreover, the one-week integration of the aerosol collection does not help, neither. Therefore, measurement at one site is certainly not sufficient to establish any causal effect between  $\Delta^{17}$ O and oxidant concentrations. More observations around the global are necessary. In the revised version, the idea of a proxy is now abandoned for the profit of more general concerns, which is the understanding of the  $\Delta^{17}$ O build up. We think that a major result of our study is to show that the seasonal trend of  $\Delta^{17}$ O sulfate and nitrate has nothing to do with the seasonal trend of  $\Delta^{17}$ O ozone, which is a new and important result for the interpretation of the  $\Delta^{17}$ O sulfate and nitrate. This result demonstrates that all the variability of  $\Delta^{17}$ O of sulfate and nitrate is somehow embedded in the oxidation schemes of NOX and SO2. As a last thought, it may in fact be impossible to establish an oxidant proxy from  $\Delta^{17}O$  of nitrate and sulfate because fundamentally sulfate and nitrate are antagonist with respect of the oxidation capacity of the atmosphere, there are too short lifetime to be integrators of a large part of the atmosphere but at the same time too long lifetime to transcribe the local oxidative state of the atmosphere.

**Technical Comments:**

2,3 it is not clear from the grammar if 'its' refers to 'The reconstruction of changes in the past oxidative capacity' or 'climate change'

2,4 assuming that past oxidative capacity is what is meant, then I don't understand the inclusion of HCFCs in the list as these are a modern anthropogenic trace gas whose lifetime is only determined by the modern oxidative capacity, no comparison to preindustrial atmospheric chemistry can be made.

Since we modified our introduction, the indicated parts were removed.

2,18 please double check definition or ROx, which would seem to indicate organic odd oxygen species (organic oxy and peroxy radicals). Is it standard to include OH and H2O2? As suggested by Referee #2, we changed the ' $RO_x$ ' into ' $HO_x$ ,  $RO_x$ , and  $H_2O_2$ ' throughout the manuscript.

**8,2 check 'summer, as \*has often been reported previously'**

We corrected to 'summer, as has often been reported previously' (p.8, 1.28-29).

14, 2-3 'complex photochemistry' 'strong oxidizing canopy' 'highly oxidative' in each case I am wondering what these modifiers mean. Complex, strong and highly relative to what? I think many locations could be found with a much higher oxidizing capacity than DDU, and also, with photochemistries more complex than at DDU.

We modified the sentence as follows to avoid ambiguous expression (p.16, 1.16-18).

- 'complex photochemistry'  $\rightarrow$  'the enhanced HOx cycle'
- 'strong oxidizing canopy'  $\rightarrow$  Deleted.
- 'highly oxidative'  $\rightarrow$  'compared to when air mass come from inland and ocean side.'

19, 33 Seinfeld and Pandis published the second edition of their book Atmospheric Chemistry and Physics in 2006 and the third edition in 2016. They did not publish any book with this title in 2012? Please specify edition and year.

We deeply thank to Referee#2 for careful reading. We checked the edition and published year, and corrected the reference list; Seinfeld, J. H., and Pandis, S. N.: Atmospheric Chemistry and Physics, 2nd edition, Wiley & Sons, New York, USA, 2006.

**Seasonal variations of triple oxygen isotopic compositions of atmospheric sulfate, nitrate and ozone at Dumont d'Urville, coastal Antarctica**

.

| 5  | Sakiko Ishino 1 , Shohei Hattori 1 , Joel Savarino 2 , Bruno Jourdain 2 , Susanne Preunkert 2 , Michel Legrand 2 , Nicolas Caillon 2 , Albane Barbero 2 , Kota Kuribayashi 3 , Naohiro Yoshida 1,4 ,                                                                                                                                                                                                                                                            | /                    | Sakiko
|----|-------------------------------------------------------------------------------------------------------------------------------------------------------------------------------------------------------------------------------------------------------------------------------------------------------------------------------------------------------------------------------------------------------------------------------------------------------------------------------------------------------------------------------------------------------------------------------|----------------------|-------------------------------|
| 10 | 1 Department of Chemical Science and Engineering, School of Materials and Chemical Technology, Tokyo Institute of
Technology, 4259 Nagatsuta-cho, Midori-ku, Yokohama 226-8502, Japan
2 Université Grenoble Alpes, CNRS, IRD, IGE, F-38000 Grenoble, France
3 Department of Environmental Chemistry and Engineering, Tokyo Institute of Technology, 4259 Nagatsuta-cho, Midori-ku,
Yokohama 226-8502, Japan
4 Earth-Life Science Institute, Tokyo Institute of Technology, Meguro-ku, Tokyo 152-8551, Japan |                      | Sakiko
PEnviron  |
|    | Correspondence to: Sakiko Ishino (ishino.s.ab@m.titech.ac.jp)                                                                                                                                                                                                                                                                                                                                                                                                                                                                                                                 | $\backslash \rangle$ | Sakiko                        |
|    | Abstract                                                                                                                                                                                                                                                                                                                                                                                                                                                                                                                                                                      |                      | 削除: 4
Sakiko    |
| Í  | Triple oxygen isotopic compositions ( $\Delta^{17}O = \delta^{17}O - 0.52 \times \delta^{18}O$ ) of atmospheric sulfate (SO 4 2- ) and nitrate (NO 3 - ) in the                                                                                                                                                                                                                                                                                                                                                                   |                      | 削除:5                          |
| 15 | atmosphere reflect the relative contribution of oxidation pathways involved in their formation processes, which potentially                                                                                                                                                                                                                                                                                                                                                                                                                                                   |                      | Sakiko
|    | provides information to reveal missing reactions in atmospheric chemistry models. However, there remain many theoretical                                                                                                                                                                                                                                                                                                                                                                                                                                                      |                      | the atmos                     |
|    | assumptions for the controlling factors of $\Delta^{17}O(SO_4^{2^-})$ and $\Delta^{17}O(NO_3^-)$ values in those model estimation. To test one of                                                                                                                                                                                                                                                                                                                                                                                                                             |                      | role in de                    |
|    | those assumption that $\Delta^{17}O$ values of ozone have a flat value and does not influence the seasonality of $\Delta^{17}O(SO_4^{2^-})$ and                                                                                                                                                                                                                                                                                                                                                                                                                               |                      |                               |
|    | $\Delta^{17}O(NO_3)$ values, we performed the first simultaneous measurement of $\Delta^{17}O$ values of atmospheric sulfate, nitrate, and                                                                                                                                                                                                                                                                                                                                                                                                                                    | ///                  |                               |
| 20 | ozone collected at Dumont d'Urville station (66°40'S, 140°01'E) throughout 2011. $\Delta^{17}$ O values of sulfate and nitrate                                                                                                                                                                                                                                                                                                                                                                                                                                                |                      |                               |
| I  | exhibited seasonal variation characterized by minima in the austral summer and maxima in winter, within the ranges of 0.9-                                                                                                                                                                                                                                                                                                                                                                                                                                                    |                      |                               |
| I  | 3.4 ‰ and 23.0–41.9 ‰, respectively. In contrast, $\Delta^{17}O$ values of ozone showed no significant seasonal variation, with                                                                                                                                                                                                                                                                                                                                                                                                                                               |                      |                               |
|    | values of 26 ± 1 ‰ throughout the year. These contrasting seasonal trends suggest that seasonality in $\Delta^{17}O(SO_4^{2^-})$ and                                                                                                                                                                                                                                                                                                                                                                                                                                          |                      |                               |
|    | $\Delta^{17}O(NO_3)$ values are not the result of changes in $\Delta^{17}O(O_3)$ but the changes in oxidation chemistry. The summer/winter                                                                                                                                                                                                                                                                                                                                                                                                                                    |                      |                               |
| 25 | trends for $\Delta^{17}O(SO_4^{2-})$ and $\Delta^{17}O(NO_3^{-})$ values are caused by sunlight-driven changes in the relative contribution of $O_{3_{el}}$                                                                                                                                                                                                                                                                                                                                                                                                                   |                      |                               |
|    | oxidation to the oxidation by HO X , RO X , and H 2 O 2 In addition to that general trend, by comparing $\Delta^{17}O(SO_4^{2^-})$ and                                                                                                                                                                                                                                                                                                                                                                                            |                      |                               |
|    | $\Delta^{17}O(NO_3)$ values to ozone mixing ratios, we found $\Delta^{17}O(SO_4^2)$ values observed in spring (September to November) were                                                                                                                                                                                                                                                                                                                                                                                                                                    |                      |                               |
|    | lower than in fall (March to May), while there is no significant spring/fall difference in $\Delta^{17}O(NO_3)$ values. The relatively                                                                                                                                                                                                                                                                                                                                                                                                                                        |                      |                               |
|    | low er sensitivity of $\Delta^{17}O(SO_4^{2-})$ values to the ozone mixing ratio in spring compared to fall is possibly explained by (i)                                                                                                                                                                                                                                                                                                                                                                                                                        |                      |                               |
| 30 | $\frac{1}{2}$ increased contribution of SO 2 oxidations by OH and H 2 O 2 caused by NO X emission from snowpack and/or (ii) SO 2 oxidation                                                                                                                                                                                                                                                                                                                                                                             |                      |                               |
|    | by hypohalous acids $(HOX = HOCl + HOBr)$ in the aqueous phase.                                                                                                                                                                                                                                                                                                                                                                                                                                                                                                               |                      |                               |
|    |                                                                                                                                                                                                                                                                                                                                                                                                                                                                                                                                                                               |                      |                               |

1

Sakiko ISHINO 15/2/2017 14:56

Sakiko ISHINO 12/1/2017 17:45

**1** Introduction**

Triple oxygen isotopic compositions ( $\Delta^{17}O = \delta^{17}O - 0.52 \times \delta^{18}O$ ) of atmospheric sulfate and nitrate have shown potential to probe the relative importance of various oxidation pathways involved in their formations (e.g., Michalski et al., 2003; Lee and Thiemens, 2001). Atmospheric ozone (O3) possesses high  $\Delta^{17}O$  values of approximately 26 ‰ (Krankowsky et al., 1995;

- 5 Johnston and Thiemens, 1997; Vicars et al., 2012, 2014), contrary to most of the oxygen bearing compounds such as O2 and H2O, which have Δ17O values of approximately 0 ‰ (Barkan and Luz, 2003, 2005). Oxygen atoms of O3 are directly or indirectly transferred to sulfate and nitrate through various oxidation pathways of their precursors, SO2 and NOX (= NO + NO2), respectively. ThereforexO3 oxidation produces sulfate and nitrate possessing high Δ17O valuesx whereas oxidation by OH, RO2 and H2O2 produces sulfate and nitrate with low Δ17O values. In general, the Δ17O(SO42-) and Δ17O(NO3-) values
- 10 can be prognosticated using the given  $\Delta^{17}O(SO_4^{2-})$  and  $\Delta^{17}O(NO_3^{-})$  values for each reaction pathways and the estimates of relative contribution of each oxidation pathways by atmospheric chemistry/transport models (e.g., Alexander et al., 2009; Morin et al., 2011; Sofen et al., 2011). Considering the higher HOX levels in summer due to enhanced photochemical activity relative to winter,  $\Delta^{17}O(SO_4^{2-})$  and  $\Delta^{17}O(NO_3^{-})$  values are expected to show the specific seasonal trend with summer minima and winter maxima in the mid to high latitude regions. By comparing those expected values to observations, missing
- 15 processes for sulfate and nitrate formation in the models have been proposed. For example, McCabe et al. (2006) performed an year-round observation of  $\Delta^{17}O(SO_4^{2^-})$  values at Alert, Canada, showing lower  $\Delta^{17}O(SO_4^{2^-})$  values during winter compared to the calculated values using a model of Feitcher et al. (1996); they suggest a twofold overestimate of  $O_3$ oxidation in sulfate formation during winter in the Northern hemisphere, and pointed out 10-18% contribution of metal catalyzed  $O_2$  oxidation in aqueous phase. An observation of  $\Delta^{17}O(NO_3^-)$  values in Alert by Morin et al. (2008) revealed
- 20 significantly higher values during spring compared to the calculated values, which is expected to be a result of NO2 oxidation by BrO. Savarino et al. (2013) also had shown significantly higher  $\Delta^{17}O(NO_3^{-})$  values at the marine boundary layer compared to an estimate of chemical transport model by Alexander et al. (2009), while the mismatch was fixed by considering a contribution of BrO in nitrate formation. The  $\Delta^{17}O(SO_4^{-2})$  and  $\Delta^{17}O(NO_3^{-})$  values are thus used to examine the lack of oxidation schemes in the models. However, the observational data of  $\Delta^{17}O(SO_4^{-2})$  and  $\Delta^{17}O(NO_3^{-})$  to constrain these
- 25 models are still not enough to demonstrate theoretical assumptions of the controlling factors of  $\Delta^{17}$ O signatures in these model estimation (e.g., Morin et al., 2011).

One of the important parameters which have to be confirmed by observations is  $\Delta^{17}O$  value of ozone, because of the possibility that  $\Delta^{17}O(SO_4^{2^-})$  and  $\Delta^{17}O(NO_3^-)$  values are influenced not only by changes in oxidation chemistry but also variations in  $\Delta^{17}O(O_3)$  values. Indeed, various  $\Delta^{17}O(O_3)$  values ranging 25-35 ‰ are used for model calculations to

30 reproduce the observed variations in  $\Delta^{17}O(SO_4^{2^-})$  and  $\Delta^{17}O(NO_3^-)$  values (Alexander et al., 2009; Morin et al., 2011; Sofen et al., 2011). Those varied  $\Delta^{17}O(O_3)$  values were assumed based on the observations using the cryogenic collection method (Johnston and Thiemens, 1997; Krankowsky et al., 1995), which showed highly varied values of 6-54 ‰. However, the

**2**

**akiko ISHINO 12/1/2017 17:57**

- Sakiko ISHINO 3/12/2016 10:53

Sakiko ISHINO 4/12/2016 19:20

Sakiko ISHINO 20/1/2017 11:2

Sakiko ISHINO 7/12/2016 06:19

**NE:** Atmospheric sulfate and nitrate are produced from  $SO_2$  and  $NO_X$  (=  $NO + NO_2$ ), respectively, and are the ultimate products of the sulfur and nitrogen cycles as results of oxidation reactions by various oxidants such as OH radicals, O3, H2O2, RO2 and/or halogen oxides (Seinfeld and Pandis, 2012). In general,

Sakiko ISHINO 7/12/2016 07:21

recent observations of  $\Delta^{17}O(O_3)$  values using the nitrite-coated filter method, which was developed by Vicars et al. (2012), and applied at Grenoble, R/V Polarstern Campaign (Vicars et al., 2014) and Dome C (Savarino et al., 2016), had shown the insignificant spatial and temporal variability with values ranging 23-27 ‰. This result suggests that the variations observed in  $\Delta^{17}O(SO_4^{2-})$  and  $\Delta^{17}O(NO_3^{-})$  values are not the result of changes in  $\Delta^{17}O(O_3)$  values, but mainly explained by the changes

- 5 in oxidation pathways. To test this hypothesis, it is necessary to investigate the spatial and temporal variability of  $\Delta^{17}O(O_3)$  values, which is expected to have the stable value, simultaneously with variations in  $\Delta^{17}O(SO_4^{-2})$  and  $\Delta^{17}O(NO_3)$  values. Antarctica is a suitable site to test this hypothesis, because of the clear seasonality in solar radiation, which is one of the main factors influencing  $\Delta^{17}O(SO_4^{-2})$  and  $\Delta^{17}O(NO_3^{-2})$  values through the changes in photochemical oxidants variations. In fact, several studies have been reported the clear seasonal variations in  $\Delta^{17}O(NO_3^{-2})$  values at coastal and inland Antarctica,
- 10 Dumont d'Urville and Dome C, showing an austral summer minimum and a winter maximum (Erbland et al., 2013; Frey et al., 2009; Savarino et al., 2007). This trend is mainly explained by increased  $NO_X$  oxidation by OH and  $RO_2$  under solar radiation in summer, relative to winter when  $NO_X$  oxidation is dominated by the reaction transferring the oxygen atoms from  $O_3$  to nitrate in dark polar winter, when assuming the values are mainly controlled by oxidation chemistry. The similar trend following solar cycle is expected for  $\Delta^{17}O(SO_4^{-2-})$  values, which is mainly controlled by the relative importance of  $SO_2$
- 15 oxidation by OH,  $H_2Q_2$  and  $O_3$ . The first attempt of the monthly scale observation of  $\Delta^{17}O(SO_4^{2-})$  values at Dome C showed the increasing trend from January to June (summer to winter) and decreasing trend from October to next January (spring to summer), despite the unexpected decline during mid-winter, July-August (Hill-Falkenthal et al., 2013). Nevertheless, the variability of  $\Delta^{17}O(SO_4^{2-})$  value throughout the year is larger than those which have ever been observed on the earth (Lee and Thiemens, 2001; Li et al., 2013; McCabe et al., 2006). It is thus ideal to examine if at Antarctica, the  $\Delta^{17}O(S_3)$  values
- 20 show a flat value throughout a year in contrast to those seasonal variability in  $\Delta^{17}O(SO_4^{--})$  and  $\Delta^{17}O(NO_3^{--})$  values, for demonstrating that the seasonality of  $\Delta^{17}O(SO_4^{--})$  and  $\Delta^{17}O(NO_3^{--})$  values is a direct consequence of changes in chemistry and not the variation in  $\Delta^{17}O(O_3)$  values.

In this study, we present the first simultaneous observations of  $\Delta^{17}$ O values of sulfate, nitrate and ozone at the coastal Antarctic site, Dumont d'Urville Station (DDU) throughout 2011, to answer the key question about the controlling factors of

- 25  $\Delta^{17}O(SO_4^{2-})$  and  $\Delta^{17}O(NO_3^{-})$  values as well as to provide a set of data that can be used to constrain chemistry/transport model scheme in the future study, The series of oxidants observations such as O3 (Legrand et al., 2009, 2016a), HOx (Kukui et al., 2012, 2014) and NOx (Grilli et al., 2013) have demonstrated that due to the katabatic winds, air masses on the East Antarctic Plateau enriched in oxidants produced via reactive nitrogen emission from surface snow and subsequent interactive reactions between HOx, NOx and RO2, are frequently exported to DDU. They also suggested the influence of bromine
- 30 chemistry is much less significant in East Antarctica compared to West coastal site such as Halley and Neumayer. However, besides the framework of that project, model-aided analyses of  $\Delta^{17}O(SO_4^{2^-})$  observations by Chen et al. (2016) pointed the significant contribution (33-50%) of SO2 oxidation by hypohalous acids on total sulfate production in remote marine boundary layer including Southern Ocean, indicating the possibility of its influence in coastal Antarctic site. Hence, we

3

**kiko ISHINO 7/12/2016 08:3**

**N**: Thus, the  $\Delta^{17}$ O values of both species trapped in ice cores reflect the past regional oxidation chemistry involved in sulfur and nitrogen cycles, and can provide insight into past oxidant variations (e.g., Alexander et al., 2004; Kunasek et al., 2010; Sofen et al., 2014).

**Sakiko ISHINO 15/2/2017 15:46**

削餘: . These measurements were conducted within the framework of the Oxidant Production over Antarctic Land and its Export project (OPALE; Preunkert et al., 2012), which enabled us to combine the isotopic measurements of sulfate and nitrate with various meteorological and other oxidant observations.

**Sakiko ISHINO 14/12/2016 12:49**

**2 Samples and analytical methods**

**5 2.1 Sampling site and aerosol sample collection**

**2.1.1 Sampling site**

Samples were collected at DDU ( $66^{\circ}40$ 'S,  $140^{\circ}01$ 'E; 40 m above the sea level), located on a small island, 1 km off the coast of Antarctica. The climate of DDU is described in Konig-Langlo et al. (1998). Compared with other parts of Antarctica, DDU is temperate, with temperatures ranging from -30 to 5°C throughout the year. Most parts of the island are free of snow,

and the sea ice disappears completely during summer. Recent observations of surface ozone and the OH radical (Legrand et al., 2016a; Kukui et al., 2012), revealed that at DDU, O3 and OH levels are approximately 2 and 10 times higher, respectively, compared to the Palmer station (Jefferson et al., 1998), due to the transportation of air masses influenced by the snowpack emission of reactive nitrogen species on the East Antarctic Plateau and the subsequent oxidants productions.

**2.1.2 Aerosol sample collection**

- 15 Aerosol samples were collected using a high-volume air sampler (HVAS; General Metal Works GL 2000H Hi Vol TSP; Tisch Environmental, Cleves, OH, USA). Coarse (> 1 μm) and fine (< 1 μm) particles were collected separately, using a four-stage cascade impactor and a backup glass fiber filter, respectively. The slotted 12.7 cm × 17.8 cm glass fiber filters were mounted on the cascade impactor, while 20.3 cm × 25.4 cm glass fiber filters were used for backup. The HVAS was placed on a platform, 1 m above ground, 50 m from the coast, and 20 m away from the closest building. Aerosol collection
- 20 was carried out at weekly intervals, with flow rates of ~1.5 m3/min, yielding an average pumped air volume of 15000 m3 per sample. Samples collected between January 2011 and January 2012 were used for this study. Once per month, a field blank was checked by mounting filters onto the filter holder, and running the cascade impactor for 1 min. After each collection period, the filters were removed from the cascade impactor inside a clean chemical hood; they were

wrapped in aluminum foil and stored in plastic bags at -20°C. The four filters on the impactor stage were grouped together as "coarse" particle samples, while the backup filters were kept as "fine" particle samples. Samples were transported back to

4

University Grenoble Alpes (France) for chemical and isotopic analyses, while frozen.

**2.1.3 Quantification of ionic species**

The soluble compounds in the aerosols were extracted with ultra-pure water (Millipore filter, 18 MΩcm; EMD Millipore, MA, USA), according to the process described in Savarino et al. (2007); more than 98% of the initial water volume was recovered. Field blank filters were processed in the same way.

5 Small aliquots of these sample solutions were taken for quantification of ionic species. Anions (Cl-, NO3-, SO42-) and sodium (Na+) concentrations were analyzed using ion chromatography systems described in Savarino et al. (2007) and Jourdain and Legrand (2002), respectively.

Atmospheric concentrations of these ionic species were calculated using the aerosol *loading for each filter*, the mean filter blank values, and the air volume pumped through the filter. The air volume was corrected to standard temperature and

10 pressure (T = 273.15 K, p = 101325 Pa) based on meteorological data from DDU provided by Meteo France. The uncertainties for atmospheric concentrations were calculated by propagating the typical uncertainty of the ion chromatography analysis (5%) and standard deviation (1 $\sigma$ ) of filter blank values.

**2.2 Oxygen isotopic analyses of sulfate and nitrate in aerosols**

**2.2.1 Definition of triple oxygen isotopic compositions**

15 Given the two isotope ratios, notated as  ${}^{17}R$  (=  ${}^{17}O/{}^{16}O$ ) and  ${}^{18}R$  (=  ${}^{18}O/{}^{16}O$ ), stable oxygen isotope ratios are conventionally scaled using a delta ( $\delta$ ) notation:

$$\delta^{x} O = \frac{x_{R_{sample}}}{x_{R_{VSMOW}}} - 1$$

(1)

where  $R_{\text{VSMOW}}$  denotes the isotope ratio of the standard material, Vienna Standard Mean Ocean Water (VSMOW); and x is 17 or 18. Despite the robust relationship of the mass-dependent law ( $\delta^{17}\text{O} = 0.52 \times \delta^{18}\text{O}$ ) in most of the oxygen-containing

20 species, (e.g.,  $O_2$  and  $H_2O$ ), atmospheric ozone does not follow mass-dependent fractionation and possesses a significant positive  $\Delta^{17}O$  (=  $\delta^{17}O - 0.52 \times \delta^{18}O$ ), inherited from mass-independent fractionation associated with its formation process (Gao and Marcus, 2001). Since non-zero  $\Delta^{17}O$  values can be observed in various atmospheric species bearing oxygen atoms inherited from  $O_3$  (e.g., sulfate and nitrate), the  $\Delta^{17}O$  signature is a powerful tracer, used to investigate the relative contribution of  $O_3$  to oxidation processes.

**25 2.2.2 Oxygen isotopic analysis of sulfate and data correction**

All  $\Delta^{17}$ O values of sulfate were measured with an isotope ratio mass spectrometer (IRMS) (MAT253; Thermo Fisher Scientific, Bremen, Germany), coupled with an in-house measurement system at Tokyo Institute of Technology. The measurement system for  $\Delta^{17}$ O(SO42-) follows Savarino et al. (2001), with modifications described in several studies

**Sakiko ISHINO 21/2/2017 14:23**

filter blank measurements

(Schauer et al., 2012; Geng et al., 2013). Briefly, 1  $\mu$ mol of sulfate is separated from other ions using ion chromatography and chemically converted to silver sulfate (Ag2SO4). This Ag2SO4 powder is transported in a custom-made quartz cup, which is dropped into a furnace at 1000°C within a high temperature conversion elemental analyzer (TC/EA; Thermo Fisher Scientific, Bremen, Germany) and thermally decomposed into O2 and SO2. Gas products from this sample pyrolysis are

- 5 carried by ultrahigh-purity He (>99.99995 % purity; Japan Air Gases Co., Tokyo, Japan), which is first purified using a molecular sieve (5Å) held at -196°C (Hattori et al., 2015). The gas products O2 and SO2 are carried through a cleanup trap (trap 1) held at -196°C to trap SO2 and trace SO3, while O2 continues to another molecular sieve (5Å) in a 1/16 inch o. d. tubing trap (trap 2) held at -196°C to trap O2 separately from the other gas products. The O2 is purified using a gas chromatograph, with a CP-Molsieve (5Å) column (0.32 mm i.d., 30 m length, 10 µm film; Agilent Technologies Inc., Santa
- Clara, CA, USA) held at 40°C, before being introduced to the IRMS system to measure m/z = 32, 33, and 34. The inter-laboratory calibrated standards (Sulf-α, β and ε; Schauer et al., 2012) were used to assess the accuracy of our measurements; our values were in good agreement with published ones (Fig. 1). As discussed by Schauer et al. (2012), this method results in the oxygen isotope exchange between the O2 products and the quartz cups as well as the quartz reactor, which shifts δ17O, δ18O, and thus Δ17O measurements. The shift in Δ17O(SO42-) value is corrected by estimating the magnitude of the oxygen
- 15 isotope exchange with quartz materials, whose  $\Delta^{17}$ O value is assumed to be approximately 0 ‰ (Matsuhisa et al., 1978). The intercept of -0.03 in Fig. 1 also supports this assumption. Since  $\delta^{17}$ O and  $\delta^{18}$ O values of each quartz materials used in this study are not known, the corrected  $\delta^{17}$ O and  $\delta^{18}$ O values of SO42- shown in Supplementary materials are unreliable, and therefore we don't discuss these values. Note that those  $\delta^{17}$ O and  $\delta^{18}$ O values of SO42- are relative values to our O2 reference gas. The precision of  $\Delta^{17}$ O is typically better than ±0.2 ‰ based on replicate analyses of the standards.
- 20 Since sea salt sulfate aerosols (ss-SO42-) are of little importance to atmospheric sulfur oxidation processes (i.e.,  $\Delta^{17}O(ss-SO_4^{2-}) = 0\%_0$ ), both total sulfate concentrations and  $\Delta^{17}O$  values were corrected for their ss-SO42- component to obtain their non-sea salt sulfate (nss-SO42-) content, using Eq. (2) and (3) below.

 $[nss - SO_4^{2-}] = [total - SO_4^{2-}] - k \times [Na^+]$

$$\Delta^{17} O(\text{nss} - \text{SO}_4^{2^-}) = \frac{[\text{total} - \text{SO}_4^{2^-}]}{[\text{nss} - \text{SO}_4^{2^-}]} \times \Delta^{17} O(\text{total} - \text{SO}_4^{2^-})$$
(3)

- 25 where "total" is the quantity measured by ion chromatography, corresponding to the sum of ss- and nss-SO42- components; and k is the mass ratio of [SO42-]/[Na+] in sea water (0.25; Holland et al., 1986). To take into account sea salt chemical fractionation processes that affect the Antarctic region in winter, when temperatures drop below -8°C in the presence of seaice (Wagenbach et al., 1998), we used a k value of 0.13±0.04, estimated from the average at winter DDU previously by Jourdain and Legrand (2002) as well as confirmed by our own dataset; this was applied to samples collected from May to
- 30 October. Note that the sea salt fractionation is a chemical fractionation and is different from an isotopic fractionation. Eq. (3) is the isotope mass balance equation between ss- and nss-SO32-, with  $\Delta^{17}O(ss-SO_3^{2^-}) = 0$  ‰. The total uncertainties for  $\Delta^{17}O(nss-SO_4^{2^-})$  values were calculated using the precision of  $\Delta^{17}O$  measurement and the uncertainty of k value, resulting in

6

Sakiko ISHINO 9/12/2016 11:21 移動 (挿入) [1] Sakiko ISHINO 9/12/2016 11:22 移動 (挿入) [2] Sakiko ISHINO 14/12/2016 14:38 削除:

(2)

the uncertainty of  $\pm 1.9$  ‰ at maximum. The propagated error for both [nss-SO42-] and  $\Delta^{17}O(nss-SO_4^{2-})$  values are shown in Supplementary material.

The measurement of  $\Delta^{17}O(SO_4^{2^-})$  value were performed only for the fine mode samples, because sulfate in the coarse mode samples consists of more than 80% ss-SO42-. The influence of ornithogenic soil emission on Na+ and SO42- concentration was not taken into account, since it mainly affects supermicron (coarse) aerosols (Jourdain and Legrand, 2002).

5

**2.2.3 Oxygen isotopic analysis of nitrate**

The  $\Delta^{17}$ O value of nitrate was measured simultaneously with  $\delta^{18}$ O and  $\delta^{15}$ N values using a bacterial denitrifier method (Casciotti et al., 2002), coupled with IRMS measurement using our in-house peripheral system at University Grenoble Alpes (Morin et al., 2009). All nitrates in our samples were converted to N2O via bacterial denitrification. This N2O was introduced

- 10 to the measurement system, separated from CO2, H2O and other volatile organic compounds, and pre-concentrated in a cold trap. The trapped N2O was converted into O2 and N2 by pyrolysis at 900°C, using a gold tube furnace, followed by separation of O2 and N2 via a 10 m Molsieve (5Å) gas chromatography column, before being introduced to the IRMS system. Measurements were performed simultaneously for samples equivalent to 100 nmol nitrate, as well as a subset of international nitrate reference materials (US Geological Survey 32, 34, and 35, as well as their mixtures) for correction and calibration of
- 15  $\Delta^{17}$ O and  $\delta^{18}$ O values relative to VSMOW and  $\delta^{15}$ N values relative to air N2. Analytical uncertainty was estimated based on the standard deviation of the residuals from a linear regression between the measured reference materials and their expected values. The uncertainties (1  $\sigma$ ) for  $\Delta^{17}$ O(NO3-) and  $\delta^{15}$ N(NO3-) were 0.4 ‰ and 0.3 ‰, respectively.

**2.3 Sampling and analytical methods of oxygen isotopic composition of ozone**

The sampling and isotopic analysis of surface ozone were performed by coupling the nitrite-coated filter method with nitrate isotopic measurements described in Vicars et al. (2012, 2014). The principle of ozone collection underlying this technique is

the filter-based chemical trapping of ozone via its reaction with nitrite:

 $\mathrm{NO}_2^- + \mathrm{O}_3 \rightarrow \mathrm{NO}_3^- + \mathrm{O}_2$

25

During R1, one of the three oxygen atoms of nitrate is transferred from one of the two terminal oxygen atoms of ozone, while the other two oxygen atoms are derived from the reagent nitrite. Since the  $\Delta^{17}$ O signature of ozone is located only on the terminal atoms of ozone (Bhattacharya et al., 2008; Janssen and Tuzson, 2006), simple mass balance implies that

 $\Delta^{17}O(O_3)_{\text{term}} \text{ is } 2/3 \text{ of } \Delta^{17}O(O_3)_{\text{bulk}}. \text{ Thus, } \Delta^{17}O(O_3)_{\text{term}} \text{ values can be inferred using the simple mass-balance of Eq. (4):}$  $\Delta^{17}O(O_3)_{\text{term.}} = 3 \times \Delta^{17}O(NO_3^-) - 2 \times \Delta^{17}O(NaNO_2), \qquad (4)$ where  $\Delta^{17}O(NaNO_2)$  of the reagent is confirmed to be zero (Vicars et al., 2012). Therefore, the  $\Delta^{17}O$  value of ozone can be

determined from the oxygen isotopic composition of nitrate produced on the coated filter via R1, determined by the same 30 measurement system described above.

were only carried out for Sakiko ISHINO 9/12/2016 11:21

上へ移動 [1]: Eq. (3) is the isotope mass balance equation between ss- and nss-SO42-, with  $\Delta^{17}O(ss-SO_4^{2^-}) = 0$  ‰.

akiko ISHINO 9/12/2016 11:22

上へ移動 [2]: The total uncertainties for  $\Delta^{17}$ O(nss: SO42-) values were calculated using the precision of  $\Delta^{17}$ O measurement and the uncertainty of k value. Sakiko ISHINO 20/2/2017 12:53

Sakiko ISHINO 7/2/2017 11:2 削除:

(R1)

Ozone sampling was carried out by pumping ambient air, using a low-volume vacuum pump (Model 2522C-02; Welch, IL, USA), through a glass fiber filter (Ø 47 mm, GF/A type; Whatman, UK), pre-coated with a mixture of NaNO2, K2CO3 and glycerol. Sampling was conducted once per week from May 2011 to April 2012, with 24–48 h sampling intervals. After sampling, filter samples and procedural blanks were extracted in 18MΩ water. Any unreacted nitrite reagent was removed

5 using the reaction with sulfamic acid, neutralized later with NaOH solutions (Granger and Sigman, 2009; Vicars et al., 2012). The sample solutions were stored in the dark at -20°C, and transported back to Grenoble. After nitrate concentration analysis using a colorimetric technique (Frey et al., 2009), the isotopic analysis of nitrate (i.e., ozone) was performed using the same protocol as the nitrate isotope analysis. In addition to the isotope measurements of ozone, we aligned the mixing ratio of surface ozone to the weekly average using data reported in Legrand et al. (2016a) to fit the time resolution of our aerosol 10 sampling.

**2.4 Complementary analyses**

To investigate relationships between the origins of the air masses and the  $\Delta^{17}$ O signatures of sulfate and nitrate, transport pathways of sampled air masses were analyzed using the NOAA's HYSPLIT (Hybrid Single-Particle Lagrangian Integrated Trajectory) model (Stein et al., 2015). The model was used with NCEP-NCAR reanalysis data fields using a regular 2.5° × 2.5° longitude-latitude grid. Five-day backward trajectories for air masses arriving at the DDU at an altitude of 40 m

above sea level were computed twice per day for each day during sampling periods. The sea ice area fraction around the Antarctic continent was derived from the Advanced Microwave Scanning Radiometer on-board NASA's Earth Observing System Aqua satellite using the ARTIST sea ice algorithm (Kaleschke et al., 2001). The contact times of these air masses with the Antarctic continent and sea ice were calculated using five-day backward trajectories and sea ice area fractions.

**3** Results**

**3.1 Sulfate**

25

15

20

Seasonal variations in atmospheric concentrations and  $\Delta^{17}$ O values of SO42- are shown in Fig. 2a. Atmospheric concentrations of nss-SO42- showed a clear seasonal trend. The [nss-SO42-] had a maximum of up to ~280 ng m-3 from January to February, corresponding to the austral summer period, but decreased to a background level (~10 ng m-3) during May to August, winter period, before increasing as summer returned. This trend in [nss-SO42-] at coastal Antarctic sites results from enhanced marine biogenic activity, emitting dimethyl sulfide (DMS) in circum-Antarctic regions in summer, as has often been reported previously (e.g., Wagenbach et al., 1998; Minikin et al., 1998; Jourdain and Legrand, 2002;

Sakiko ISHINO 1/12/2016 13:41

Preunkert et al., 2008). As Antarctica is surrounded by ocean, DMS is the major source of atmospheric non-sea salt sulfur (Minikin et al., 1998; Jourdain and Legrand, 2002). Interestingly, a sample from 18-25 July had an anomalously high value of 46 ng m-3, four times the monthly mean level for July ( $\sim$ 12 ng m-3).

The  $\Delta^{17}O(nss-SO_4^{2-})$  values showed the reverse trend, with a summer minimum and a winter maximum. The  $\Delta^{17}O(nss-SO_4^{2-})$ 5 SO42-) value increased from 1.0 ‰ observed in January to a maximum of 3.4 ‰ at the end of June, decreasing to 0.9 ‰ in December. The annual weighted mean value of  $\Delta^{17}O(nss-SO_4^{2-})$  was  $1.4 \pm 0.1$  ‰. Higher values (greater than 2 ‰) were generally observed during April to July, but the anomalous peak from 18–25 July was characterized by a low  $\Delta^{17}O(nss SO_4^{-}$ ) value of 0.9 ‰. Consequently, the monthly mean value had a maximum in July (2.6 ± 0.6 ‰), when the 18–25 July data were excluded.

**10 3.2 Nitrate**

Seasonal variations in atmospheric concentrations and  $\Delta^{17}O$  values for nitrate are shown in Fig. 2b. Nitrate concentrations increased to 55 ng m-3 in January but gradually decreased to less than 10 ng m-3 in March to May. In July, a significant peak of 28 ng m-3 was observed, followed by a seasonal increase as summer returned. The  $\Delta^{17}O(NO_3^{-7})$  values showed a simple seasonal variation, with a summer minimum and a winter maximum.  $\Delta^{17}O(NO_3^{-})$  increased from 27 ‰ in January to over 40 ‰ in July, decreasing moderately to a minimum value of 23 ‰ in December. These trends in nitrate concentrations and

15  $\Delta^{17}O(NO_3^{-})$  values are consistent with those observed at this site 10 years ago (Savarino et al., 2007).

**3.3 Ozone**

Daily averaged ozone mixing ratios are presented in Fig. 2c; these exhibit a distinct seasonal variation, with a summer

20

minimum and a winter maximum. The minimum ozone mixing ratio was observed in January, having a value lower than 10 ppbv, while the maximum was observed during July to August, having a value higher than 35 ppbv. From November to December, sudden increases in ozone levels to values over 30 ppbv were observed a few times, consistent with seasonal trends for ozone at DDU (Legrand et al., 2009). The  $\Delta^{17}O(O_3)_{bulk}$  values showed an insignificant variation, with a summer maximum of 28 ‰ and a winter minimum of 23 ‰, and an annual mean  $\Delta^{17}O(O_3)_{bulk}$  value of  $26 \pm 1$  ‰.

**4 Discussion**

5

**4.1 $\Delta^{17}O$ values and atmospheric formation pathways of sulfate and nitrate**

Atmospheric sulfate and nitrate are produced from the oxidation of their precursor, SO2 and NOX, by various oxidants. Therefore,  $\Delta^{17}O(SO_4^2^-)$  and  $\Delta^{17}O(NO_3^-)$  values for sulfate and nitrate produced via each oxidation pathways are determined by the  $\Delta^{17}O$  values of their precursors and the  $\Delta^{17}O$  values of oxidants, which provides oxygen atom to the products in different transferring factors. To interpret the our data, we estimated the  $\Delta^{17}O(SO_4^2^-)$  and  $\Delta^{17}O(NO_3^-)$  values produced via each oxidation pathways. The  $\Delta^{17}O(SO_4^2^-)$  and  $\Delta^{17}O(NO_3^-)$  values for the oxidation by O3 were estimated using the mean  $\Delta^{17}O(O_3)$  value (i.e., 26 ‰) observed in this study. Each oxidation pathways and corresponding  $\Delta^{17}O$  values of products sulfate and nitrate are summarized in Table 1.

**10 4.1.1 $\Delta^{17}$ O values of sulfate**

Since SO2 quickly exchanges its oxygen atoms with abundant water vapor in the atmosphere, the  $\Delta^{17}O(SO_2)$  value is assumed to be 0 ‰ (Holt et al., 1983). Thus, the  $\Delta^{17}O(SO_4^{2-})$  value is dependent only on the oxidation pathway of SO2 to SO42-. SO2 oxidation by OH ( $\Delta^{17}O(OH) = \sim 0$  ‰) in the gas phase produces sulfuric acid (H2SO4) which possesses the  $\Delta^{17}O(SO_4^{2-})$  value of approximately 0‰.

| 15 | $SO_2 + OH \xrightarrow{O_2,H_2O,M} H_2SO_4$ (R2)                                                                                                     | Sakiko ISHINO 24/12/2016 13:46                                                                                                                                                      |
|----|-------------------------------------------------------------------------------------------------------------------------------------------------------|-------------------------------------------------------------------------------------------------------------------------------------------------------------------------------------|
|    | $SO_2$ can also dissolve into the aqueous phase on aerosol surfaces, where it can be oxidized by $O_3$ , $H_2O_2$ or metal-catalyzed                  | 前除: $\Delta^{17}O(SO_4^{2-}) = 0\%$                                                                                                                                                 |
|    | oxidation by O 2 , to form sulfate (Seinfeld and Pandis, 2006), Given that $\Delta^{17}O(O_3)_{bulk}$ values of approximately 26 % have    |                                                                                                                                                                                     |
|    | been observed, the $\Delta^{17}O(SO_4^{2^-})$ value of sulfate produced by ozone should be around 6.5 ‰, based on a $\Delta^{17}O$ signature          | Sakiko ISHINO 24/12/2016 13:47                                                                                                                                                      |
| I  | transfer factor of 0.25 (Savarino et al., 2000).                                                                                                      | H p : where it can react with various oxidants $(O_3, H_2O_2 \text{ or } O_2)$ catalyzed by transition metal ions (such as Fe(III) and Mn(II)) to form sulfate (Seinfeld and |
| 20 | $SO_3^{2-} + O_3 \longrightarrow SO_4^{2-} + O_2$ (R3)                                                                                                | Pandis, 2012).                                                                                                                                                                      |
| 1  | Give the $\Delta^{17}O(H_2O_2)$ values of 1.6 ‰ on average (Savarino et al., 1999), the $\Delta^{17}O(SO_4^{2^-})$ of sulfate produced by $H_2O_2$ is | Sakiko ISHINO 28/12/2016 21:16
|    | estimated to be 0.8 ‰, using a transfer factor of 0.5 (Savarino et al., 2000).                                                                        | Sakiko ISHINO 24/12/2016 13:46                                                                                                                                                      |
|    | $HSO_3^{2-} + H_2O_2  HSO_4^{-} + H_2O  (R4)$                                                                                                         | 削除: $\Delta^{17}O(SO_4^{2-}) = 6.5\%$                                                                                                                                               |
| I  | The $\Delta^{17}O(O_2)$ value was measured to be -0.3 % (Barkan and Luz, 2003), producing sulfate with a $\Delta^{17}O(SO_4^{2-})$ value of           | Sakiko ISHINO 24/12/2016 13:47                                                                                                                                                      |
| 25 | almost 0 ‰ (Savarino et al., 2000).                                                                                                                   | 則除: $\Delta^{1/}O(SO_4^{-}) = 0.8\%$                                                                                                                                                |
|    | $SO_3^{2-} + O_2 \xrightarrow{Fe,Mn} SO_4^{2-}$ (R5)                                                                                                  | Sakiko ISHINO 24/12/2016 13:47                                                                                                                                                      |
|    | Additionally, aqueous phase $SO_2$ oxidation by hypohalous acids (HOX = HOCl, HOBr) has been proposed as one of the                                   | 削除: $\Delta^{17}O(SO_4^{2-}) = -0.1\%$                                                                                                                                              |

10

major reactions in marine boundary layer (Vogt et al., 1996; von Glasow et al., 2002). Details are discussed in section 4.3.

Thus,  $\Delta^{17}O(SO_4^{2-})$  of nss-SO42- results from a subtle balance between various oxidation reactions, each one transferring a specific amount of  $\Delta^{17}$ O signature to sulfate.

**4.1.2 $\Delta^{17}$ O values of nitrate**

The  $\Delta^{17}O(NO_3^-)$  value is dependent on both the  $\Delta^{17}O(NO_2)$  value and the oxidation pathways of  $NO_2$  to  $NO_3^-$ . The  $5 = \Lambda^{17} \Omega(NO_2)$  is determined by the relative contribution of NO oxidation nations during the following photochemical cycle

| С  | $\Delta O(NO_2)$ is determined by the relative contribution of NO oxidation pathways during the following photochemical cycle                                                                                                          |                                                                                                |
|----|-----------------------------------------------------------------------------------------------------------------------------------------------------------------------------------------------------------------------------------------------|------------------------------------------------------------------------------------------------|
|    | $NO_2 \xrightarrow{hv} NO + O(^1D)$ (R6)                                                                                                                                                                                                      | Sakiko ISHINO 24/12/2016 13:49
|    | $N0 + 0_3 \longrightarrow N0_2 + 0_2 $ (R7)                                                                                                                                                                                                   |                                                                                                |
|    | $NO + RO_2 \longrightarrow NO_2 + RO$ (R8)                                                                                                                                                                                                    |                                                                                                |
|    | Since all non-zero $\Delta^{17}$ O of ozone is positioned in the terminal oxygen atoms (Bhattacharya et al., 2008), which preferential                                                                                                        | ly                                                                                             |
| 10 | react with NO (Savarino et al., 2008), NO 2 formed by ozone exhibits a higher isotopic value than the bulk $\Delta^{17}O(O_3)$ . On the bulk $\Delta^{17}O(O_3)$ is the bulk $\Delta^{17}O(O_3)$ is the bulk $\Delta^{17}O(O_3)$ . | he                                                                                             |
|    | other hand, NO + RO 2 reaction produces the nitrate with lower $\Delta^{17}O(NO_3^{-})$ value because $\Delta^{17}O(RO_2)$ is approximately 0.9                                                                                    | %                                                                                       |
|    | (Morin et al., 2007). NO 2 is then converted into nitrate through one of the following reactions.                                                                                                                                  |                                                                                                |
|    | $NO_2 + OH \xrightarrow{M} HNO_3$ (R9)                                                                                                                                                                                                        |                                                                                                |
|    | $NO_2 + O_3 \longrightarrow NO_3$ (R10)                                                                                                                                                                                                       |                                                                                                |
| 15 | $NO_3 \xrightarrow{hv} NO_2 + O(^1D)$ (R11)                                                                                                                                                                                                   |                                                                                                |
|    | $NO_3 + RH \longrightarrow HNO_3 + products$ (R12)                                                                                                                                                                                            |                                                                                                |
|    | $NO_3 + NO_2 \iff N_2O_5 \xrightarrow{H_2O} 2HNO_3$ (R13)                                                                                                                                                                                     |                                                                                                |
|    | It has been pointed out that BrO plays a significant role in both NO and NO 2 oxidation in marine boundary layer (Savarino                                                                                                         | et                                                                                             |
|    | al., 2013) through the following reactions;                                                                                                                                                                                                   |                                                                                                |
| 20 | $NO + BrO \longrightarrow NO_2 + Br$ , (R14)                                                                                                                                                                                                  |                                                                                                |
|    | $NO_2 + BrO \longrightarrow BrONO_2 \xrightarrow{H_2O} HNO_2 + HOBr$ (R15)                                                                                                                                                                    | Sakiko ISHINO 24/12/2016 13:52                                                                 |
|    | while they are thought to have little importance (2% at maximum) on the Antarctic Plateau during the austral summer, due                                                                                                                      | to NO 2 with high $\Delta^{17}$ O values (Morin et al., 2007),                      |
|    | low BrO levels up to 2-3 pmol/mol (Frey et al., 2015; Savarino et al., 2016). The oxidation by BrO may also produce NO                                                                                                                        | O 2 because BIO is inought to possess the terminal oxygen atom of ozone.            |
|    | and nitrate with high $\Delta^{17}$ O values (Morin et al., 2007), because BrO is thought to possess the terminal oxygen atom of ozor                                                                                                         | ne                                                                                             |
| 25 | (Bhattacharya et al., 2008).                                                                                                                                                                                                                  |                                                                                                |
|    | Following the principle that two of the three oxygen atoms in $NO_3^-$ come from $NO_2$ and one arises through conversion                                                                                                                     | of                                                                                             |
| l  | NO 2 to NO 3 - , the $\Delta^{17}O(NO_3^{-})$ value of nitrate produced by each pathway can be expressed as Eq. (5).                                                                                         | Sakiko ISHINO 24/12/2016 13:53                                                                 |
|    |                                                                                                                                                                                                                                               | - TR   UT                                                                               |

 $\Delta^{17}O(NO_3^-) = \frac{2}{3} \times \Delta^{17}O(NO_2) + \frac{1}{3} \times \Delta^{17}O(Oxidant)$

(5)

For a given value of  $\Delta^{17}O(NO_2)$ , the NO2 + OH pathway produces the lowest  $\Delta^{17}O(NO_3)$  value, while the NO3 + RH pathway or BrONO2 hydrolysis produce the highest  $\Delta^{17}O(NO_3^{-})$  values.

**4.2 General trend of seasonal variations in $\Delta^{17}$ O values of sulfate and nitrate**

The  $\Delta^{17}$ O signatures of atmospheric sulfate and nitrate originate from the oxygen transfers from ozone, via oxidation of their precursors. Thus, changes in  $\Delta^{17}O(O_3)$  values likely affect both  $\Delta^{17}O(nss-SO_4^{2^-})$  and  $\Delta^{17}O(NO_3^-)$  values. However, given the small seasonal variability of the  $\Delta^{17}O(O_3)_{bulk}$  (ca. 5 %), and assuming that all oxygen atoms transferred to NOx are from the

terminal oxygen of ozone, the expected variability of  $\Delta^{17}O(nss-SO_4^{2-})$  and  $\Delta^{17}O(NO_3^{-})$  should not exceed 1.3 ‰ and 7.5 ‰, respectively. Clearly, these upper limits do not explain the 2.5 % and 19 % seasonal variability observed in  $\Delta^{17}O(nss-SO_4^{2^-})$ and  $\Delta^{17}O(NO_3)$  at DDU, respectively. Furthermore, the seasonal variation in  $\Delta^{17}O(O_3)$  values, with a summer maximum and

a winter minimum, is the reverse pattern to  $\Delta^{17}O(nss-SO_4^{2-})$  and  $\Delta^{17}O(NO_3^{-})$  values, with summer minima and winter 10 maxima. These inconsistencies suggest that variability in  $\Delta^{17}O(O_3)$  values is not the major factor influencing the seasonal variations in  $\Delta^{17}O(nss-SO_4^{2^-})$  and  $\Delta^{17}O(NO_3^-)$  values.

Meanwhile,  $\Delta^{17}O(nss-SO_4^{2-})$  and  $\Delta^{17}O(NO_3^{-})$  values are dependent on the relative importance of various oxidation pathways involved in their formation as described in the previous sections. Since the relative importance of these oxidation pathways

- is sensitive to the relative concentrations of oxidants in the atmosphere, and there is seasonal variation for ozone mixing 15 ratios at a continental scale (Crawford et al., 2001; Legrand et al., 2009), the mixing ratio of ozone is expected to correlate with  $\Delta^{17}O(\text{nss-SO}_4^{-7})$  and  $\Delta^{17}O(\text{NO}_3^{-7})$  values. Indeed,  $\Delta^{17}O(\text{nss-SO}_4^{-7})$  and  $\Delta^{17}O(\text{NO}_3^{-7})$  values, as well as ozone mixing ratios, all display similar seasonal variations, as shown in Fig. 2. The seasonal variation in the ozone mixing ratios at DDU is generally explained by accumulation of ozone in winter, and its photochemical destruction in summer (Legrand et al., 2009;
- 20 2016a), which induces the production of  $HO_X$ ,  $RO_X$ , and  $H_2O_X$  in the summer period. Therefore, we propose that seasonal variations in  $\Delta^{17}O(nss-SO_4^{-2})$  and  $\Delta^{17}O(NO_3^{-1})$  result from a shift in oxidation pathways from O3 to HOX, ROX, and H2O4 Decreases in  $\Delta^{17}O(nss-SO_4^{2-})$  and  $\Delta^{17}O(NO_3^{-})$  values are caused by the combining effect of decrease in the ozone concentration and decrease in the transfer efficiency of  $\Delta^{17}O(O_3)$  to the final products. Thus, the changes in relative concentrations of O3 vs. HOx, ROx, and H2O3, along with the changes in sunlight level, are the main factors controlling the seasonal variations of  $\Delta^{17}O(nss-SO_4^{2-})$  and  $\Delta^{17}O(NO_3^{-})$  values.

25

5

A similar seasonal variation in  $\Delta^{17}O(nss-SO_4^{2-})$  values has been observed at Dome C, an inland Antarctic site (Hill-Falkenthal et al., 2013). However, the  $\Delta^{17}O(nss-SO_4^{2-})$  values observed at Dome C significantly declined in July and August, in contrast to our observations that showed only a single significant decline in  $\Delta^{17}O(nss-SO_4^{2-})$  values during the period of 18–25 July. This low  $\Delta^{17}O(nss-SO_4^{2-})$  sample is also characterized by high nss-SO42- concentration (Fig. 2a). We don't have

30 any evidence of contamination from station activities or laboratory works. The preliminary result of sulfur isotope analysis of sulfate in the same sample, showing  $\delta^{34}$ S value of 17.6 ‰ (to be published), suggests that this sulfate results from marine

12

akiko ISHIN 削除: SO2 and

|   | Sakiko ISHINO 14/12/2016 10:46                               |
|---|--------------------------------------------------------------|
| ١ | Sakiko ISHINO 14/12/2016 10:46                               |
| Ì | Sakiko ISHINO 14/12/2016 10:47                               |
| ١ | Sakiko ISHINO 14/12/2016 10:47                               |
| ١ | Sakiko ISHINO 14/12/2016 10:48                               |
|   |                                                              |

biogenic sulfur (i.e., DMS) which possesses  $\delta^{34}$ S values ranging 16–20 ‰ (Oduro et al., 2012; Amrani et al., 2013). However, the results of the back trajectory analyses exhibit that DDU was under continental outflow condition over this period, as well as throughout July (Fig. 3). It is hence difficult to identify the origin of this low  $\Delta^{17}$ O(nss-SO42-) value during 18–25 July. Nevertheless, this point doesn't change the interpretation that  $\Delta^{17}$ O(nss-SO42-) values are generally lower in

5 summer and higher in winter. Thus, this sample was excluded from consideration for the following discussion and does not impair our further interpretation.

For sulfate, the relative contribution of  $O_3$  oxidation for sulfate formation ( $f(nss-SO_4^2)_{03}$ ) was calculated. Since positive  $\Delta^{17}O(nss-SO_4^2)$  values result from only  $H_2O_2$  and  $O_3$  oxidation and the other pathways result in  $\Delta^{17}O(nss-SO_4^2)$  of approximately 0 ‰, we calculated the maximum and the minimum  $f(nss-SO_4^2)_{03}$  by assuming no contribution of  $H_2O_2$

- 10 oxidation, and assuming the contribution from only  $H_2O_2$  and  $O_3$  oxidation, respectively, using the simple mass balance equation. Consequently, the mean summer (Jan., Feb., and Dec.)  $\Delta^{17}O(nss-SO_4^{-2})$  value of 1.2 ‰ is corresponding to *f*(nss- $SO_4^{-2}O_{03}$  of 0.07-0.18, whereas the mean winter (Jun. Aug.)  $\Delta^{17}O(nss-SO_4^{-2})$  value of 2.4 ‰ is corresponding to *f*(nss- $SO_4^{-2}O_{03}$  of 0.28-0.37. The relative contribution of  $O_3$  oxidation is thus 2 to 4 folds higher in winter than in summer. However, it is important to mention that DMS levels are quite low at DDU in winter (Preunkert et al. 2007); the sulfate
- 15 collected at DDU is likely produced at a lower latitude region under more sunlight. Thus, the observed  $\Delta^{17}O(\text{nss-SO}_4^{2-})$ values in winter may not solely reflect oxidation chemistry in the local atmosphere at DDU. The sulfur sources need to be constrained for the interpretation of winter data. Also for nitrate, it is believed that during Antarctic winter period, the atmospheric nitrate results mainly from the deposition of polar stratospheric clouds (Santacesaria et al., 2001; Savarino et al, 2007) and  $\Delta^{17}O(\text{NO}_3^-)$  values are not representative of the oxidation chemistry of the atmosphere at this site. On the other
- 20 hand, since a mean summer  $\Delta^{17}O(NO_3)$  value of 27.1 ‰ requires the contribution of  $NO_2+OH$  pathway ( $f(NO_3)_{OH}$ ), the maximum and minimum  $f(NO_3)_{OH}$  value was calculated by assuming no contribution of  $N_2O_5$  hydrolysis, and by assuming no contribution of  $NO_3+RH$  and BrONO2 hydrolysis pathways, respectively. As a result,  $f(NO_3)_{OH}$  value corresponding to the summer  $\Delta^{17}O(NO_3)$  value of 27.1 ‰ is expected to range 0.28-0.52. However, this is in contradiction to an expectation that the termination reaction is only  $NO_2+OH$  pathway under permanent sunlight. For further constraints on oxidation
- 25 chemistry, a coupled stratosphere/troposphere chemical transport model will be necessary, which is beyond the scope of this paper.

**4.3 Sensitivity of sulfate and nitrate $\Delta^{17}O$ values to the ozone mixing ratio**

30

To examine the response of  $\Delta^{17}O$  values to the changes in oxidant concentration, we compared  $\Delta^{17}O(\text{nss-SO}_4^2)$  and  $\Delta^{17}O(\text{NO}_3)$  values to ozone mixing ratio (Figure 4), which was only one oxidant observed in year-round scale at the same time. In Fig. 4,  $\Delta^{17}O(\text{nss-SO}_4^2)$  and  $\Delta^{17}O(\text{NO}_3)$  values are generally co-varied with ozone mixing ratios, suggesting that a change in the ozone mixing ratio is one of the main factors controlling  $\Delta^{17}O(\text{nss-SO}_4^2)$  and  $\Delta^{17}O(\text{NO}_3)$  values. Note that as

13

[revised manuscript text omitted]

- 5 increase oxidants at DDU, compared with the Palmer Station, about 2-fold for O3 and more than 10-fold for OH. Chemical transport models over the Antarctic continent show that NOx emitted from snow during summer increase O3 and OH by a factor of 2 and 7, respectively, compared with estimations not including snow NOx emissions (Zatko et al., 2016). Additionally, 15N depletion in nitrate starts from the beginning of September (See Supplement), which is consistent with previous measurement at DDU (Savarino et al., 2007), supporting that the snow NOx emission happens in early spring at
- 10 DDU. Thus, by snow NOX emission, OH production is enhanced more efficiently than O3 production in spring, possibly resulting in lower  $\Delta^{17}O(\text{nss-SO}_4^{2^-})$  spring values.

Another possible explanation is that hypohalous acids (HOX = HOCl, HOBr) act as important oxidants of SO2 via the aqueous phase reaction ( $R_16$  and  $R_17$ ) in the marine boundary layer (Fogelman et al., 1989; von Glasow, 2002):

$$\begin{array}{cccc} HOX + SO_3^{2-} \rightarrow OH^- + XSO_3^{2-} & , & (R\_6) \\ 15 & XSO_3^{2-} + H_2O \rightarrow SO_4^{2-} + X^- + 2H^+ & . & (R\_7) \end{array}$$

This reaction is expected to produce sulfate with  $\Delta^{17}O = 0$  ‰, as all oxygen atoms of sulfate originate from water (Fogelman et al., 1989; Troy and Margerum, 1991: Yiin and Margerum, 1988), leading to lower  $\Delta^{17}O(nss-SO_4^{2^-})$  values in the atmosphere. Indeed, unexpectedly low  $\Delta^{17}O(nss-SO_4^{2^-})$  values have been observed in marine aerosols, which is possibly explained by a contribution of HOX oxidation of 33%–50% to total sulfate production in the marine boundary layer (Chen et

- al., 2016). Chen et al. (2016) estimated that a minimum concentration of gaseous HOX of 0.1 pptv could account for half of the sulfate production in the marine boundary layer. At DDU, year-round observations of gaseous inorganic bromine species ([Bry\*] = [HBr] + [HOBr] + 0.9[Br2] + 0.4[BrO] + [BrNO2] + [BrONO2] + [Br]) revealed that maximum concentrations are observed in September, with values of 13.0 ± 6.5 ng m-3 (~ 3.6 pptv) (Legrand et al., 2016b). Even if only one third of the Bry\* corresponds to HOBr, as estimated using the model calculations by Legrand et al. (2016b) under summer conditions,
- 25 then it is expected that HOX at DDU in spring is > 1 pptv; thus, HOX could play a significant role in sulfate production in spring. Likewise,  $Br_y^*$  concentration at DDU is at minimum in May, with values of  $3.4 \pm 1.0$  ng m-3 (Legrand et al., 2016b), corresponding to less than one third of spring values. This would lead to a lower contribution of HOX to sulfate production in fall, compared with spring. Hence, sulfate production via aqueous oxidation by HOX may explain the lower  $\Delta^{17}O(SO_4^{2-})$  in spring, relative to fall.
- 30 A change in pH of the aqueous phase on the aerosol surfaces may also explain the spring/fall difference in  $\Delta^{17}O(\text{nss-SO}_4^{-7})$ . Given that SO2 oxidation by ozone in the aqueous phase is favored at high pH (>5.5) (Seinfeld and Pandis, 2006), if the pH of aerosol droplets is higher in fall than in spring, then the relative importance of the SO2 + O3 reaction resulting in higher  $\Delta^{17}O$  values would increase in fall. However, ion concentration analyses of aerosols collected at DDU exhibited higher

15

Sakiko ISHINO 14/12/2016 08:2 削除: 12 alkalinity in spring than in fall (Jourdain and Legrand, 2002; Legrand et al., 2016b), which is inconsistent with this explanation. Hence, this process can be excluded from further consideration.

It should be noted that smaller difference was observed between spring and fall  $\Delta^{17}O(NO_3^{-})$  values, which would be also affected by the above two processes. Snow NOX emission would decrease  $\Delta^{17}O(NO_3^{-})$  through depression of the

5 contribution of  $O_{3}$  oxidation relative to the other oxidation pathways by  $HO_X$  and  $RO_3$  while halogen chemistry would lead to high  $\Delta^{17}O(NO_3^-)$  values through an oxygen atom transfer from O3 to BrO and consequently to nitrate (Morin et al., 2007; Savarino et al., 2013). Although it is difficult to identify the precise processes involved, observations of  $\Delta^{17}O(nss-SO_4^{2-})$  and  $\Delta^{17}O(NO_3^-)$  values at an inland site (e.g., Concordia Station) would enable us to determine which process causes the spring/fall difference in the oxidation chemistry in the DDU atmosphere. If snow NOX emission is the source of low

10  $\Delta^{17}O(\text{nss-SO}_4^{2-})$  at DDU in spring, then  $\Delta^{17}O(\text{nss-SO}_4^{2-})$  and  $\Delta^{17}O(\text{NO}_3^{-})$  at an inland site would also exhibit lower values in spring than in fall.

**4.4 Air mass origin analysis**

Using observations of several oxidants at DDU and Concordia Station (e.g., Legrand et al., 2009, 2016a; Kukui et al, 2012, 2014; Grilli et al., 2013), it has been suggested that the oxidative capacity of the atmosphere at DDU is influenced by air

- 15 masses transported from the East Antarctic Plateau during katabatic wind outflows. The NOX emission from snowpack in inland Antarctica stimulates the ozone, HOX, ROX, and H2O2 production through the enhanced NOX cycle. Thus, the atmosphere at DDU is enriched in all of those photochemical oxidants, when air masses are from inland regions, compared to when air masses are from the ocean (Legrand et al., 2009). Therefore, we expected that  $\Delta^{17}$ O values for sulfate and nitrate also depend on air mass origin.
- 20 In Fig. 5, the  $\Delta^{17}O$  values for sulfate and nitrate are given as a function of the time that air masses were over the continent during the five-days travel prior to arriving at DDU. Summer data show that the contact times of air masses with the continent varied between 20 and 120 h (i.e., continental air and oceanic air are well mixed). Similarly, their  $\Delta^{17}O$  values show insignificant variation, having low values of around 1‰ and 25‰ for sulfate and nitrate, respectively. This trend reflects two different phenomena, decreased  $\Delta^{17}O$  values in summer because of the high contribution of photo-oxidants to
- 25 atmospheric chemistry, and increased import of oceanic air. In contrast, plots for other seasons show that the  $\Delta^{17}$ O values exhibit high variation, although most of the air masses originate from the continent. It is important to note that this noncorrelation does not mean that there is no link between  $\Delta^{17}$ O values and air mass origin. The influence of air mass transport on oxidative capacity has been demonstrated in daily observations of oxidants (e.g., Legrand et al., 2009). Given that air masses at this site are from a variety of directions, and are mixed together, this makes interpretation of weekly averaged
- 30 analyses more complicated. Hence, no significant correlation between  $\Delta^{17}$ O values and air mass origin in weekly data could reflect a real lack of correlation or a too broad time resolution for these data.

16

| Sakika ISHINO 8/2/2017 10:29                                            |
|-------------------------------------------------------------------------|
| Sakiko ISHINO 8/2/2017 19:38                                            |
| Sakiko ISHINO 21/2/2017 16:22                                           |
| Sakiko ISHINO 8/2/2017 19:38                                            |
| Sakiko ISHINO 1/12/2016 13:50                                           |
| Sakiko ISHINO 14/12/2016 10:51                                          |
| Sakiko ISHINO 1/12/2016 13:51                                           |
| Sakiko ISHINO 1/12/2016 13:52                                           |
| Sakiko ISHINO 1/12/2016 13:53                                           |
| Sakiko ISHINO 1/12/2016 13:54                                           |
|                                                                         |
|                                                                         |

**5 Summary**

To develop an understanding of the factors influencing  $\Delta^{17}$ O values of atmospheric sulfate and nitrate, seasonal variations of

- Δ17O values of atmospheric sulfate, nitrate and ozone were analyzed using the aerosol samples collected at DDU throughout
   2011. Both Δ17O(nss-SO42-) and Δ17O(NO3-) values exhibited clear seasonal variations, with summer minima and winter maxima. In contrast, Δ17O values of ozone showed limited variability throughout the year, indicating that Δ17O(O3) values do not significantly influence summer/winter trends in Δ17O(nss-SO42-) and Δ17O(NO3-) values. We hence, for the first time, demonstrated that Δ17O(nss-SO42-) and Δ17O(NO3-) values are direct results of local oxidation chemistry of their precursors. The summer/winter trends of Δ17O(nss-SO42-) and Δ17O(NO3-) values are likely to reflect sunlight-driven changes in the
- 10 relative importance of  $\rho$ xidation pathways; oxidation by HOX, ROX and H2O2 are increased during summer when the solar radiation enhances the production of those oxidants, whereas the relative contribution of oxidation reaction transferring oxygen atoms of O2 to sulfate and nitrate is increased during winter period Interestingly, by comparing  $\Delta^{17}O(\text{nss-SO}_4^{-2})$  and  $\Delta^{17}O(\text{NO}_3)$  values to ozone mixing ratios, we found that the  $\Delta^{17}O(\text{nss-SO}_4^{-2})$  values in spring months were lower than in fall months despite of similar ozone levels for spring and fall, whereas there was no clear difference between  $\Delta^{17}O(\text{NO}_3)$  values.
- Possible explanations for the spring/fall differences for sulfate include: (i) low relative contribution of O3 oxidation, in spring induced by reactive nitrogen emissions from snowpack at inland sites being transported to coastal sites; and (ii) effects of SO2 oxidation by hypohalous acids (HOCl, HOBr), enhanced in spring by interaction of sea salt particles with photo-oxidants. Further observations of Δ17O(nss-SO42-) and Δ17O(NO3-) in aerosols collected at Antarctic inland sites will help us to identify the processes causing such different sulfate and nitrate formation in spring and fall. Nevertheless, the dataset of this study can be dedicated to atmospheric chemical transport models to better constraints on unique local oxidation chemistry at DDUx

**Author contributions**

S. Ishino, S. Hattori and J. Savarino designed the research. M. Legrand, B. Jourdain and S. Preunkert provided qualified
 complementary data, organized the Antarctic field campaign, and collected samples. S. Ishino, S. Hattori, J. Savarino, A. Barbero and N. Caillon performed experiments. S. Ishino, S. Hattori, J. Savarino, K. Kuribayashi, and N. Yoshida analyzed data. S. Ishino, S. Hattori and J. Savarino prepared the manuscript, with contributions from all other co-authors.

| Sakiko ISHINO 7/2/2017 11:48                                                |
|-----------------------------------------------------------------------------|
| the $\Delta^{1'}O(nss-SO_4^{2^-})$ and $\Delta^{1'}O(NO_3^-)$ values is     |
| Sakiko ISHINO 7/2/2017 11:48                                                |
| Sakiko ISHINO 7/2/2017 11:49                                                |
| Sakiko ISHINO 7/2/2017 13:18                                                |
| Sakiko ISHINO 7/2/2017 13:18                                                |
| Sakiko ISHINO 7/2/2017 13:18                                                |
| $\Delta^{17}O(nss-SO_4^{2^-})$ and $\Delta^{17}O(NO_3^-)$ values with ozone |
| mixing ratios.                                                              |
| Sakiko ISHINO 7/2/2017 13:41                                                |
| Sakiko ISHINO 7/2/2017 16:31                                                |
| Sakiko ISHINO 14/12/2016 10:56                                              |
| Sakiko ISHINO 14/12/2016 10:56                                              |
| Sakiko ISHINO 24/12/2016 16:15                                              |
| connecting $\Delta^{17}$ O values with the oxidation processes              |
| involved in sulfate and nitrate formation are                               |
| spatio-temporal patterns of $\Delta^{17}$ O signatures.                     |
| Sakiko ISHINO 7/12/2016 18:22                                               |

**Data availability**

The data used for the figures and the interpretations are shown in Supplement materials.

**Acknowledgments**

- 5 Financial support and field supplies for winter and summer campaigns at DDU were provided by the Institut Paul Emile Victor from Program 414, program 1011 (SUNITEDC) and the French Environmental Observation Service CESOA (Etude du cycle atmosphérique du Soufre en relation avec le climat aux moyennes et hautes latitudes Sud; http://www-lgge.ujfgrenoble.fr/CESOA/rubri-que.php3? id\_rubrique=2) dedicated to the study of the sulfur cycle at middle and high southern latitudes, and supported by the National Centre for Scientific Research (CNRS) Institute for Earth Sciences and Astronomy
- 10 (INSU). This work is also supported by a Grant in-Aid for Scientific Research (S) (23224013) from the Ministry of Education, Culture, Sports, Science and Technology (MEXT), Japan, a Grant in-Aid for Young Scientist (A) (16H05884) of MEXT, Japan, and by a research fund from Asahi Group Foundation. S.H., S. I., and N. Y. are supported by a Japan–France Research Cooperative Program (SAKURA and CNRS) of MEXT, Japan. J. S. and N. C. thank the CNRS/INSU (PRC program 207394) and the PH-SAKURA program of the French Embassy in Japan (project 31897PM) for financial support
- 15 for this collaboration. This work has been supported by a grant from Labex OSUG@2020 (Investissements d'avenir ANR10 LABX56. S. I. thanks the Academy for Co-creative Education of Environment and Energy Science for financial support. We are grateful to Basile de Fleurian who carried out winter sampling at DDU in 2011. The authors gratefully acknowledge Erwann Vince of the Laboratoire d'étude des Transferts en Hydrologie & Environnement forpreparation of bacterial cultures used for nitrate and ozone isotope analyses. Meteo France are acknowledged for providing meteorological
- 20 data. Lars Kaleschke and his team members provided sea ice data. We also acknowledge the Air Resources Laboratory (ARL) for the use of their HYSPLIT transport and dispersion model available on the READY Web site (http://www.arl.noaa.gov/ready.html).

| Table 1: Summary of $\Delta_{k}^{1/0}$ values of sulfate and nitrate produced via each reaction pathways with oxidants. $\Delta^{1/0}$ values of sulfate |
|----------------------------------------------------------------------------------------------------------------------------------------------------------|
| are calculated based on Savarino et al. (2000). $\Delta^{17}$ O values of nitrate refer to estimate of box model by Morin et al. (2011). $\Delta^{17}$ O |
| value of nitrate produced by BrONO2 hydrolysis is assumed to be equal to the value for NO3 + RH pathway.                                                 |

|                                     | Oxidation pathway                                                                                                                                                                                                                  | Δ 17 O (oxidant) (‰) | Transferring factor                                                   | Δ 17 O (product) (‰) |  |
|-------------------------------------|------------------------------------------------------------------------------------------------------------------------------------------------------------------------------------------------------------------------------------|-----------------------------------------------|-----------------------------------------------------------------------|-----------------------------------------------|--|
| SO42-                               | $SO_2 + OH$                                                                                                                                                                                                                        | 0 a                                           | -                                                                     | 0                                             |  |
|                                     | $SO_3^{2-} + O_3$ (aq.)                                                                                                                                                                                                            | 26 b                               | $0.25\times\Delta^{17}O(O_3)_{bulk}$                                  | 6.5                                           |  |
|                                     | $HSO_{3}^{-} + H_{2}O_{2}$ (aq.)                                                                                                                                                                                                   | 1.6 °                                         | $0.50\times\Delta^{17}O(\mathrm{H_2O_2})$                             | 0.8                                           |  |
|                                     | $SO_3^{2-} + O_2$ (cat. Fe, Mn)                                                                                                                                                                                                    | -0.3 d                             | $0.25 \times \Delta^{17} O(O_2)$                                      | -0.1                                          |  |
|                                     | $\mathrm{SO_3^{2-}} + \mathrm{HOX} + \mathrm{H_2O}$                                                                                                                                                                                | $39 (HOX)^e$ , $0 (H_2O)^f$                   | -                                                                     | 0                                             |  |
| NO 3 -        | $NO_2 + OH$                                                                                                                                                                                                                        | 0                                             | $2/3 \times \Delta^{17}O(NO_2)$                                       | 17.3-25.1                                     |  |
|                                     | N 2 O 5 hydrolysis                                                                                                                                                                                           | 26 (O 3 ), 0 (H 2 O)    | $2/3\times\Delta^{17}O(NO_2)+1/6\times\Delta^{17}O(O_3)_{term}$       | 31.0-35.2                                     |  |
|                                     | $NO_3 + RH$                                                                                                                                                                                                                        | 26 (O 3 )                          | $2/3\times\Delta^{17}O(NO_2)+1/3\times\Delta^{17}O(O_3)_{term}$       | 38.0-42.7                                     |  |
|                                     | BrONO 2 hydrolysis                                                                                                                                                                                                      | 39 (BrO)e                                     | $2/3 \times \Delta^{17}O(NO_2) + 1/3 \times \Delta^{17}O(O_3)_{term}$ | 38.0-42.7                                     |  |
| a Holt et a
and Luz ( | a Holt et al. (1983); b Vicars et al. (2014); c Savarino et al. (1999); d Barkan and Luz (2003); e assumed based on Bhattacharya (2008); f Barkan and Luz (2005) |                                               |                                                                       |                                               |  |

